# The climate and vegetation of Europe, North Africa and the Middle East during the Last Glacial Maximum (21,000 years BP) based on pollen data

Basil A.S. Davis[1], Marc Fasel[2], Jed O. Kaplan[3], Emmanuele Russo[4], Ariane Burke[5]

[1]Institute of Earth Surface Dynamics, University of Lausanne, Lausanne, 1015, Switzerland
[2]enviroSPACE lab, Institute for Environmental Sciences, University of Geneva, Geneva, 1211, Switzerland
[3]Department of Earth Sciences, The University of Hong Kong, Hong Kong, Peoples Republic of China
[4]Department of Environmental Systems Science, ETH Zurich, Zurich, 8092, Switzerland
[5]Laboratoire d'Ecomorphologie et de Paleoanthropologie, Departement d'Anthropologie, Universite de Montreal, Montreal, Quebec, H3C 3J7, Canada

*Correspondence to*: Basil A. S. Davis (basil.davis@unil.ch)

**Abstract.** Pollen data represents one of the most widely available and spatially-resolved sources of information about the past land cover and climate of the Last Glacial Maximum (21,000 years BP). Previous pollen data compilations for Europe, the Mediterranean and the Middle East however have been limited by small numbers of sites and poor dating control. Here we present a new compilation of pollen data from the region that improves on both the number of sites (63) and the quality of the chronological control. Data has been sourced from both public data archives and published (digitized) diagrams. Analysis is presented based on a standardized pollen taxonomy and sum, with maps shown for the major pollen taxa, biomes and total arboreal pollen, as well as quantitative reconstructions of forest cover and winter, summer and annual temperatures and precipitation. The reconstructions are based on the modern analogue technique (MAT) with a modern pollen dataset taken from the latest Eurasian Modern Pollen Database (~8000 samples). A site-by-site comparison of MAT and Inverse Modelling methods shows little or no significant difference between the methods for the LGM, indicating that no-modern-analogue and low $CO_2$ conditions during the LGM do not appear to have had a major effect on MAT transfer function performance. Previous pollen-based climate reconstructions using modern pollen datasets show a much colder and drier climate for the LGM than both Inverse Modelling and climate model simulations, but our new results suggest much greater agreement. Differences between our latest MAT reconstruction and those in earlier studies can be largely attributed to bias in the small modern dataset previously used. We also find that quantitative forest cover reconstructions show more forest than that previously suggested by biome reconstructions, but less forest than that suggested by simple percentage arboreal pollen, although uncertainties remain large. Overall, we find that LGM climatic cooling/drying was significantly greater in winter than in summer, but with large site to site variance that emphasizes the importance of topography and other local factors in controlling the climate and vegetation of the LGM.

# 1 Introduction

During the Last Glacial Maximum (LGM) ~21,000 years BP (Mix et al., 2001), the climate, vegetation and landscape of Europe and its surrounding regions were very different than today. Scandinavia and a large part of the British Isles were covered by a single ice sheet, with separate ice sheets covering the Alps and Pyrenees, while many smaller and lower mountainous areas were also glaciated (Ehlers et al. 2011). As a result of this global build-up of ice on land, sea levels were around 120 meters lower than today, resulting in the retreat of Atlantic and Mediterranean coastlines and the emergence on land of the English Channel and North Sea basin. Falling sea levels also led to the disconnection of the Black Sea from the Mediterranean, and a subsequent drop in Black Sea water levels as evaporation exceeded inflow (Arslanov et al. 2007). On land, permafrost and periglacial processes occurred immediately to the south of the Scandinavian ice sheet, while the massive discharge of glacial clays and sands provided material to be redeposited by the wind as belts of loess across northern France, Benelux, Germany and central Europe (Lehmkuhl et al. 2021). Under these cooler and drier climatic conditions, forests are thought to have retreated to the relative shelter of Southern Europe and the Mediterranean, while relatively unproductive steppe and tundra dominated the region north of the Alps (Grichuk 1992).

This traditional view of the LGM has been established for many years, but many details concerning the climate and vegetation of the LGM remain debated. Much of this debate concerns information derived from the pollen record, which represents one of the most widely available and spatially-resolved sources of information concerning LGM vegetation and climate, and the primary terrestrial proxy used to evaluate climate models in the Palaeoclimate Modelling Intercomparison Project (PMIP) (Bartlein et al., 2011; Harrison et al., 2014).

For example, climate model simulations continue to indicate a climate that is less cold and more humid than pollen-based reconstructions (Jost et al., 2005). These results are similar to reconstructions based on glaciological modelling (Allen et al., 2008b). On the other hand, the pollen-based reconstructions that show the greatest disagreement with climate models have themselves been criticized for not considering the possible effect of low atmospheric $CO_2$ on the physiological relationship between plants and climate (Ramstein et al., 2007). Methods that use modern pollen samples are based on the assumption that the relationship between vegetation and climate remains the same through time, and that this is independent of change in $CO_2$ concentration. Studies have shown however that plant growth processes and plant resilience are sensitive to $CO_2$ concentration, and particularly water-use efficiency which would make plants more drought sensitive in low $CO_2$ environments (Cowling & Sykes 1999). Atmospheric $CO_2$ during the LGM was around 190 ppm, some 100 ppm lower than the pre-industrial period, and 200 ppm lower than the levels experienced in the last 50 years. Concerns about the effects of lower $CO_2$ during the LGM has directly led to the development of pollen-climate reconstruction methods that can take account of $CO_2$ effects, either through use of a process-based vegetation model run in inverse mode (Guiot et al. 2000, Guiot et al. 2009), or through the use of a correction algorithm (Prentice et al. 2017). Pollen-climate reconstructions based on inverse modelling that account for these low $CO_2$ effects show less cooling and drying and consequently greater agreement with climate models (Ramstein et al., 2007; Wu et al., 2007).

Further data-model discrepancies have also been highlighted concerning LGM vegetation cover. Earlier pollen synthesis studies, especially those that applied the biomisation method

(Elenga et al., 2000) give the impression that non-glaciated areas of LGM Europe were
dominated by treeless steppe, while vegetation models driven by climate model simulations
indicate large areas of forest and woodlands (Binney et al., 2017; Kaplan et al., 2016;
Velasquez et al., 2021). The apparent data-model discrepancy associated with steppe has led
to the suggestion that early humans, which are not included in vegetation models, could have
reduced the forest cover with only a relatively moderate use of fire because of the cold
climate and slow speed of vegetation recovery (Kaplan et al., 2016). This debate is important
because of studies that have shown the sensitivity of the climate system to vegetation
boundary conditions during the LGM (Ludwig et al., 2017; Velasquez et al., 2021). This
suggests that accurate knowledge of the vegetation cover during the LGM is a necessary
prerequisite to understanding the role of other influences on the climate system at this time.
More recent pollen and macrofossil studies from eastern Central Europe have shown that at
least in this region there existed areas of open boreal forest and woodland with some
temperate broadleaf species (Kuneš et al., 2008; Willis and Van Andel, 2004). The evidence
of forest, and particularly elements of temperate broadleaf forest, north of the Alps has come
to represent a challenge to the traditional view that forest species only survived the LGM in
sheltered refugia far to the south of the Fenoscandian ice sheet and close to the moderating
influence of the Mediterranean Sea. The presence of micro-refugia north of the Alps is
important because it would represent a very different baseline for understanding the later rate
and route of plant migrations under the rapid warming that occurred during the Late Glacial
to Holocene transition (Douda et al., 2014; Giesecke, 2016; Krebs et al., 2019; Nolan et al.,
2018), as well as understanding patterns of present-day genetic diversity (Normand et al.,
2011; Svenning et al., 2008). Modelling studies have shown difficulty in supporting the very
high rates of postglacial expansion that would be necessary for southern refugia (Feurdean et
al., 2013, Nogués-Bravo et al. 2018).
Much of this debate has been informed by an increasing number of LGM pollen studies from
an ever-broader geographical area, and especially from an increasing number of studies from
north of the Alps. Nevertheless, the synthesis of these studies into a single narrative is made
difficult by several factors, for instance: different taxonomic definitions, pollen percentages
calculated from non-standardized pollen sums, and quantitative analyses such as climate
reconstructions that are based on different training sets and methodologies. This has led to
some modelling studies ignoring the pollen record completely, on the basis that data from the
LGM is too scarce (Janská et al., 2017). Where standardized methods have been applied to
multiple LGM pollen records, poor dating control has resulted in the inclusion of many
records that may not actually be from the selected LGM time window. This is particularly
important because the 21 ± 2.0 ka time slice commonly used to represent the LGM period in
PMIP data-model comparisons and other synthesis studies (MARGO members, 2009;
Bartlein et al., 2011) occurs immediately after the glacial maxima in the Alps around 26-23
ka (Heiri et al., 2014; Spötl et al., 2021) and Heinrich stadial HS-2 (24.3-26.5), whilst also
being closely followed by Heinrich stadial HS-1(15.6-18.0 ka) (Sanchez-Goñi & Harrison,
2010). These closely associated time periods can therefore be expected to represent both a
different vegetation and climate than the LGM itself.
For example, of the 18 European pollen records used in the PMIP benchmarking dataset
(Bartlein et al., 2011), 10 fall into the worst class ('poor') in the COHMAP chronological
quality classification scheme if relative dating such as pollen correlation is excluded. More
recent synthesis studies have also relied heavily on records from the European Pollen
Database (EPD) which currently has 116 records with samples of LGM age (as of June
2022). Many of these records however are based on chronologies that are considered reliable
for the Holocene (Giesecke et al., 2014), but have large uncertainties for the LGM as a result
of 1) excessive extrapolation back in time from Holocene age dates, 2) the use of pollen
correlation or other relative dating despite poorly defined regional biostratigraphy, or 3) the
inappropriate use of radiocarbon dates contaminated with old carbon. We found that 104 of
these 116 EPD records (Neotoma, 2021) fall into the worst class ('poor') in the COHMAP
chronological quality classification.
Here we address these problems using a new synthesis of LGM pollen records from
throughout Europe, the Mediterranean and the Middle East (EurMedMidEst) based on
rigorous quality control criteria. Records were compiled from an extensive review of public
databases and archives, and the scientific literature. Pollen records were selected according to
the robustness of their chronological control around the PMIP LGM time-window ($21 \pm 2$
ka), and combined into a single dataset based on a harmonized taxonomy and standardized
pollen sum. The dataset was then analysed so that standardised maps could be produced to
show the distribution of the major pollen taxa, biomes and total arboreal pollen at the LGM.
In addition, quantitative reconstructions of forest cover as well as winter, summer and annual
temperatures and precipitation were undertaken using the Modern Analogue Technique
(MAT), utilizing the latest Eurasian Modern Pollen Database v2 dataset. These climate
reconstructions are compared and evaluated against previous LGM pollen-climate
reconstructions, as well as reconstructions based on other proxies. The dataset and results are
fully documented and the complete data files are provided in the supplementary information.
**2 Methods**
**2.1 Pollen Data**
LGM fossil pollen data from Europe and bordering regions including North Africa and the
Middle East were selected and collated into a single standardized project database. This data
was sourced from the EPD/Neotoma database (Williams et al., 2021), the Pangaea data
archive, publications in scientific journals, and from the original authors. We selected LGM
pollen sites/data according to strict quality control criteria. Where possible, primary raw
pollen counts were used where this was available. Where the original electronic data was not
available, the data was digitized from the published diagram. Overall we have included 63
records in our study, of which 35 were digitized and 28 consisted of the original pollen
counts (Table 1).
The distribution of the 63 sites reflects the distribution of suitable archives, with fewer
records available from climatically or environmentally challenging regions (Fig. 1). High
rates of erosion and a drier and colder climate during the LGM reduced the number of
suitable anoxic sediment sinks for pollen preservation, especially in Central Europe between
the Scandinavian and Alpine ice sheets. Nevertheless, our dataset includes sites from this
region, as well as North Africa and eastern Central Europe through to Iran, although most
sites are located in an arc across eastern Spain, the Alps, and Italy. Lakes sites are the most
numerous archive and tend to be located in the more sheltered and topographically favourable
regions of Southern Europe and the Mediterranean. Peat is the next most important archive,
followed by alluvial and colluvial sediments, as well as cave sites, the later also often being
known for their archaeological significance. Sites located at the ice margins that appear to be
under the ice reflect uncertainties in the location of the ice margin both in time and space
during the LGM, as well as the fact that the selected time window for this study ($21 \pm 2$ ka) is

later than the maximum ice advance in some regions (Hughes and Gibbard, 2015). For completeness, we also include 7 marine records which have the advantage of more continuous deposition and often better dating over the LGM period, but which are prone to taphonomic biases compared to terrestrial records. These biases are discussed later in this section.

LGM pollen records were selected according to a number of quality control criteria, but primary amongst these was the existence of sufficient independent chronological control points to accurately identify samples that would fall within the $21 \pm 2$ ka BP time-slice of interest. We have used all of the samples within this time frame where the samples have been available in electronic form, else we have used the sample closest to the target time (21 ka BP). For records taken from the EPD we have used the latest Bayesian age-depth models where these were available (Giesecke et al., 2014), otherwise we have used the dates and chronology proposed by the original authors. We classified chronologies according to the COHMAP chronological quality scheme for the LGM period (Anderson et al., 1988; Yu and Harrison, 1995), which classifies record quality from 1-6 depending on whether a date falls within 2000 14C years (or less) of the time being assessed, or whether bracketing dates fall within 6000 and 8000 14C years (or less) about the time being assessed (Table A1). Chronologies based on dates that fall outside of these limits fall into COHMAP class 7, and are regarded as 'poorly dated' with respect to the LGM. Importantly, we have only included radiometric and other absolute dates (such as varves) in this assessment, and have excluded dates based on correlation with regional pollen records. These pollen-based stratigraphic dates have been widely used in previous LGM studies, but do not include estimates of uncertainty and are generally regarded as unreliable at this time given the sparsity of well dated pollen sites and samples on which to base any correlation (Giesecke et al., 2014).

All records that were classified as poorly dated (COHMAP class 7) were subsequently excluded from our analysis. This has meant that many of the pollen records used in previous studies were excluded, including 16 of the 26 LGM records used by PMIP and associated studies in Europe (Bartlein et al., 2011; Elenga et al., 2000; Tarasov et al. 2000, Jost et al., 2005; Peyron et al., 1998; Wu et al., 2007; Cleator et al., 2020). We also excluded 104 of the 116 records in the EPD with samples that fall within our LGM time window. Many of these EPD pollen records have been used in more recent studies, although the exact record (EPD Entity number) is often not stated. We estimate that we have excluded 16 of the 17 European sites used by Binney et al. (2017) (this study only included sites above latitude 40N), 5 of the 6 European sites used by Allen et al. (2010), 28 of the 33 sites used by Cao et al. (2019) and 27 of the 71 sites used by Kaplan et al. (2016).

Other quality control criteria were also used in the selection of LGM pollen records. Published pollen diagrams that only included a small part of the terrestrial pollen assemblage, or only presented summary taxa, were excluded. Records were also excluded where the dating information was incomplete, for instance where radiocarbon dating uncertainties were not published or where it was not possible to determine if the date shown was in calibrated or uncalibrated radiocarbon years.

The modern pollen data for the climate and tree cover reconstructions were sourced from the latest version 2 of the Eurasian Modern Pollen Database (Davis et al., 2020), which is managed as part of the EPD. The EMPD2 includes 8133 modern pollen samples from across the Palearctic biogeographic region from Europe to the far East of Asia. The taxa from both the fossil and modern pollen data were consolidated into 120 of the most commonly-

occurring terrestrial taxa types. This taxa list was designed to be compatible with the biomisation scheme used in our study (Peyron et al., 1998; Tarasov et al., 2000) and that used in the Holocene mapping study of Brewer et al. (2017). The count of *Larix* was amplified by a factor of 10 due to its low pollen representation (Edwards et al. 2000, Bigelow et al. 2003, Tarasov et al. 1998, 2000, 2013, Binney et al., 2017).

**2.2 Biomisation**

We converted pollen assemblages to biomes based on the European biomisation scheme of Peyron et al (1998), which in turn is based on Prentice et al. (1996). The method is described in detail in Collins et al. (2012). We expanded the number of taxa included in the biomisation procedure proposed by Peyron et al (1998) to include taxa from the Northern Eurasian biomisation procedure of Tarasov et al. (1998). The inclusion of additional Northern Eurasian taxa reflects recent evidence that modern analogues of LGM vegetation occur in parts of Siberia (Magyari et al., 2014a). The biomisation procedure (Prentice et al. 1996) assigns each taxa to a plant functional type (PFT) and calculates a score for each of these PFT's based on the sum of the square root of the percentage of each of the taxa included in that PFT. To reduce the influence of long-distance transport, taxa below 0.5% are removed at the start of the procedure. Each biome is then assigned one or more PFT's and a score for each biome is calculated as the sum of the associated PFT scores. The biome with the highest score is then viewed as the dominant biome. Where the highest score is the same for more than one biome, the dominant biome is decided based on a hierarchy of unique PFT's. Peyron et al. (1998) also included a procedure for distinguishing warm and cold steppe biomes based on re-assigning certain steppe PFT's according to the presence or otherwise of PFT's indicative of cold or warm conditions. Following the Biome6000 project (Elenga et al., 2000) and Allen et al. (2010), we did not apply this additional procedure and present only the merged steppe biome. In summary, the biomisation procedure categorised 39 arboreal pollen taxa and 39 non-arboreal taxa into 22 plant functional types (PFT's), which were then combined into 12 biomes.

**2.3 Quantitative climate reconstruction**

We reconstructed climate from pollen data based on a standard Modern Analogue Technique (MAT) that used PFT scores to match fossil samples with modern pollen samples (as used by Davis et al., 2003). "Other methods using PFT scores and artificial neural network techniques have been developed to reconstruct the climate of Europe during the LGM from pollen data (Peyron et al. (1998) and Jost et al (2005). PFT scores have been used in previous large-scale European pollen-based climate reconstructions for the Holocene (Davis et al., 2003; Mauri et al., 2014, 2015), where performance was found to be better than the conventional approach based on individual taxa (eg Marsicek et al., 2018). A particular advantage of the PFT approach for the LGM is that it can help overcome problems associated with vegetation (pollen) assemblages that may have no modern analogue (Davis et al. 2003). This can be a problem during the LGM when the climate and environment could be expected to be very different from today, and when many taxa formed unusual vegetation assemblages as a result of their forced retreat to sheltered refugia locations. The problem of modern analogues is also addressed in our reconstruction by using the latest EMPD2 modern pollen dataset. The EMPD2 provides a large number of potential modern analogues for many different LGM vegetation types and climates found today across the Palearctic region. PFT scores were calculated according to the methods outlined already in the Biomisation section, then

normalized so that each sample was proportional to every other sample (Juggins and Birks,
294  2012).
The MAT method was applied using the Rioja program for R (Juggins, 2020). The modern
pollen data was taken from the latest version 2 of the EMPD (as detailed earlier). The
EMPD2 includes 8133 samples, which is considerably larger than the modern datasets used
in previous LGM pollen-based reconstructions. For instance, Peyron et al. (1998) used a
modern pollen dataset of 683 samples, which was updated by Jost et al (2005) to include an
additional 185 samples. These datasets were also mainly taken from the steppes of Kazakstan
and Mongolia, while the EMPD2 covers a much wider area, spanning most of the Eurasian
Palearctic region (Davis et al., 2020). The size and distribution of the modern training set in
climate and vegetation space is important because in order for the method to work
effectively, it is necessary to have samples representative of the likely vegetation and climate
space that could be occupied by the fossil assemblage (Turner et al. 2021, Chevalier et al.,
2020; Salonen et al. 2012, Juggins, 2013).
A known problem with MAT is the role of spatial auto-correlation in providing
unrealistically low estimates of uncertainty (Chevalier et al., 2020; Telford and Birks, 2009).
This results from the fact that closely analogous modern pollen samples can also be located
closely in physical space, and therefore in climate space. To reduce this problem it is possible
to exclude closely located samples from the analogue matching process using a filter based
on a set distance (h-block filter) (Telford and Birks, 2009). While this approach can help,
there are also three main problems associated with it. The first is error substitution, since
removing samples also reduces the number of potential analogues, creating a different source
of error that is not easy to categorise. Secondly, multiple samples taken from the same
location are actually a strength of pollen training sets, since they are more likely to capture
the full range of the assemblage diversity associated with a given climate. Thirdly, current
methods that limit spatial range such as the h-block filter only do so on the horizontal axis,
and do not consider the fact that samples can also be found at different elevations. In hilly or
mountainous regions samples can therefore be excluded because they are closely located in
horizontal space, but in fact they actually occupy very different climates and vegetation
associations, contradicting the logical premise of the h-block filter. It was therefore decided
not to apply this filter.
Uncertainties for the pollen-climate reconstructions were calculated using a standard method
for MAT (Juggins 2020) based on the spread of the climates associated with the best modern
pollen analogues used for each fossil sample. The closer the climates of the best modern
pollen analogues (6 in the case of this study) then the smaller are the calculated uncertainties
assigned to the reconstructed climate of the fossil pollen sample.
Climate reconstructions are presented as anomalies. These have been calculated with respect
to modern climate (1970-2000 average) at each core site location using WorldClim 2 (Fick
and Hijmans, 2017) (Table A2), which was also used to assign the modern climate for the
modern pollen samples in the transfer function (Davis et al., 2020).
**2.4 Quantitative tree cover reconstruction**
It has long been recognized that the proportional representation of individual pollen taxa in a
pollen assemblage does not necessarily reflect the proportion of land area covered by that
taxa in the pollen source area surrounding the sample site (Davis 1963, Gaillard et al. 2010,
Zanon et al. 2018). These differences can be caused by varations in pollen productivity,
differential transport, deposition and preservation of pollen grains, and even the ease or
otherwise of the identification of pollen grains themselves. This can make the interpretation
of pollen taxa percentages difficult, even for relatively simple questions such as the
proportion of forest to non-forest in the landscape.
There have been two main methods developed to account for this quantification problem, one
using a physical modelling technique (PMT) based on estimates of pollen production for
individual taxa (Gaillard et al., 2010), and the other using a MAT very similar to that used in
pollen-climate reconstructions (Williams and Jackson, 2003). Both approaches have been
widely applied during the Holocene in Europe (Zanon et al., 2018), but we know of no
previous study that has applied either of these approaches to the LGM. The LGM presents a
number of challenges, not least the problem of potential missing vegetation analogues, as
well as low atmospheric $CO_2$, which has been shown to influence pollen productivity (Leroy
and Arpe, 2007).
Here we use the MAT to provide quantitative estimates of forest cover, following the
approach of Zanon et al. (2018) who applied this method to the Holocene pollen record of
Europe. We apply MAT in exactly the same way as for the climate reconstructions described
earlier, including the use of PFT scores to match fossil and modern pollen samples. Instead of
modern climate values, we assigned an estimate of modern forest cover to each of our
modern pollen sites. To do this we use a high resolution (~100m) remote sensing dataset
derived from satellite observations (Hansen et al., 2013). Zanon et al. (2018) have shown that
the MAT calibrated in this way gives comparable results to the PMT approach in Europe, at
least for the Holocene. One of the main differences however is that the PMT is designed to
provide estimates of the proportions of different taxa, whereas the MAT (as applied here) is
designed to provide estimates of the proportion of forest cover. Where the PMT can only
reconstruct the proportion of forest forming trees, irrespective of their size, the MAT
(following Zanon et al. 2018) is calibrated specifically to reconstruct forest composed of trees
over 5m tall. This follows the FAO definition of forest as "land spanning more than 0.5
hectares with trees higher than 5 meters and a canopy cover of more than 10 percent, or trees
able to reach these thresholds in situ" (FAO Terms and definitions 2020
http://www.fao.org/3/I8661EN/i8661en.pdf).
**2.5 Maps**
We present our results in the form of maps that include the main physiographic features of
the LGM in the study area. The maps are based on the WGS84 projection. Coastlines reflect
LGM sea level at 120m below present, while ice sheets are based on Ehlers et al. (2011).
Modern national country boundaries are also included for reference.
**2.6 Marine pollen records**
We have included marine pollen records in our analysis for reasons explained below, but it is
important that these records should be viewed with caution, particularly when used for biome
and quantitative MAT reconstructions, and when compared with terrestrial records from
different archives. Biomisation methods have been applied to individual marine pollen
records (Combourieu Nebout et al., 2009), as well as multi-site synthesis studies such as the
ACER project (ACER project members et al., 2017). However, marine records were
specifically excluded from the Biome6000 project (Elenga et al., 2000). Similarly,

quantitative climate methods have been applied to individual marine pollen records (Combourieu Nebout et al., 2009; Fletcher et al., 2010), as well as multi-site synthesis studies (Sánchez Goñi et al., 2005; Brewer et al., 2008; Salonen et al., 2021). However, marine records have also been specifically excluded from other major pollen-climate studies (Cheddadi et al., 1996; Davis et al., 2003; Marsicek et al., 2018), as well as quantitative forest cover reconstructions (Zanon et al. 2018).

Discussion on the advantages and problems associated with marine records can be found elsewhere (Chevalier et al., 2020; Daniau et al., 2019), but are reviewed briefly here where relevant to the methodologies applied in this study. Marine sedimentary records provide continuous and well dated pollen records for the LGM that are often lacking from many terrestrial regions, especially in arid areas with few alternative anaeorobic sediment sinks. Conversely however, pollen source areas for marine sites may be many hundreds of kilometers from the coring site and may be liable to change through time in response to changes in distance to the coastline, rates of river discharge and ocean and atmospheric dynamics. This can theoretically give rise to changes in the vegetation shown in the pollen assemblage recorded at the marine site without any actual change in climate or other environmental pressure. The large and indeterminable source area of marine records also mean that it is difficult to apply quantitative MAT reconstruction methods, not least because the mean climate or forest cover of the source area is almost impossible to determine. In addition, the fossil pollen record and the modern pollen dataset to which it is being compared are composed largely of terrestrial lakes and bog sites with much smaller and more homogeneous source areas. This creates a series of problems, the more obvious of which is the calculation of anomalies, since we cannot assume that the modern climate at the (marine) coring site location is representative of the (terrestrial) source area. In this study we have taken the closest point on land as the modern climate for the calculation of anomalies, but provide the absolute values for all sites so that these can be recalculated if necessary (Table A2). The next problem is that the large source area may capture a combination of different vegetation types that is not going to be represented in a modern pollen dataset based on samples from terrestrial sites with much smaller source areas, for instance a mixture of coastal and mountain vegetation, or even vegetation from different continents (Magri and Parra, 2002). However, in our analysis we did not find any sample from a marine record (or terrestrial record) that did not have a reasonable modern analogue in our training set (chord distance <0.3)(Huntley, 1990), even though we did not adjust the pollen assemblage for the over-representation of *Pinus* (and other Pinaceae) in the marine pollen samples.

Typically, the Pinaceae component is excluded from the terrestrial pollen sum when calculating percentages for marine pollen samples, and in some cases as been excluded entirely from the samples used in marine pollen-climate reconstructions (Combourieu Nebout et al., 2009). The problem with excluding *Pinus* is two-fold, the first is that *Pinus* often represents the main forest forming tree in the Koeppen Csb climate zone on the Atlantic coast where many marine sites are located (García-Amorena et al., 2007), as well as representing the most abundant tree taxa in Europe during the LGM (Figure A3c).

The effect of excluding Pinaceae on the biomisation algorithm and MAT climate reconstruction process has not been widely investigated. We therefore decided to evaluate this problem for 1) biomisation, and 2) pollen-climate reconstruction. In table S3 we show the biomisation results for 8213 modern pollen samples taken from the EMPD2 modern pollen database. Using this as the control, we then artificially varied the amount of Pinaceae (*Pinus*, *Abies* and *Picea*) in the assemblage of each pollen sample and compiled the results

(Table S3). This shows quite clearly that removing all of the Pinaceae has a much more
profound effect on the biomisation process than artificially inflating the amount of Pinaceae
(as might be expected in a marine sample where Pinaceae can be over-represented). Even
when Pinaceae was artificially inflated by as much as 400% of the original value, the biomes
were changed in only 2348 samples, compared to 5860 samples if all the Pinaceae was
removed entirely. In terms of the effects on individual biomes, removing the Pinaceae
considerably increased the amount of CLDE, STEP and TUND, whilst greatly reducing the
amount of XERO, almost eliminating the amount of TAIG, and completely eliminating the
COCO biome. In contrast, the effect of inflating the amount of Pinaceae tended to be more
evenly distributed between the biomes, with the biggest increase seen in TUND and biggest
decrease in STEP. This suggests that even if the over-representation of Pinaceae was quite
extreme in marine pollen samples, the effect on biome classification (and by definition, the
underlying PFT scores) is less than removing Pinaceae completely from the pollen
assemblage.
In a second test, we compared the reconstruction of LGM climate from marine pollen
samples when Pinaceae was included, and excluded. The results are shown in table S4 and
indicate reconstructed temperatures are generally 1-2C cooler, and precipitation slightly
higher when Pinaceae is excluded. The differences between the two methods however are
small, and generally less than half of the uncertainties, suggesting that differences are
statistically indistinguishable when considered in the context of the overall uncertainties.
In summary we find that including Pinaceae in the biomisation process is less likely to lead to
miss-assignment of the biome than excluding Pinaceae, except in extreme cases of over-
representation. Percentages of Pinaceae in the LGM marine samples range on average
between 23-88%, suggesting that while Pinaceae was high at some sites, it does not appear to
completely overwhelm the assemblage as might be expected if over-representation was to be
a significant problem. We also find that including Pinaceae in the pollen assemblage of the
LGM marine pollen samples gives pollen-climate reconstructions that are statistically
indistinguishable from those obtained by excluding Pinaceae from the assemblage. Including
Pinaceae in marine samples also provides compatibility with terrestrial samples, particularly
when calculating and plotting pollen taxa percentages. For these reasons we have included
Pinaceae in the analysis of all marine pollen samples in this study, although it is important to
recognize that Pinaceae in such samples can be subject to over-representation and that the
results presented here from marine sites should consequently be viewed with caution.
**3. Results**
**3.1 Vegetation & Biomes**
Results of the biomisation analysis shows that steppe (STEP) was the most common biome at
the LGM across the study area, occurring at 36 out of 63 sites, indicating that the landscape
was largely dominated by cool temperate grasslands across much of western Central Europe,
central and eastern Mediterranean, as well as North Africa and the Middle East (Fig. 2).
However, at the same time we also find that there were a significant number of sites where
we find that woody and forest biomes occur, more particularly in southern and eastern Iberia,
northern Italy and central eastern Europe. The most dominant of these forest and woody
biomes are taiga (TAIG) in the north, and cool-mixed forest (COMX) and xerophytic
woodlands (XERO) in the south.
As would be expected, the dominance of STEP biomes is generally reflected in low arboreal
pollen percentages across the same areas/sites (Fig. 3 & 4). Exceptions to this rule can be
found at marine sites such as [MD99-2331 site #3] and [MD01-2430 site #58] where STEP is
reconstructed despite arboreal pollen percentages of 71 and 80 percent respectively. This
apparent contradiction illustrates some of the idiosyncrasies of the biomisation method,
especially when applying the method to marine pollen samples. In this case it is important to
remember that the AP% is calculated from the sum of the percentages of each relevant taxa,
but the score for each biome is the sum of the square root of the percentages of each of its
constituent taxa. This results in biomes with taxa with large percentage values scoring
proportionally smaller, and biomes with taxa with small percentage values scoring
proportionally larger. For example, a single taxa at 50% has a square root of 7.07, but the
sum of the square roots of 10 taxa each at 5% is 22.36 even though the sum of the
percentages is the same 50%. This effect can be particularly pronounced in marine pollen
samples because they are usually dominated by a single taxa (*Pinus*) that forms a high
percentage of the total assemblage. Since there are often more non-arboreal taxa than
arboreal taxa in a pollen assemblage, the non-arboreal taxa can dominate in the biomisation
process even if collectively their percentage of the assemblage is a lot less than the arboreal
taxa, resulting in a non-arboreal biome such as STEP having the highest biome score.
Of the main arboreal biomes, Taiga (TAIG) is the dominant biome at 3 sites at the eastern
end of the Alpine ice sheet, as well as at a site just to the north in northern Germany and a
site in Slovakia, while Cool Conifer Forest (COCO) is found at 1 site close to the
Scandinavian ice sheet in Lithuania. Cool Mixed Forest (COMX) is found much more widely
at 8 sites south of the Alps from south-west Iberia to Romania, with Xerophytic Scrub
(XERO) occurring at 8 sites with a similar distribution but not as far east or west. Cold
Mixed Forest (CLMX) occurs at just two sites in Georgia and the Alboran Sea at the far east
and west of the study area, while Warm Mixed Forest (WAMX) is the dominant biome at just
1 site in Southern Spain. We do not record Temperate Deciduous Forest (TEDE), Tundra
(TUND) or Desert (DESE) as the dominant biome at any site at the LGM, although they do
occur as sub-dominant biomes.
An alternative picture of LGM tree-cover is provided by the MAT reconstructions (Fig. 4).
MAT performance statistics for tree cover are shown in table 2, based on an evaluation using
the modern training set. This shows a relatively large root mean square error (RMSE) of
21.03. and an R2 of 0.52 that is not as good as for the MAT climate analysis, but overall the
results are comparable with previous MAT tree cover studies (Zanon et al., 2018). In general,
the MAT values (site average 34%) show forest-cover around 16% less than that suggested
from AP% (site average 50%) (Fig. A1), although sites with very low AP% also show higher
values based on MAT. These differences are consistent with comparisons between MAT and
AP% in Zanon et al (2018), although it should be noted that uncertainties related to the MAT
reconstructions are large (± 23%). Zanon et al (2018) found that the differences between
MAT and AP% were greatest over Northern Europe and in Arctic and sub-Arctic climate
regions that are likely to be comparable to many areas of Europe during the LGM. These
regions today are associated with tree-forming taxa such as Birch that fail to grow to a height
of 5m or more, developing only as shrubs or krummholz forms.
Pollen taxa percentages are shown in supplementary figure A2, and distribution maps of the
33 most common taxa are shown in the supplementary figures A3a-f. Of the 21 arboreal taxa,
*Pinus* generally has the highest values and is the most widespread, being present at all 63
sites. Other acicular arboreal taxa include *Juniperus*, which also has a wide distribution
across EurMedMidEst although at lower values. The rest of the acicular arboreal taxa have
more regional distributions. *Picea* is found mainly to the north of the study region, away from
the Mediterranean, whilst *Abies* is generally found more to the south. *Larix* occurs only in the
central European area including the northern edge of the Po plain just south of the Alps,
whilst *Cedrus* is found mainly across south and west Europe in locations much further north
than its Holocene and modern distribution which is confined mainly to Morocco and Lebanon
(Collins et al., 2012). Temperate broadleaf arboreal taxa which also include cold-tolerant
species such as *Betula* and *Salix* are relatively widely spread across the EurMedMidEst
during the LGM, while less dought tolerant taxa such as *Alnus*, *Carpinus* and *Corylus* are
found more to the south-west through to the north-east. Other temperate broadleaf arboreal
taxa such as *Quercus* (deciduous) and *Ulmus* have a much more southern distribution, with
*Fraxinus*, *Olea*, and *Quercus* (evergreen) being more prevalent in the south-west. In contrast,
*Fagus* occurs more to centre and the east, while *Tilia* is found even in more northern
locations of central Europe. The remaining arboreal taxa are more shrubby and drought
adapted, with *Ephedra* and particularly *Ephedra fragilis* having a southern distribution,
whilst the more cold adapted *Hippophae* being found even in the north of central Europe
(similar to *Tilia*).
The main non-arboreal taxa generally indicate cool, dry and environmentally disturbed
conditions across much of the EurMedMidEst. The most widely distributed taxon is Poaceae,
which like Pinus, is found in all records. Other non-arboreal taxa with a widespread
distribution include Rubiaceae, Apiaceae and Asteraceae (Asteroideae), while *Plantago*,
Cayophyllaceae, Brassicaceae and Asteraceae (Cichorioideae) have a more southern and
western distribution. *Thalictrum* can be found mostly at sites in the centre of the
EurMedMidEst, along with *Helianthemum* which also extends to sites in the south-west.
Other taxa such as *Chenopodiaceae* and *Artemisia* have a more southern distribution,
reflecting their preference for drier and less cold climates.
**3.2 Climate reconstruction evaluation**
Evaluation of transfer function performance based on the modern training set is presented in
table 2. This shows that root mean square error predicted (RMSEP) values were smallest for
summer temperatures (2.21C), and largest for winter temperature (3.35C), with mean annual
temperatures in between (2.28C). The weaker performance for winter temperatures largely
reflects the much greater range of winter temperatures in the training set. In turn, this
contributes to a better R2 performance for winter temperatures (0.91) than annual
temperatures (0.9) and summer temperatures (0.81). Overall R2 performance for precipitation
is weaker than for temperature, which is typical because of the higher spatial variability of
precipitation compared to temperature. Summer precipitation has the strongest R2
performance (0.75) compared to winter and annual precipitation (both 0.69), as well as
smaller RMSE values (52mm) than winter (78mm).
Given the widespread occurrence of steppe during the LGM, we also undertook a separate
evaluation of transfer function performance in this type of environment. For this we used a
subset of 1588 pollen samples from the EMPD2 that are classified with the steppe pollen-
biome (Davis et al. 2020). The results indicate (Table A5) little difference in performance
compared to the full dataset, with a small decrease in performance in annual and summer
seasons in both precipitation and temperature, and a slight increase in performance in winter.

The results overall indicate good transfer function performance especially for temperature,
and are comparable with those found in other continental scale pollen-climate studies
(Bartlein et al., 2011). It is important to remember though that comparisons between studies
can only be made with caution because results are often heavily dependent on the nature of
the modern pollen dataset used as the training set, which is not the same in all studies
(Juggins, 2013).
**3.3 Climate reconstruction**
Reconstructed LGM temperatures indicate an overall mean annual cooling of -7.2 ± 3.3C,
with a greater cooling of around -9.3 ± 4.5C in winter and -5.0 ± 3.2C in summer (Fig. 5). All
sites apart from Lake Van [site #62] in eastern Turkey show cooler temperatures at the LGM
compared to modern (Fig. 6), and even at this site cooler conditions fall within the
uncertainties. With greater cooling in winter compared to summer, the difference in
temperature between winter and summer also increased (shown by positive anomalies) at
most (but not all) sites (Fig. 6). This increase in continentality was around +4.2C on average
across all sites (Fig. 5).
We reconstruct an overall decline in mean annual precipitation of around -91 ± 270mm (-
13%) at the LGM. Most of this decline is in winter (-38 ± 90mm) (-21%), while in summer a
small increase is shown (10 ± 57mm) (6%), although uncertainties are large (Fig. 7).
Compared to temperature there is significant seasonal and spatial variability in positive and
negative precipitation anomalies (Fig. 8). Positive anomalies appear more predominant in
eastern and southern Spain and in central eastern Europe in both summer and winter, while
positive anomalies are found more generally in summer across sites in Southern Europe and
the Mediterranean. These more positive summer anomalies also reflect a relative shift from
winter to summer in the seasonality of precipitation in this region.
**4.0 Discussion**
Before we consider the results of our analysis it is important to provide some context in terms
of European LGM geography and environment, which was very different from today (Fig. 1).
Major ice sheets covered Scandinavia and much of the UK, the Alps, and the Pyrenees. Sea
level was 120m lower, resulting in much of the North Sea and English Channel becoming dry
land, and the European coastline extending over 100 km out into the Atlantic and
Mediterranean, especially around the Bay of Biscay and Adriatic. The Black Sea was no
longer connected to the Mediterranean, and was smaller with a water level around 100m
lower than today (Genov, 2016). These changes in sea or water level had two main
consequences, the first being that the marine sites were closer to land, and therefore closer to
(low lying) terrestrial vegetation and (pollen carrying) river discharge points than they are
today. The second consequence of lower seas levels is that terrestrial pollen sites were
located further from the moderating effect of the ocean than they are today, resulting in a
localised modification of the climate experienced by the site irrespective of regional or global
changes (Geiger, 1960).
The maps used in our analysis shows the maximum ice sheet at 21k ± 2k (Ehlers et al., 2011).
The precise geographical location of the ice sheet is difficult to resolve at a fine spatial scale,
however, which explains why some sites close to the ice margin appear to be actually located
under the ice (for example sites Kersdorf-Briesen site #46 & Mickunai site #54). The
resolution of the map also shows the occurrence of permanent ice not only to the north and
over the Alps, but also on many subsidiary areas of high ground across central and southern
Europe, including areas such as the Pyrenees, Massif Central, Vosges and Carpathian
Mountains. While global ice volume may have peaked ~21 ka individual ice sheets in Europe
and other areas are known to have reached their maximum extent at different times (Hughes
et al., 2016). The larger ice sheets are likely to have had a significant influence on regional
climate and environmental conditions across Europe, but the smaller ice sheets had similar if
more localized impacts as well. Surrounding each ice sheet would have been an unglaciated
area of active peri-glacial processes and newly created and unstable ground. This would
include outwash plains, impounded lakes and recently drained lake beds, seasonally and
sporadically flooded areas, moraines, kettle holes and other glaciological and peri-glacial
features. Soils in these areas would be non-existent or skeletal, and vegetation would find it
difficult to obtain nutrients and water for survival, irrespective of the prevailing climatic
conditions. Outside of these areas, permafrost is also likely to have been present, particularly
north of the Alps (Vandenberghe et al., 2014), which would also act as an impediment to
vegetation growth.
In terms of regional climate, the major ice sheets would have provided significant barriers to
westerly atmospheric circulation, or even north-south circulation in the case of the Alps and
Pyrenees. As well as representing a physical obstruction, the thermodynamic response of the
atmosphere to these high, cold obstructions would have been to encourage the formation of
areas of semi-permanent high pressure, similar to those found today for instance over the
Greenland ice sheet. In addition, the Laurentide ice sheet located over North America would
have generated downstream effects over Europe (COHMAP, 1988). These physical and
thermodynamic effects would have affected the direction of storm tracks, as well as more
local climatic effects commonly associated with ice sheets such as strong katabatic winds
(Kageyama,et al. 2021, Velasquez et al. 2021, Luetscher et al. 2015, Lefort et al. 2019)
**4.1 Vegetation Cover**
The nature and extent of forest cover during the LGM remains a matter of considerable
debate. Vegetation models driven by LGM climate model simulations generally indicate
extensive areas of boreal forest north of the Alps, and a mix of temperate and warm-
temperate woodland to the south across southern Europe and much of the Mediterranean.
Treeless areas such as steppe are mainly confined to those areas where it is also found today,
namely inland Iberia, Ukraine, southern Russia and Turkey, while Tundra is found to the
north close to the Scandinavian Ice Sheet (Allen et al., 2010; Cao et al., 2019; Prentice et al.,
2011; Velasquez et al., 2021).
Evaluation of these vegetation-model simulations against data has been largely based on
comparison with compilations of pollen-biome reconstructions (Prentice et al., 2011; Allen et
al., 2010; Cao et al., 2019; Velasquez et al., 2021). Early studies were based on only a limited
number of sites from southern Europe, and showed steppe at all sites in contradiction with
model simulations (Elenga et al. 2000). More recent pollen compilations have included more
sites especially to the north that have revealed a more mixed picture of vegetation cover, with
forest biomes at some sites both south and north of the Alps that appear more consistent with
model simulations (Binney et al., 2017; Cao et al., 2019). However, many of these pollen
sites used in these studies were assigned an LGM age based on poor or incorrect dating
control, and likely date to MIS3, the Late-Glacial or even the Holocene. Nevertheless, based
on our compilation of more securely dated LGM pollen sites, we also show a wider
distribution of forest biomes particularly in Iberia, northern Italy and Central Europe,
although with greater areas of steppe than suggested by the models over the remaining
regions.
However, the interpretation of biome reconstructions requires care since the forest cover and
vegetation composition may not be as clear as the dominant biome suggests. For instance, we
find that steppe is still reconstructed as the dominant biome at some sites despite arboreal
pollen forming 70-80% of the pollen assemblage. In addition, it is important to remember
that pollen-biomes are based only on the proportion of taxa that can form forest and
woodland, while these taxa may in fact exist only as shrubs or stunted krummholz forms in
the challenging climate and environment of the LGM. Alternatively, similar conditions may
favour low-lying non-arboreal taxa forms with poor pollen dispersal or even insect
pollinated taxa forms that may be poorly represented in the pollen assemblage, giving greater
prominence to arboreal taxa whose pollen may be the result of long-distance transport
particularly *Pinus*. However there also appear to be plenty of samples with low or even very
low (<20%) arboreal percentages, so not all sites in open areas may be affected by long-
distance transport of *Pinus* in the same way.
Quantitative MAT based reconstructions of forest cover can overcome some of these
problems, where they can be detected, based on the composition of the pollen assemblage
when compared with the modern land-cover. Chord-distance measurements of the match
between fossil and modern pollen assemblages indicate good LGM analogues exist in our
large Eurasian modern pollen dataset. The results of the MAT forest cover reconstruction
indicates that forest cover was low but not entirely devoid of woodland in most areas, similar
to the modern boreal forests of Siberia and consistent with a steppe-tundra-woodland mosaic
proposed by many authors (e.g. Birks and Willis, 2008; Willis and Van Andel, 2004). This is
confirmed in an analysis of the most commonly found modern analogue ecoregions for LGM
pollen samples at each site (Table A6). Uncertainties are large, but for comparison the MAT
site-average of 33% forest cover is slightly less than the average today over the Boreal region
of Europe (43%) and slightly more than the average today over Mediterranean region (27%)
(Zanon et al. 2018).
By calculating the percentage of each of the taxa in each LGM pollen sample using a
standardized pollen sum, we are able to make direct comparisons between different LGM
pollen records and their taxa percentages (Figure A2, A3). The results show a preponderance
of boreal forest taxa to the north of the Alps, consistent with biome results mentioned earlier.
*Pinus* is the most common forest forming taxa in this boreal zone, together with *Picea*, and
including *Larix* to the east and *Abies* to the west. The occurrence of *Betula* and *Juniperus*
also suggests shrubby elements consistent with arctic shrub-tundra, although high Poaceae
and other herbaceous taxa such as *Artemisia* and *Chenopodiaceae* indicate more steppe than
tundra. Other deciduous taxa found north of the Alps include cold tolerant generalists such as
*Corylus* and *Alnus*, as well as low percentages of relatively thermophilous taxa in the east,
such as *Carpinus* and *Tilia*.
These results are consistent with charcoal (Magyari et al., 2014a; Willis and Van Andel,
2004), malacological (Juřičková et al., 2014), biomarkers (Zech et al., 2010) and genetic
evidence (Stivrins et al., 2016; Willis and Van Andel, 2004) that the main forest region north
of the Alps was in the eastern region of Central Europe around the Carpathian basin. This
was also an area where cold and moisture sensitive deciduous taxa were also able to survive
(Magyari et al., 2014), although evidence of temperate taxa found in the pollen record has yet
to be supported by charcoal and macrofossil records (Feurdean et al., 2014). Our pollen
evidence indicates an open taiga or cool mixed forest that extended in central and eastern
Europe to areas close to the Scandinavian and Alpine ice caps, as proposed by Willis and Van
Andel (2004) and Huntley and Allen (2003), although whether this represents isolated
pockets of forest or an extended open steppe-forest is difficult to determine (Kuneš et al.,
2008). Even steppe or tundra areas in western Europe show a low but significant presence of
the pollen of tree taxa at sites close to the ice sheets that are unlikely to be solely the result of
long distance transport or reworking (Kelly et al., 2010). The presence of woodland in these
areas is also supported by mammalian remains, for instance at Kents Cavern in SW England
(Stewart and Lister, 2001).
Overall however, our results clearly show a much greater predominance of thermophilious
and moisture sensitive deciduous taxa south of the Alps, particularly in Iberia and Northern
Italy, where temperate broadleaf forests survived in sheltered refugia (Kaltenrieder et al.,
2009). Most of these appear to be in hilly areas with the ability to generate orographic rainfall
(Monegato et al., 2015), on south facing slopes to make the most of the sun's radiant energy
and located above the valley floor to escape frost and flooding. We might also expect these
areas to be sheltered from cold northerly winds, and benefit from relatively mild and moisture
laden winds coming from the Mediterranean Sea. For instance, the presence of woodland and
low glacier altitudes along the southern slopes of the Alps around the Po Valley and Trentino
region is consistent with strong orographic rains generated by southerly and easterly winds
that today can be generated by low pressure located south of the Alps in the Gulf of Genoa,
and consistent with a southerly storm track around the Alps (Kehrwald et al., 2010; Luetscher
et al., 2015). Generally, as might be expected, areas of forest reconstruct similar or increased
precipitation compared to today, and areas of steppe indicate deceased precipitation (see next
section).
Independent evidence of LGM vegetation is provided by archaeozoological data. This data
supports the palynological evidence for the existence of forest and woodland refugia across
the ice-free areas of Europe at latitudes north of the Alps. For instance, large vertebrates in
these areas show patterns of extirpation and extinction in response to shifts in climate and
vegetation cover that is different for different species, indicating a variety of environments
and niches (Lister and Stuart, 2008; Stewart and Lister, 2001). As with the pollen record, the
presence of temperate adapted large vertebrate taxa within the glacial landscape of Western
Europe also suggests the existence of temperate "micro-refugia" (Stewart and Lister, 2001) ,
consistent with suggestions that temperate arboreal taxa were not entirely extirpated from the
region during the LGM (Magri, 2010). Further east, mammal assemblages indicate
generalized loss of forest components in the East European Plain (Demay et al. 2021,
Puzachenko et al., 2021) which is consistent with our data indicating low forest cover in this
region. In other areas, evidence of the prevailing land cover at the LGM comes from studies
of small vertebrate communities, which have a closer affinity to the prevailing environment
than large vertebrates (López-García and Blain, 2020) that have the propensity to migrate
large distances, often on a seasonal basis. These studies of small vertebrate assemblages also
support the existence of temperate "micro-refugia" in France (Royer et al., 2016) and the
existence of woodland components in many regions across Southern Europe including parts
of Iberia (Bañuls-Cardona et al., 2014) Italy (Berto et al., 2019) and the Balkan Peninsula
(Mauch Lenardić et al., 2018).
Other paleobotanical evidence also supports our land cover reconstruction. Schafer et al.
(2016) suggest leaf wax patterns from palaeosols in Spain may indicate the presence of

drought intolerant deciduous trees and more humid conditions during the LGM. Significantly, none of the pollen sites indicate that temperate broadleaf forests were dominant, and broadleaf temperate taxa always appear part of a mixed woodland together with cold or aridity adapted evergreen and needleaf taxa, including typical Mediterranean taxa. This type of mixed vegetation probably extended to the Balkans where the hilly terrain and proximity to the Mediterranean would appear to have provided favourable climatic conditions, although we still lack LGM sites from this region. At sites in central and southern Italy and east through Greece and Turkey to the Middle East (and including North Africa), the vegetation appears drier with a greater prevalence of steppe. Only a site in Georgia at the edge of the Caucasian mountains indicates the presence of significant amounts of forest (mainly *Pinus*), a result that was also found by Tarasov et al. (2000), and probably linked to favourable orographic precipitation and proximity to the Black Sea.

Comparison with LGM land cover from vegetation modelling studies driven by climate model simulations indicate a much wider presence of forest than that shown by the pollen data (Kaplan et al., 2016). Data-model agreement appears to be closest over eastern-central Europe where pollen indicates the presence of open Boreal forest, and over south-west Europe with the presence of cool mixed temperate forest, including broadleaf deciduous and thermophilious elements (Prentice et al., 2011; Allen et al., 2010; Cao et al., 2019; Velasquez et al., 2021). Nevertheless, agreement still appears to be weak over western-central Europe and Southern and Eastern Europe through to the Middle East, where pollen data continues to indicate widespread steppe. One proposed explanation for this data-model discrepancy has been the role of fire (including man-made fire) in maintaining forest openness, a factor influencing forest cover that is not included in most vegetation models (Kaplan et al., 2016). In the Carpathian basin Magyari et al. (2014a) noted that charcoal increased as forest cover declined, suggesting that wildfires played a role in decreasing forest cover during the LGM. Other studies have noted low levels of charcoal and therefore fires during the LGM, although these tend to be from steppe areas with low biomass and fuel availability (Connor et al., 2013; Kaltenrieder et al., 2009). Recent LGM vegetation simulations that include fire indicate much lower values of forest cover than those without fire over western central Europe, while forest remains in central eastern Europe (see figure 6 in Velasquez et al., 2021). This appears closer to the data, but the values are perhaps too low compared with our MAT reconstructions here (Figure 4).

**4.2 Climate**

**4.2.1 Comparison with previous pollen-based reconstructions**

The climate of the LGM is generally considered to have been cooler and drier than today, but data-model comparisons continue to highlight important discrepancies, not only in the degree of cooling and drying but also in their seasonal and spatial distribution. Data-model comparisons over Europe have mainly used pollen-based climate reconstructions, especially the Paleoclimate Modelling Intercomparison Project (PMIP/CMIP) (Kageyama et al., 2021, Bartlein et al., 2011; Harrison et al., 2015; Kageyama et al., 2006). The most commonly used reconstructions have been based on two main methods, a neural-network methodology (ANN) of Peyron et al. (1998) and Jost et al. (2005), and an Inverse Modelling approach (INV) applied by Wu et al. (2007). The ANN method uses modern pollen samples and does not include any correction for CO2 effects, being similar in these respects with the MAT method used in this study. In contrast the INV method does not use modern pollen samples, but instead uses a process-based vegetation model run in inverse mode. Ordinarily, a

vegetation model will use climate as an input to generate a vegetation as an output, but in
inverse mode the model is reconfigured to generate climate as an output given a particular
vegetation (pollen) assemblage as an input. One of the advantages of the INV method is that
$CO_2$ can also be varied as an input, and therefore the effect of changes in $CO_2$ on the
vegetation, and therefore reconstructed climate, can be investigated. Comparison of these
ANN and INV reconstructions have shown important differences, with the INV
reconstruction generally not as cold and somewhat drier than ANN (Wu et al. 2007). These
differences between pollen-climate methods have often been attributed to $CO_2$ effects (Wu et
al. 2007) but this is not clear since there may be other factors, such as the size and location of
the training set used in the ANN reconstruction.
We make a comparison with these earlier reconstructions based on 10 sites/records in our
dataset which we identified as also being included in these earlier studies (Fig. 9). While we
were able to identify the site and data source, as well as the time window, we were unable to
establish if the the data represented a single sample or the mean of multiple samples within a
time-window or the exact depth of those samples, or the actual sediment core in the case of
multiple cores from the same site. While these aspects are unknown, it seems likely that the
pollen data we used in our analysis was very similar if not identical in most cases, and
reconstructed biomes for these sites from our pollen dataset are identical to the biomes
reconstructed using the earlier pollen dataset (Elenga et al., 2000).
We compare our MAT with the ANN and INV reconstructions in figure 9. On average across
all 10 records, the MAT and INV methods give almost identical results for both anomalies of
mean annual temperature (MAT -6.6C, INV -7.2C) and precipitation (MAT 158mm, INV
165mm). Uncertainties are also similar for both methods. In contrast, the ANN method gives
much cooler mean annual temperature anomalies (ANN -13.9C) and drier precipitation
anomalies (ANN -474mm). On a site by site basis the MAT and INV methods show closer
agreement for temperatures than precipitation, although precipitation has proportionally
larger uncertainties. The reconstructions based on these two methods are close enough that
the uncertainties overlap at all sites for both temperature and precipitation, except the
precipitation reconstruction at Lac de Bouchet (site #25). The reason for this is not clear, but
there could easily be minor differences with the pollen data analysed by Wu et al. (2007) in
their INV reconstruction since the pollen record (Reille and de Beaulieu, 1988) includes
multiple cores each with many different samples covering the LGM period.
This comparison shows that our MAT reconstructions are very similar to the INV method,
but not as cold or dry as the ANN method. This has two main implications. The first is that
our reconstructions indicate greater agreement with the results of climate model simulations
since climate models indicate temperatures closer to the INV reconstructions (Latombe et al.,
2018) than the ANN reconstructions (Jost et al., 2005; Kageyama et al., 2006). The difference
between our MAT and earlier ANN reconstructions is likely the result of the modern pollen
datasets used, since the ANN reconstruction was based on a considerably smaller number of
samples taken mainly from the cold dry steppes of Kazakstan and Mongolia.
The second implication is that the MAT method may not be significantly impacted by the
effects of lower $CO_2$ (Cowling and Sykes, 1999; Prentice and Harrison, 2009; Williams et
al., 2000) or indeed insolation changes during the LGM, since the MAT results are similar to
those based on the INV method which specifically takes account of these non-climatic factors
(Wu et al., 2007). This would suggest that MAT could also work well for pollen-based
climate reconstructions on longer glacial-interglacial timescales where insolation and $CO_2$

vary significantly from their modern values. This is consistent with the findings of Pini et al. (2021) who applied a correction algorithm developed by Prentice et al. (2017) and Cleator et al. (2020) to a MAT reconstruction of mean annual precipitation at Lake Fimon in Northern Italy. This shows a very small correction of 0mm to 30mm for samples across the LGM time-window, which indicates that $CO_2$ is not a very significant factor in influencing this type of reconstruction, at least compared to the overall uncertainties (+/- 200mm) of the reconstruction itself. The uncertainties associated with the correction algorithm are not discussed, but given that inputs include estimates of both LGM temperature and cloud cover, it seems likely that these could be significant. Importantly, both Pini et al (2021) and Cleator et al (2020) specifically exclude the necessity of applying a correction algorithm to temperature reconstructions, since they consider only hydrological variables to be affected by changes in atmospheric $CO_2$.

### 4.2.2 Comparison with climate reconstructions based on other proxies

### 4.2.2.1 Temperature

Proxies that are not based on plants should remain unaffected by the $CO_2$ problem during the LGM, and provide an alternative basis for evaluating pollen-based reconstructions. Samartin et al. (2016) reconstructed LGM summer temperatures based on chironomid remains from Lago della Costa (site #34) in Northern Italy. They also undertook pollen analysis on the same samples down the core, allowing us to make a sample-by-sample comparison between the chironomid temperature record and our MAT reconstruction (Fig. 10). Our pollen-climate reconstruction is for JJA mean temperate, while the chironomid reconstruction is for July mean temperature, with the anomalies based on the modern equivalent JJA and July mean temperatures respectively. The average anomaly values for all 8 samples reconstructed by the pollen-climate MAT are -10.2 ± 3.5C, and for the chironomids -9.5 ± 3.0C. This indicates that pollen and chironomid average summer temperature reconstructions are very similar on average, taking into account the overlapping uncertainties, while also showing a strong similarity on a sample-by-sample basis throughout the time-series.

Other reconstructions based on other proxies provide a basis for more general regional comparisons (Figure A4, A5). We reconstruct both summer and winter temperatures and show that cooling in winter was greater than in summer at most sites, associated with an increase in continentality (increased temperature difference between summer and winter). A similar seasonal pattern of temperature change has also been shown in other studies that reconstruct both summer and winter LGM temperatures, including Prud'homme et al. (2016) using d18O analysis of earthworm calcite granules at Nussloch near the French-German border, Bañuls-Cardona et al. (2014) using faunal remains of small mammals at 4 locations in western Spain, and Ferguson et al. (2011) who examined seasonal temperature change using d18O and Mg/Ca analysis of limpet shells at Gibraltar in southern Spain. The increase in continentality at Nussloch (Prud'homme et al., 2016) was reconstructed at between 11.6 to 15.6 ºC, comparable at the lower end with nearby pollen sites [La Grotte Walou site #28] 10.4 ± 5.8 ºC and [Bergsee site #29] 7.9 ± 5.7 ºC. The faunal sites in western Spain studied by Bañuls-Cardona et al. (2014) gave much reduced increases in continentality, but nevertheless similar to nearby pollen sites. For instance at Valdavara 5.1 ºC [MD99-2331 site #3] 5.2± 3.1 ºC , El Miron 1.2 ºC [Tourbiere de l'Estarres site #19] 5.1 ± 6.2 ºC, El Portalon 0.9C [Torrecilla de Valmadrid site #16] 2.8 ± 1.8 ºC and Cueva de Maltrvieso 6.1C [SU81-18 site #2] 4.8 ± 3.4 ºC. Further south at Gibraltar the limpet-based study of Ferguson et al.

(2011) also shows a relatively small increase of 2 ºC. The nearest pollen site [Gorham Cave
site #5] however shows a larger increase of 4.7 ± 2.3 ºC, although differences could be
expected given the different temporal resolution of annual laminae on mollusk shells
compared to pollen assemblages that reflecting much slower changes in trees and other long-
lived flora.
Summer temperatures were warm enough during the LGM over the Alpine areas that Swiss
lakes were largely ice free in summer, while glacier ELA's around the time of the LGM
suggest summers were -6.5 to -7.7 ºC cooler compared to the LIA (Heiri et al., 2014). This
cooling was similar to that found at Nussloch some 200km north of the Swiss border by
Prud'homme et al. (2016), who reconstructed anomalies of -6 to -8 ± 4 ºC from d18O
analysis of earthworm calcite granules (representing warm season May-September
temperatures). Slightly less cooling was found close by at the nearby site of Achenheim
where analysis of Mollusc assemblages gave summer (August) cooling estimates of -3.5 to -
6.5 ºC based on MAT (Rousseau, 1991), and -5.5 to -9.5 ºC based on the Mutual Climatic
Range method (Moine et al., 2002). These reconstructions appear somewhat cooler than
nearby pollen sites [La Grotte Walou site #28] -1.4 ± 3.6 ºC and [Bergsee site #29] -2.7 ± 5.1
ºC, although comparable with the pollen site [Pilsensee site #32] -7.3 ± 5.0 ºC 200 km further
east. Similar differences also occur at the site of Les Echets on the western edge of the Alps
where a diatom based reconstruction of summer (July) temperatures (Ampel et al., 2010)
indicated a greater cooling (-10.5 to -11.5 ºC) than our pollen reconstruction [Les Echets G
site #27] (-4 ± 2.7 ºC). However, the authors caution that the results were based on poor
analogues and rare taxa, as well as a small training set of only 90 lakes in Switzerland.
South of the Alps, other proxies show the opposite relationship with the pollen
reconstructions. For instance, at Lago dela Costa in the Po valley, a summer (July)
temperature chironomid reconstruction by Samartin et al. (2016) is around 1-2 ºC less cool
than the pollen reconstruction (JJA) for the same site [Lago della Costa site #34] -11.4 ±
2.7C, although both reconstructions fall within their respective uncertainty ranges (Figure 8).
In the Pindus Mountains in Greece, Hughes et al. (2006) estimated LGM summer
temperature anomalies of - 7 ºC based on glacier modelling, which is comparable with that
reconstructed at the nearest pollen site [Ioannina site #51] -7.7 ± 2.8 ºC. In Spain the analysis
of small mammal remains by Bañuls-Cardona et al. (2014) shows similarly less cooling in
summer or even warmer than present positive anomalies compared to the nearest pollen sites,
such as Valdavara 1.4 ºC [MD99-2331 site #3] -2.3 ± 2.8 ºC , El Miron -2.3 ºC [Tourbiere de
l'Estarres site #19] -5.7 ± 5.4 C, El Portalon 0.8 ºC [Torrecilla de Valmadrid site #16] -2.6 ±
1.1 ºC and Cueva de Maltrvieso -1.1C [SU81-18 site #2] -10.4± 2.8 ºC. Further south at
Gibraltar, the limpet-based study of Ferguson et al. (2012) suggests an anomaly of around -7
ºC, which is a greater cooling than the pollen reconstruction from this location [Gorham Cave
site #5] -1.3 ± 2.2 ºC, although comparable with other pollen sites slightly further east.
Winter temperature reconstructions from non-pollen proxies show a similar pattern in relation
to pollen reconstructions as for summer temperatures. North of the Alps at Achenheim,
Prud'homme et al. (2016) use d18O on earthworm remains to reconstruct particularly cold
winter anomalies of -17.6 to -23.6 ºC compared to nearby pollen sites [La Grotte Walou site
#28] -11.8 ± 8.0 ºC and [Bergsee site #29] -10.6 ± 6.3 ºC. South of the Alps in Spain, the
analysis by Bañuls-Cardona et al (2014) based on the remains of small mammals shows less
cooling in winter compared to the nearest pollen sites, in particular Valdavara -3.7 ºC
[MD99-2331 site #3] -7.5 ± 3.4 ºC , El Miron -3.5 ºC [Tourbiere de l'Estarres site #19] -10.8
± 7.0 ºC, El Portalon -0.1 ºC [Torrecilla de Valmadrid #16] -5.4 ± 2.5 ºC and Cueva de
Maltrvieso -7.2C [SU81-18 site #2] -15.2 ± 4.0 ºC. And again, in southern Spain at Gibralter,
analysis of limpet shells by Ferguson et al (2011) suggests winter cooling of around -9 ºC
while the pollen reconstruction suggests [Gorham Cave site #5] -6.0 ± 2.5 ºC, although sites
further east indicate cooler conditions.
A number of additional proxies have also been used to reconstruct LGM mean annual
temperature. Heyman et al. (2013) applied glacier mass balance modelling at sites located in
the smaller mountain regions north of the Alps. These are generally slightly cooler than our
pollen-based reconstructions at sites close to the Vosge Mountains -12.7 ± 2.0 ºC and Black
Forest -11.4± 2.3 ºC [Bergsee site #29] -8.2 ± 3.3 ºC, Bavarian Forest -10.7± 2.2 [Pilsensee
site #32] -9.2 ± 1.2 ºC and Giant Mountains -8.5 ± 1.8 [Kersdorf-Briesen site #46] -7.3 ± 0.3
ºC. These values obtained by Heyman et al. (2013) are warmer than Pud'homme et al. (2016)
who estimated annual mean temperature anomalies of -15.1 to -19.1 ºC based on d18O of
earthworm calcite at the Nussloch site just north of the Vosge and Black Forest. The annual
temperatures reconstructed by Heyman et al. (2013) are also around 2C warmer than Allen et
al. (2008) who applied a similar, although simpler method to over 29 different mountainous
regions across Europe that had been glaciated during the LGM. Since glacier mass balance is
a function of both snowfall and temperature, these estimated temperatures vary according to
estimated changes in precipitation. For instance, mean annual temperature estimates by Allen
et al. (2008a) are much cooler than reconstructed by pollen, with an average anomaly of -13.2
ºC for the 29 sites assuming a 40% reduction in precipitation, but this is reduced to -11.8 ºC
assuming the same precipitation as modern. This compares with -7.2 ºC for our 63 pollen
sites. The glacier mass balance modelling by Allen et al. (2008a) assumes a seasonal
distribution of precipitation that is similar to the present day, and does not consider increases
in winter precipitation or mean annual precipitation above present day levels. Both of these
are suggested by the pollen data in some regions, and both could explain glacier extent found
during the LGM based on less extreme temperature anomalies more comparable with the
pollen data.
To the east of the Alps in the Panonian basin, mean annual temperature anomaly estimates
have been made from noble gas measurements on groundwater ranging from -2 to -4 ºC
(Stute and Deak, 1990) up to -9 ºC (Varsányi et al., 2011). These are similar to estimates
ranging from -2 to -9 ºC from oxygen isotope ratios from mammoth tooth enamel (Kovács et
al., 2012) and are comparable with nearby pollen sites [Feher Lake site #50] -8.2 ± 3.3 ºC and
[Kokad site #52] -4.5 ± 2.3 ºC. On a broader scale, Sanchi et al (2014) estimated LGM
cooling in the Danube and Dneiper basins based on Lipid biomarkers in a core from the
Black Sea and came up with similar mean annual temperature anomalies between -6 to -10
ºC, which again are comparable with pollen sites from the region that range from
[Nagymohos site #48] -10.5 ± 4.1 ºC to [Straldzha site #57] -4.3 ± 5.8 ºC.
Further south and west, García-Amorena et al. (2007) reported mean annual temperature
anomalies of -2.0 to -11.3 ºC at LGM sites along the Portuguese coast, based on an indicator
species method using plant macrofossils. This is similar to the closest marine pollen sites off
the coast, which recorded values of [MD95-2039 site #1] -10.5 ± 4.6 ºC and [MD99-2331 site
#3] -5.3± 2.9 ºC. Meanwhile, in the far east of the study area, Zaarur et al. (2016) estimated a
mean annual temperature anomaly of around -3 ºC based on clumped isotope analysis of
Melanopsis shells from LGM sediments in the Sea of Galilee. This limited cooling appears
similar to the nearest pollen site [Lake Zeribar site #63] where we reconstruct a cooling of -
2.2 ± 4.6 ºC.

Reconstructions of LGM sea surface temperatures (SST's) provide yet another source of
comparison with our terrestrial pollen-based reconstructions, although many of the physical
processes controlling surface sea temperatures such as upwelling, surface mixing, surface
currents, stratification and thermal inertia through the seasonal cycle, represent quite different
processes to those controlling surface temperatures over land, particularly at the sub-regional
scale. Nevertheless, the Atlantic coastal waters of Iberia and the waters throughout the
Mediterranean Sea include many SST sites that lie in relative proximity to our terrestrial
pollen-sites, allowing us to make a comparison at the largest scale. Within this area the
MARGO database (MARGO Members, 2009) includes 13 Alkenone, 2 Mg/Ca and 41
Foraminifera based SST records of mean annual temperature, with the Foraminifera records
also providing an additional 41 winter (JFM) and summer (JAS) SST estimates. We compare
the SST records with the 36 closest terrestrial pollen records which fall within a box of -11 to
35 degrees longitude and 32 to 43 degrees latitude containing all of the SST records. A
simple site average indicates a mean annual SST anomaly of -5.5 ± 1.0 ºC which is relatively
close to the value of -7.2 ± 3.4 ºC obtained from the terrestrial pollen sites [sites #1-4, 5, 7-
24, 25, 26, 30, 35-38, 41, 47, 51, 53, 56-59]. Interestingly the inter-site variance (standard
deviation of the reconstructed temperatures across all sites) is almost identical for the two
datasets, 2.57 ºC for the SST sites and 2.63 ºC for the pollen sites, despite representing very
different environments, proxies and uncertainties. However, when we look at the seasonal
temperature anomalies, we find very different results. Site averaged winter SST anomalies
are -3.7 ± 1.1 ºC compared to -9.3 ± 4.2 ºC for winter temperatures from terrestrial pollen
sites, while in summer the values are reversed, -7.0 ± 0.8 ºC compared to -5.38 ± 3.3 ºC
respectively. This suggests that SST's experienced greater cooling in summer compared to
winter, which is the opposite to that generally found in terrestrial seasonal temperature
reconstructions throughout the region, although this is consistent with model simulations
(Mikolajewicz, 2011).
**4.2.2.2 Precipitation**
Few proxies apart from pollen provide quantitative reconstructions of precipitation during the
LGM. Glacier mass balance modelling includes assumptions about precipitation in order to
derive temperatures (Allen et al., 2008a), but neither is independent of the other. Hughes et
al. (2006) estimate from glacier modelling that mean annual precipitation during the LGM at
sites in the Pindus mountains in Greece was around 2300 ± 200mm, which they consider to
be similar to the present day (>2000mm). A small change in precipitation compared to
modern values is also indicated by the nearest pollen site, which is around 47 km to the south
[Ioannina #51], and indicates a mean annual precipitation anomaly of -152 ± 294mm,
representing just 15% of the modern value. A larger reduction in mean annual precipitation of
-45% (maximum) is reconstructed by García-Amorena et al. (2007) based on plant
macrofossil remains from sites on the Portuguese coast. In comparison, the closest pollen
sites record values which are a little lower, ranging from [MD95-2039 site #1] -22% to
[MD99-2331 site #3] -34%. Further north in south-west Germany, Prud'homme et al. (2018)
reconstructed mean annual precipitation from the delta 13C of earthworm calcite granules at
Fussloch. They estimate a field site average of 333 (159-574) mm/yr at the LGM, which
represents an anomaly of -503 mm/yr (-60%) relative to the modern precipitation of 836
mm/yr. This is comparable with the closest pollen site [Bergsee #29] with an anomaly of -
540 mm/yr.
As with glaciers, lake levels reflect changes in moisture balance that includes the effects of
both temperature (via evapotranspiration) and precipitation, rather than just precipitation.
They also represent semi-quantitative data at best, with changes often described relative to
the modern or other baseline. There are few lake level records available north of the Alps, but
to the south, many records indicate high lake levels in areas such as Spain (Lacey et al., 2016;
Moreno et al., 2012; Vegas et al., 2010), Italy (Belis et al., 1999; Giraudi, 2017), Greece and
Turkey (Harrison et al., 1996; Reimer et al., 2009) and the Middle East (Kolodny et al., 2005;
Lev et al., 2019). These lake records are also supported by evidence of higher river levels in
Morocco (El Amrani et al., 2008). The cause of the higher lake levels has been the subject of
some debate, since many pollen records (and especially early biome reconstructions) show
steppe vegetation that would suggest aridity that appears incompatible with higher lake
levels. Prentice et al. (1992) proposed that the co-existence of steppe vegetation and high lake
levels could be possible if precipitation increased outside of the summer growing season,
while summers themselves were drier and cooler with decreased evaporation. However, the
results of our analysis tend to indicate the opposite in regions with higher lake levels, with
increased summer rainfall and decreased winter rainfall. In addition, the increase in summer
precipitation was enough to compensate for the decrease in winter rainfall, leading to an
overall increase in mean annual precipitation at many pollen sites in Spain and Greece for
instance. This together with depressed temperatures and consequently decreased evaporation
could explain the higher lake levels, whilst also limiting the growth of trees as a result of
cooler temperatures and prolonged aridity outside of the summer season. Davis & Stevenson
(2007) also note a differential hydrological response between summer and winter rainfall in
the Mediterranean during the Holocene that may also provide an explanation. In this case
sporadic summer storms may result in high rates of runoff that may fill run-off fed lakes, but
low rates of soil moisture recharge that fails to benefit vegetation in the same way winter
rainfall does.
Overall, we reconstruct only a small reduction in precipitation during the LGM of around
91mm (13%) averaged over all sites, which is less than the ~200mm reduction based on the
sites in the pollen-climate compilation used by PMIP (Bartlein et al., 2011). Since our
precipitation reconstruction on average matches that of the INV reconstruction by Wu et al
(2007), we can attribute much of the difference to the greater aridity shown in the ANN
reconstruction by Peyron et al and Jost et al (2005) (see figure 9). As with temperature, this is
probably a reflection of the modern training set used in the ANN reconstruction which is
much smaller than our training set and is largely taken from the arid steppes of Kazakhstan
and Mongolia. However, it is also important to recognize the significant spatial variability in
precipitation, which means that a simple average of different sets of sites from different
regions may not accurately reflect the change in LGM precipitation at the European scale.
Nevertheless, one of the most consistent signals in our dataset is for an increase in summer
precipitation over many areas of Southern Europe and the Mediterranean. This is also found
in climate models, where it has been attributed to an increase in convection-driven
precipitation, although the amount of precipitation generated by this mechanism varies
significantly between models (Beghin et al., 2016). It may seem counter-intuative to see an
increase in reconstructed precipitation in the same regions where we also find a
preponderance of steppe or xerophytic biomes and taxa, including *Artemisia* and
Chenopodiaceae. This is attributable to the fact that climate can change quite markedly with
necessarily invoking a major change in vegetation, and especially the pollen biome. For
instance, a semi-arid climate ranges from 250-500mm rainfall a year, so we could expect a
semi-arid vegetation to be dominant even if the rainfall increases 250mm (100%).
A more consistent response in models is for an increase in winter precipitation across
Southern Europe and the Mediterranean related to a stronger and more southerly displaced jet

stream, with winter precipitation also accounting for much of the change in mean annual precipitation (Beghin et al., 2016). Our reconstruction of winter precipitation however shows less support for this scenario with a more general decrease in winter precipitation apart from southern and eastern Iberia, and with summer precipitation generally more important in those sites that show an overall increase in mean annual precipitation. This may not necessarily contradict the models in terms of the strength and position of the winter jet stream, but may instead indicate that models over-estimate the amount of moisture being carried westward from the cold North Atlantic along the storm track, especially across the far northern Mediterranean. The increase in winter precipitation across southern and eastern Iberia is however entirely consistent with a strengthened and more southerly jet stream, which also brings increased winter precipitation to the region today as a result of blocking over northern Europe/Atlantic and a negative NAO (Vicente-Serrano et al., 2011).

Other areas that show an increase in winter precipitation include pollen sites around the eastern end of the Alps. This is consistent with a recent study by Spötl et al (2021) who argued, on the basis of cryogenic carbonates preserved in a cave in Austria, that heavy winter (and autumn) precipitation was a significant factor in driving LGM glaciation in the region. The seasonally specific nature of this precipitation is also supported by the same pollen sites, which do not show any increase in summer precipitation at this time.

**5.0 Conclusions**

We have reconstructed the climate and vegetation cover across Europe, North Africa and the Middle East at the time of the LGM based on 63 pollen records. These records were selected using strict quality control criteria, with particular attention paid to dating control, which led to the exclusion of many records that have been used in previous studies. This fully documented dataset represents the most chronologically precise and spatially resolved view of LGM climate and vegetation during the PMIP benchmarking time window at $21 \pm 2$ ka. Nevertheless, it is important to recognize that there are still significant spatial gaps in pollen sites especially north of the Alps, the Balkans, Turkey and the Middle East, and we continue to have only a partial understanding of the LGM over these areas.

One of the key questions concerning the vegetation landscape of the LGM in Europe has been the extent to which forest rather than steppe covered the continent, and to what extent temperate elements could be found north of the classical refugia areas of Southern Europe and the Mediterranean. Our results show that although steppe and tundra was extensive at the time of the LGM, areas of open forest also occurred in many regions, particularly (but not exclusively) in Iberia, northern Italy and Central Europe. These forest or woodland stands are likely to have been located in environmentally favourable areas, with good soils, elevated rainfall and shelter from cold, desiccating winds. In those areas where woodland existed, Boreal taxa generally dominated north and east of the Alps, while temperate and thermophilious (mainly drought adapted) taxa were generally confined to areas south of the Alps and around the Mediterranean. The temperate deciduous forests that compose the climax community in many areas of Europe today were displaced to the south and reduced to a partnership role with Boreal elements. Overall our new reconstruction indicates greater agreement with model land cover simulations, but models still appear to over-estimate the amount of forest and woodland over areas such as France and the Benelux, Greece, Turkey and the Far East.

Another key question about the LGM concerns the ability of climate models to simulate the
climate of this period and whether pollen-based climate reconstructions which show
disagreement with models have been biased by the effects of low CO2 on plant physiology.
We find that our new pollen-climate reconstruction shows much closer agreement with
climate models than previous reconstructions that did not take account of low CO2 effects.
We also find close agreement with previous reconstructions that did take account of CO2
effects. Since our MAT method itself does not specifically take account of low CO2 effects,
this would suggest that this problem is not a significant hindrance to MAT performance at the
time of the LGM, at least not compared to other uncertainties. Instead, we suggest that the
main factor in the performance of pollen-climate transfer functions that use modern analogue
methods is the provision of a large enough modern pollen dataset with suitable LGM
analogues.
This conclusion is supported by comparison with climate reconstructions based on other
proxies. We found little difference between our MAT reconstruction and a Chironomid-based
summer temperature record based on a downcore sample by sample comparison, as well as
comparsons with records from a variety of other proxies at a regional scale. However, it is
notable that some studies using glacier mass balance modelling methods indicate LGM
temperatures that are much cooler than our pollen-based reconstruction. The reasons behind
this are not clear, but our pollen-based results indicate higher than present precipitation in
some areas that could potentially explain low elevation glacier ELA's without the need for
such cold temperatures.
We also find that although our pollen-based reconstruction and those of SST's generally
agree in terms of mean annual temperatures, SST's indicate greater cooling in summer
compared to winter, while terrestrial records indicate greater cooling in winter compared to
summer. These seasonal differences are also reproduced in climate models, and probably
reflect the different processes driving seasonal temperature change in the terrestrial and
marine domain.
Our reconstructions of precipitation show large spatial and seasonal variability, but generally
indicate less overall aridity than previously suggested from smaller scale studies which
sampled less of the spatial domain. We find that in some regions of Southern Europe
precipitation may actually have been greater than present, especially in summer, but also in
winter in southern and eastern Iberia and around the southern slopes of the Alps. This may
have important implications in understanding the development of LGM glaciation, which
may be less a function of temperature than previously supposed. This could also help better
explain the observed asynchronous nature of glaciation even within relatively small regions
such as Europe, as a result of more localized controls on ice sheet development such as
precipitation.
We hope that this new continental-scale dataset of climate and vegetation reconstructions will
provide an improved baseline for data-model comparisons and other studies that will allow us
to better understand the complex LGM environment.
**Code/Data availability**

All of the data shown in the figures together with the fossil and modern pollen datasets will
be made available on pangaea.de once the review process has been completed and these
datasets are therefore no longer subject to change.

**Author contribution**


BASD designed the study, undertook the analysis and wrote the manuscript. MF and ER
designed and prepared the maps. JOK and AB reviewed the manuscript and provided
additional input.

**Competing interests**


The authors declare that they have no conflict of interest.

**Acknowledgements**


This work was supported by a grant from the Fonds de Recherche du Québéc Société et
Culture (2019-SE3-254686) to AB. Data were obtained from the European Pollen Database
(EPD), based within the Neotoma Paleoecology Database (http://www.neotomadb.org). The
work of data contributors, data stewards, and the Neotoma and EPD community is gratefully
acknowledged. We dedicate this paper in memory of Eric Grimm, whose tireless work for the
EPD and Neotoma helped make this study possible.

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

## Tables

| Site | Site Name | Country/Ocean | Latitude | Longitude | Elevation | Site Type | Data Type | Samples | Source | Reference |
|---|---|---|---|---|---|---|---|---|---|---|
| 1 | MD95-2039 (M) | Atlantic | 40.578333 | -10.348333 | -3381 | Marine | Raw Count | 21 | EPD (E#1472) | Roucoux et al. 2005 |
| 2 | SU81-18 (M) | Atlantic | 37.77 | -9.82 | -3135 | Marine | Raw Count | 10 | ACER | Turon et al. 2003 |
| 3 | MD99-2331 (M) | Atlantic | 41.15 | -9.68 | -2110 | Marine | Raw Count | 41 | ACER | Naughton et al. 2006 |
| 4 | Carn Morval | United Kingdom | 49.926111 | -6.313889 | 5 | Lake | Digitised | 1 | Publication | Scourse 1991 |
| 5 | Gorham Cave | Spain | 36.132826 | -5.347358 | 0 | Cave | Publication | 1 | Publication | Carrion et al. 2008 |
| 6 | Dozmary Pool | United Kingdom | 50.5347222 | -4.5358333 | 265 | Lake | Raw Count | 32 | Author | Kelly et al. 2010 |
| 7 | Bajondillo | Spain | 36.619722 | -4.496389 | 20 | Cave | Raw Count | 1 | EPD (E#1570) | Cortes-Sanchez et al 2011 |
| 8 | Laguna del maar de Fuentillejo | Spain | 38.937996 | -4.0539 | 637 | Lake | Digitised | 1 | Publication | Ruiz-Zapata et al. 2009 |
| 9 | Padul-1 | Spain | 37.016338 | -3.608503 | 785 | Peat Bog | Digitised | 13 | Publication | Pons & Reille 1988 |
| 10 | Padul-2 | Spain | 37.010833 | -3.603889 | 726 | Peat Bog | Digitised | 1 | Publication | Camuera et al. 2019 |
| 11 | Cova di Carihuela | Spain | 37.4489 | -3.4297 | 1020 | Cave | Digitised | 1 | Publication | Carrion 1992 |
| 12 | Ifri El Baroud | Morocco | 34.75 | -3.3 | 539 | Cave | Digitised | 1 | Publication | Poti et al. 2019 |
| 13 | MD95-2043 (M) | Mediterranean | 36.14 | -2.621 | -1841 | Marine | Raw Count | 7 | ACER | Fletcher et al. 2008 |
| 14 | San Rafael | Spain | 36.773611 | -2.601389 | 0 | Peat Bog | Raw Count | 2 | EPD (E#574) | Pantaléon-Cano 1997 |
| 15 | Siles | Spain | 38.24 | -2.3 | 1320 | Lake | Digitised | 1 | Publication | Carrion 2002 |
| 16 | Torrecilla de Valmadrid | Spain | 41.4469444 | -0.895 | 570 | Colluvium | Digitised | 1 | Publication | Valero-Garces et al. 2004 |
| 17 | Navarrés-1 | Spain | 39.1 | -0.683333 | 225 | Peat Bog | Raw Count | 1 | EPD (E#469) | Carrión & Dupré-Olivier 1996 |
| 18 | Navarrés-2 | Spain | 39.1 | -0.683333 | 225 | Peat Bog | Raw Count | 1 | EPD (E#470) | Carrión & Dupré-Olivier 1996 |
| 19 | Tourbiere de l'Estarres | France | 43.0933 | -0.3792 | 356 | Lake | Digitised | 1 | Publication | Jalut et al. 1988 |
| 20 | Cova de les Malladetes | Spain | 39.058 | -0.321 | 20 | Cave | Digitised | 1 | Publication | Dupré Ollivier 1988 |
| 21 | Lourdes | France | 43.033333 | -0.075 | 430 | Lake | Digitised | 15 | Publication | Reille & Andrieu 1995 |
| 22 | Lake Estanya | Spain | 42.0333333 | 0.53333333 | 670 | Lake | Digitised | 1 | Publication | Vegas-Villarubia et al. 2013 |
| 23 | Freychinede | France | 42.7833 | 1.4333 | 1350 | Lake | Digitised | 1 | Publication | Jalut et al. 1992 |
| 24 | Banyoles | Spain | 42.133333 | 2.75 | 173 | Lake | Raw Count | 13 | EPD (E#931) | Pérez-Obiol & Julia 1994 |
| 25 | Lac du Bouchet B5 | France | 44.916667 | 3.783333 | 1200 | Lake | Digitised | 14 | Publication | Reille & de Beaulieu 1988 |
| 26 | MD99-2348 (103) (M) | Mediterranean | 42.692778 | 3.841667 | -296 | Marine | Raw Count | 41 | EPD (E#1474) | Beaudouin et al. 2007 |
| 27 | Les Echets G | France | 45.9 | 4.93 | 267 | Peat Bog | Digitised | 136 | ACER | de Beaulieu & Reille 1984 |
| 28 | La Grotte Walou | Belgium | 50.585278 | 5.536389 | 252 | Cave | Digitised | 1 | Publication | Damblon 2011 |
| 29 | Bergsee | Germany | 47.5722222 | 7.93638889 | 382 | Lake | Digitised | 1 | Publication | Duprat-Oualid et al. 2017 |
| 30 | Garaat El-Ouez | Algeria | 36.818333 | 8.33333 | 45 | Peat Bog | Raw Count | 6 | EPD (E#1501) | Benslama et al 2010 |
| 31 | Pian del Lago | Italy | 44.321561 | 9.485682 | 833 | Lake | Digitised | 1 | Publication | Guido et al. 2020 |
| 32 | Pilsensee | Germany | 48.0267 | 11.1883 | 534 | Lake | Digitised | 1 | Publication | Küster 1995 |
| 33 | Orgiano | Italy | 45.29 | 11.43 | 19 | Peat Bog | Digitised | 1 | Publication | Paganelli 1996 |
| 34 | Lago della Costa | Italy | 45.2702778 | 11.7430556 | 7 | Lake | Digitised | 8 | Publication | Kaltenrieder et al. 2009 |
| 35 | Lagaccione | Italy | 42.566667 | 11.85 | 355 | Lake | Raw Count | 7 | ACER | Magri 1999 |
| 36 | Lago Vico | Italy | 42.3166667 | 12.1666667 | 510 | Lake | Digitised | 15 | Publication | Magri & Sadori 1999 |
| 37 | Stracciacappa | Italy | 42.13 | 12.32 | 220 | Lake | Raw Count | 2 | ACER | Giardini 2007 |
| 38 | Lago di Monterosi | Italy | 42.2166667 | 12.4333333 | 237 | Lake | Raw Count | 1 | Publication | Bonatti 1970 |
| 39 | Venice | Italy | 45.629523 | 12.654086 | 0 | Peat Bog | Digitised | 1 | Publication | Miola et al. 2006 |
| 40 | Azzano Decimo | Italy | 45.8833 | 12.7165 | 10 | Alluvial Fan | Raw Count | 6 | ACER | Pini et al. 2009 |
| 41 | Valle di Castiglione | Italy | 41.89 | 12.75 | 44 | Lake | Raw Count | 2 | ACER | Follieri et al. 1989 |
| 42 | Travesio | Italy | 46.2 | 12.87 | 220 | Lake | Digitised | 1 | Publication | Monegato et al. 2007 |
| 43 | Orvenco | Italy | 46.252088 | 13.169771 | 380 | Alluvial Fan | Digitised | 1 | Publication | Monegato et al. 2007 |
| 44 | Rio Doidis | Italy | 46.12 | 13.19 | 152 | Lake | Digitised | 1 | Publication | Monegato et al. 2007 |
| 45 | Billerio | Italy | 46.22 | 13.21 | 300 | Lake | Digitised | 1 | Publication | Monegato et al. 2007 |
| 46 | Kersdorf-Briesen | Germany | 52.333704 | 14.269142 | 44 | Lake | Digitised | 1 | Publication | Strahl 2005 |
| 47 | Lago Grande di Monticchio | Italy | 40.944444 | 15.6 | 1326 | Lake | Raw Count | 6 | EPD (E#932) | Watts et al. 1996 |
| 48 | Nagymohos | Hungary | 48.326944 | 20.436389 | 297 | Peat Bog | Raw Count | 14 | Publication | Magyari et al 1999 |
| 49 | Safarka | Slovakia | 48.8819444 | 20.575 | 600 | Peat Bog | Digitised | 1 | Publication | Jankovska 2008 |
| 50 | Feher Lake | Hungary | 46.45 | 20.65 | 86 | Lake | Raw Count | 10 | Publication | Magyari et al. 2014 |
| 51 | Ioannina | Greece | 39.75 | 20.85 | 470 | Peat Bog | Raw Count | 20 | ACER | Tzedakis et al. 2004 |
| 52 | Kokad | Hungary | 47.4027778 | 21.9286111 | 112 | Peat Bog | Raw Count | 2 | Publication | Magyari et al. 2019 |
| 53 | Lake Xinias | Greece | 39.05 | 22.27 | 500 | Lake | Raw Count | 5 | EPD (E#976) | Bottema 1979 |
| 54 | Mickunai | Lithuania | 54.722114 | 25.532218 | 143 | Lake | Digitised | 1 | Publication | Satkunas & Grigiene 2012 |
| 55 | Lake Sfanta Anna | Romania | 46.1263889 | 25.8880556 | 946 | Lake | Digitised | 1 | Publication | Magyari et al. 2014 |
| 56 | Megali Limni | Greece | 39.1 | 26.3 | 323 | Lake | Digitised | 1 | Publication | Margari et al. 2009 |
| 57 | Straldzha | Bulgaria | 42.630278 | 26.77 | 138 | Peat Bog | Raw Count | 3 | Publication | Connor et l. 2013 |
| 58 | MD01-2430 (M) | Turkey | 40.796833 | 27.725166 | -580 | Marine | Digitised | 1 | Publication | Valsecchi et al. 2012 |
| 59 | Lake Iznik | Turkey | 40.433889 | 29.533056 | 88 | Lake | Raw Count | 7 | EPD (E#714) | Miebach et al 2016 |
| 60 | M72/5 628-1 (M) | Black Sea | 42.1035 | 36.62383 | -418 | Marine | Raw Count | 6 | Pangaea (833387) | Shumilovskikh et al. 2014 |
| 61 | Dziguta | Georgia | 42.99 | 41.07 | 35 | Peat Bog | Digitised | 1 | Publication | Arslanov et al. 2007 |
| 62 | Lake Van LG | Turkey | 38.667 | 42.669 | 1649 | Lake | Raw Count | 10 | Pangaea (853779) | Pickarski et al. 2015 |
| 63 | Lake Zeribar | Iran | 35.533333 | 46.116667 | 1286 | Lake | Raw Count | 17 | EPD (E#714) | van Zeist & Bottema 1977 |

Table 1. List of selected sites


| | RMSE | R2 |
|---|---|---|
| TANN | 2.28 | 0.9 |
| TDJF | 3.35 | 0.91 |
| TJJA | 2.21 | 0.81 |
| PANN | 224.94 | 0.69 |
| PDJF | 78.51 | 0.69 |
| PJJA | 52.49 | 0.75 |
| Tree Cover | 21.03 | 0.52 |

Table 2. MAT performance statistics based on the modern pollen sample training set. This
includes Mean Annual Temperature and Precipitation (TANN and PANN), Mean Winter
Temperature and Precipitation (TDJF and PDJF) and Mean Summer Temperature and
Precipitation (TJJA and PJJA).

**Figures**


Figure 1. Site locations and archives (Site numbers are as shown in Table 1)


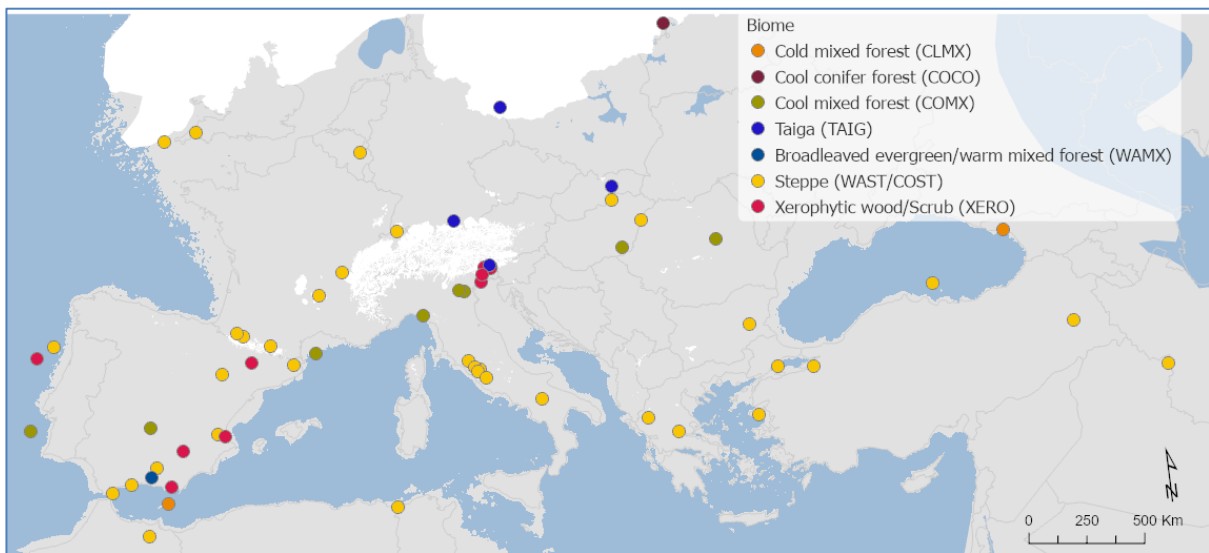

Figure 2. Pollen biomes

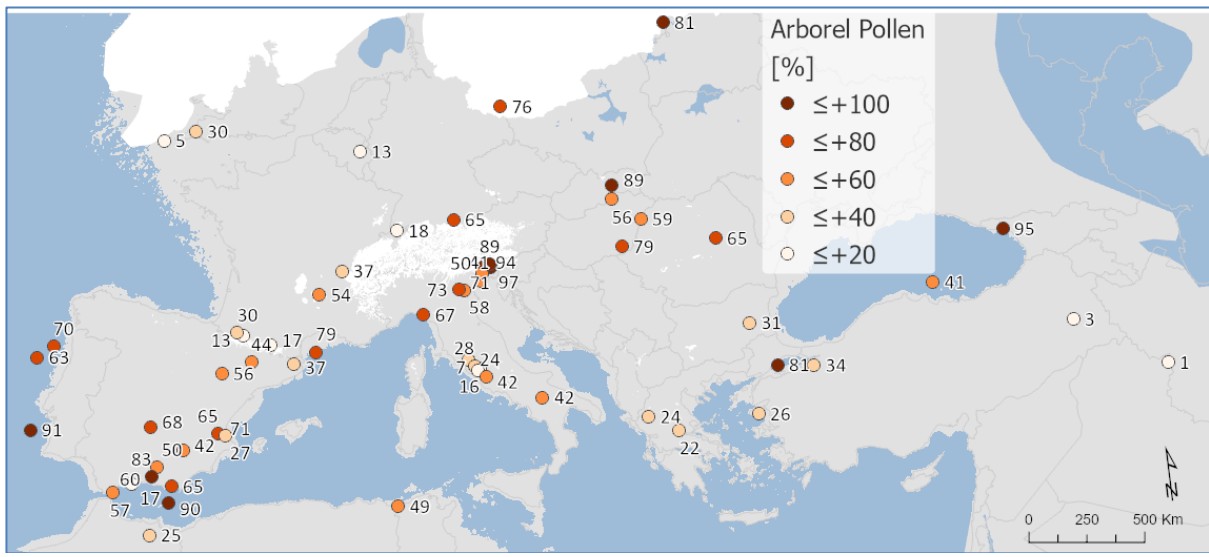

Figure 3. Arboreal Pollen (AP) % forest cover

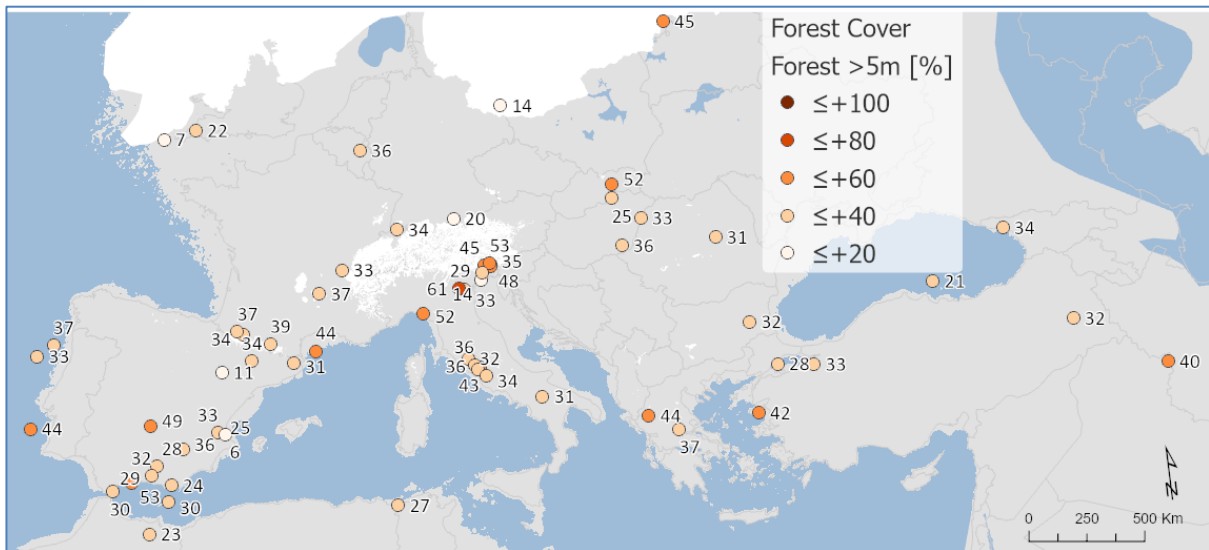

Figure 4. Modern Analogue Technique (MAT) % forest cover


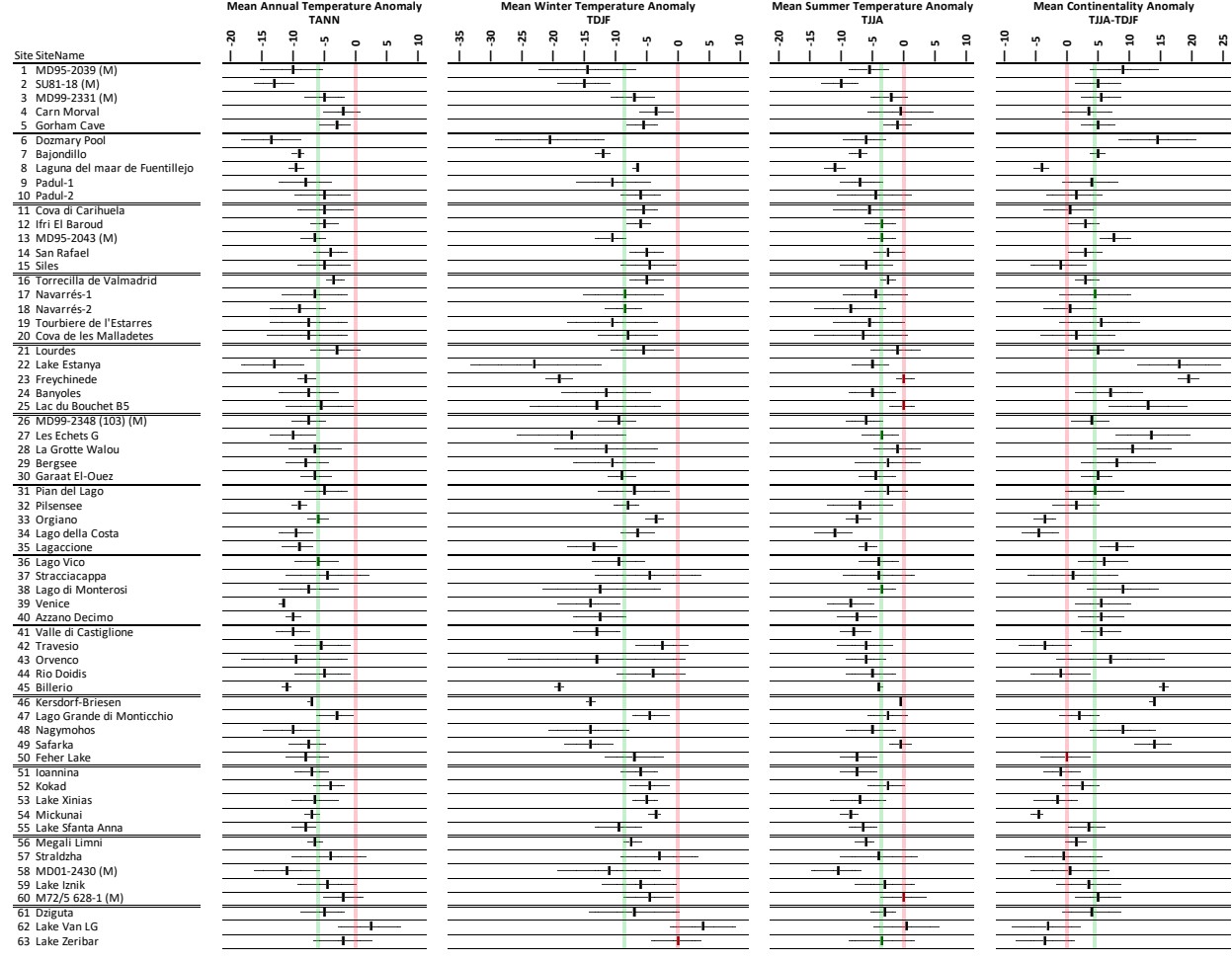

Figure 5. Pollen-based MAT reconstructions for LGM annual, winter and summer
temperature anomalies (uncertainties represent one standard deviation). Continentality
represents the difference in temperature between summer and winter, with positive anomalies
indicating an increase in the temperature difference between summer and winter. All values
are expressed as anomalies compared with the present day. The green line indicates the mean
for all the sites.

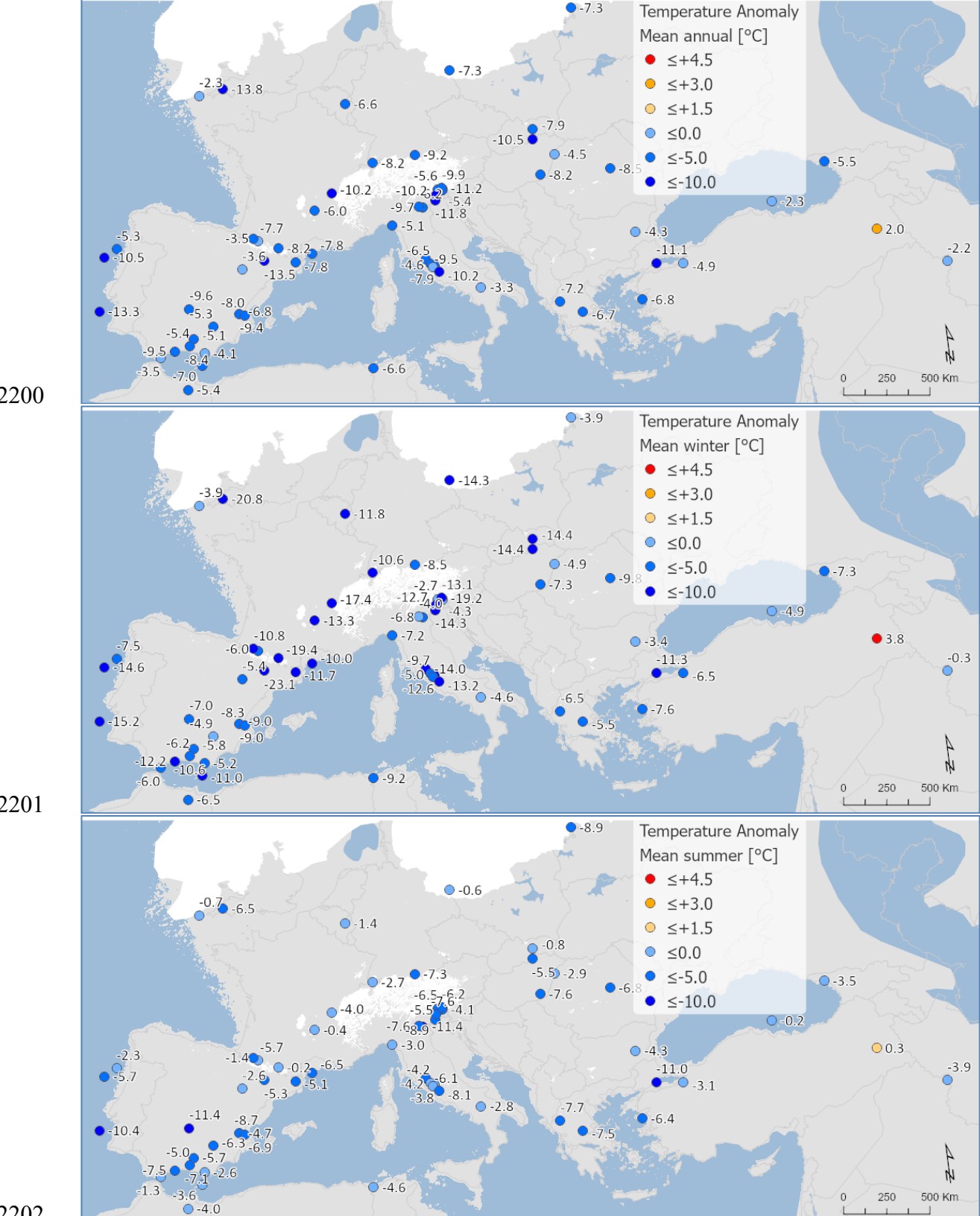




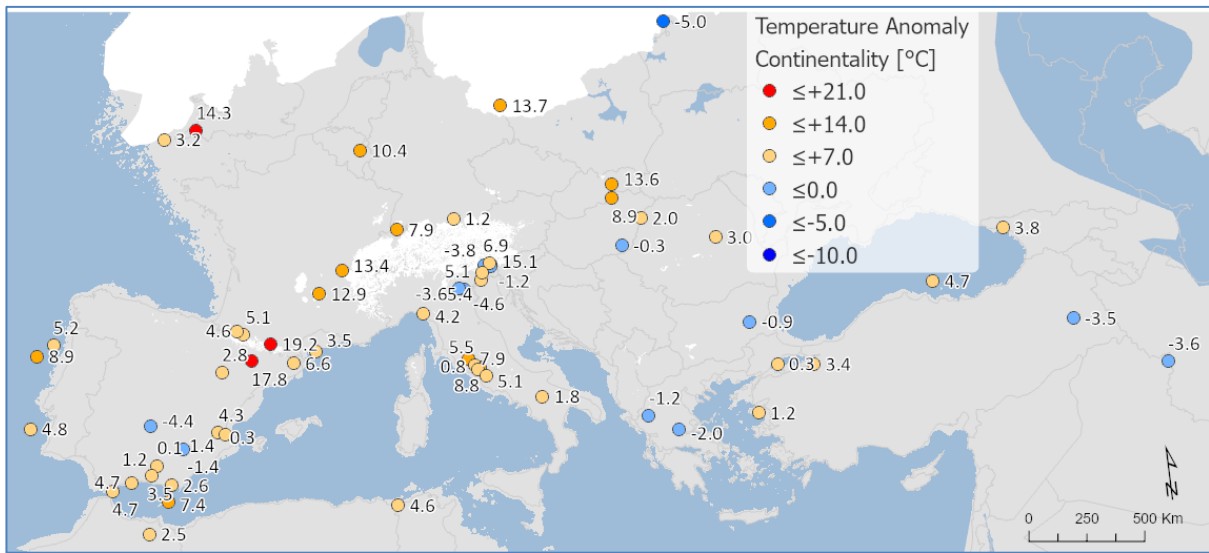

Figure 6. Maps of pollen-based MAT reconstructions for LGM annual, winter and summer
temperature anomalies (as shown in figure 9). Continentality represents the difference in
temperature between summer and winter, with positive anomalies indicating an increase in
the temperature difference between summer and winter. All values are expressed as
anomalies compared with the present day.


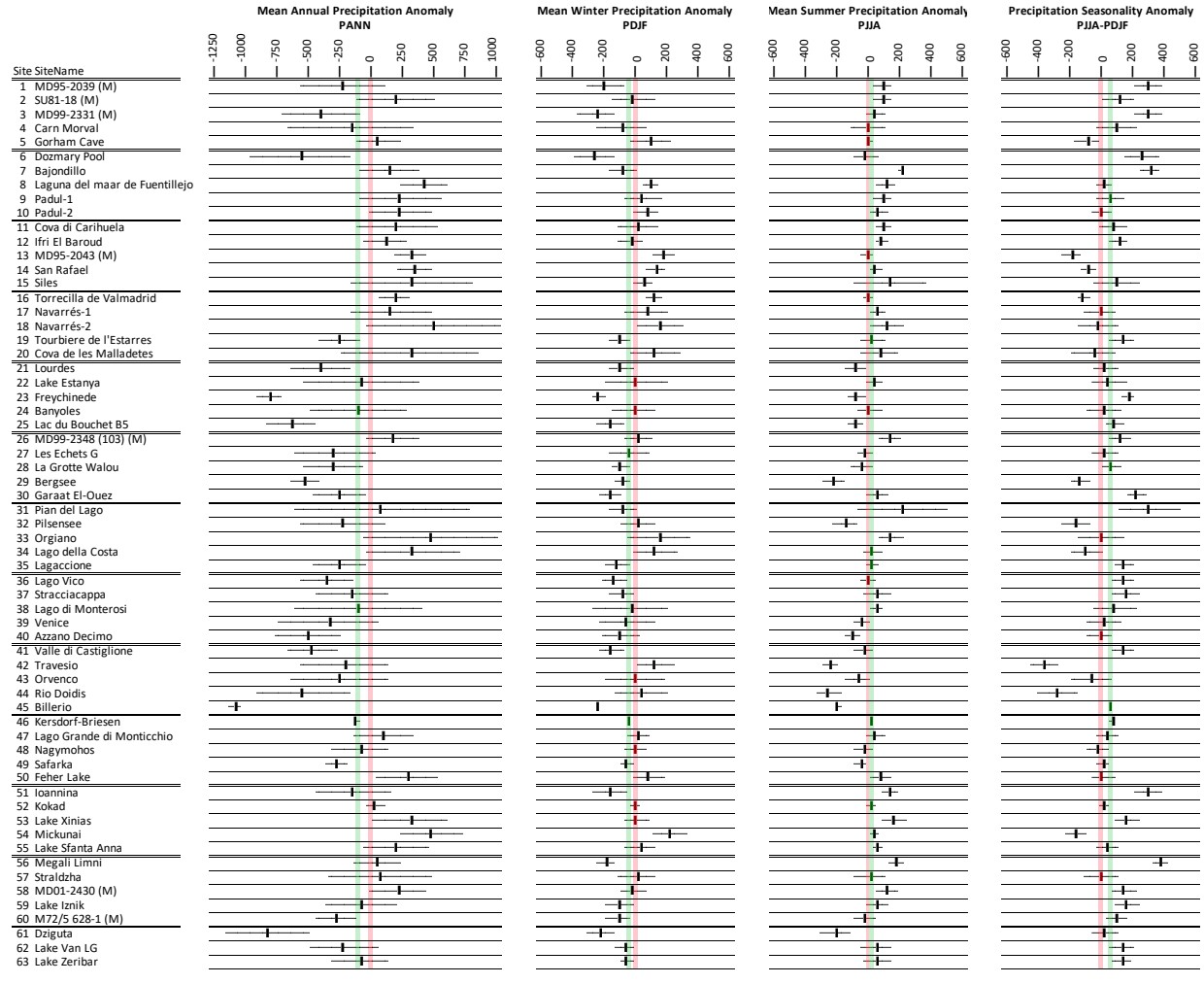



Figure 7. Pollen-based MAT reconstructions for LGM annual, winter and summer
precipitation anomalies (uncertainties represent one standard deviation). Seasonality
represents the difference in precipitation between summer and winter, with positive
anomalies indicating an increase in summer precipitation compared to winter. All values are
expressed as anomalies compared with the present day. The green line indicates the mean for
all the sites.

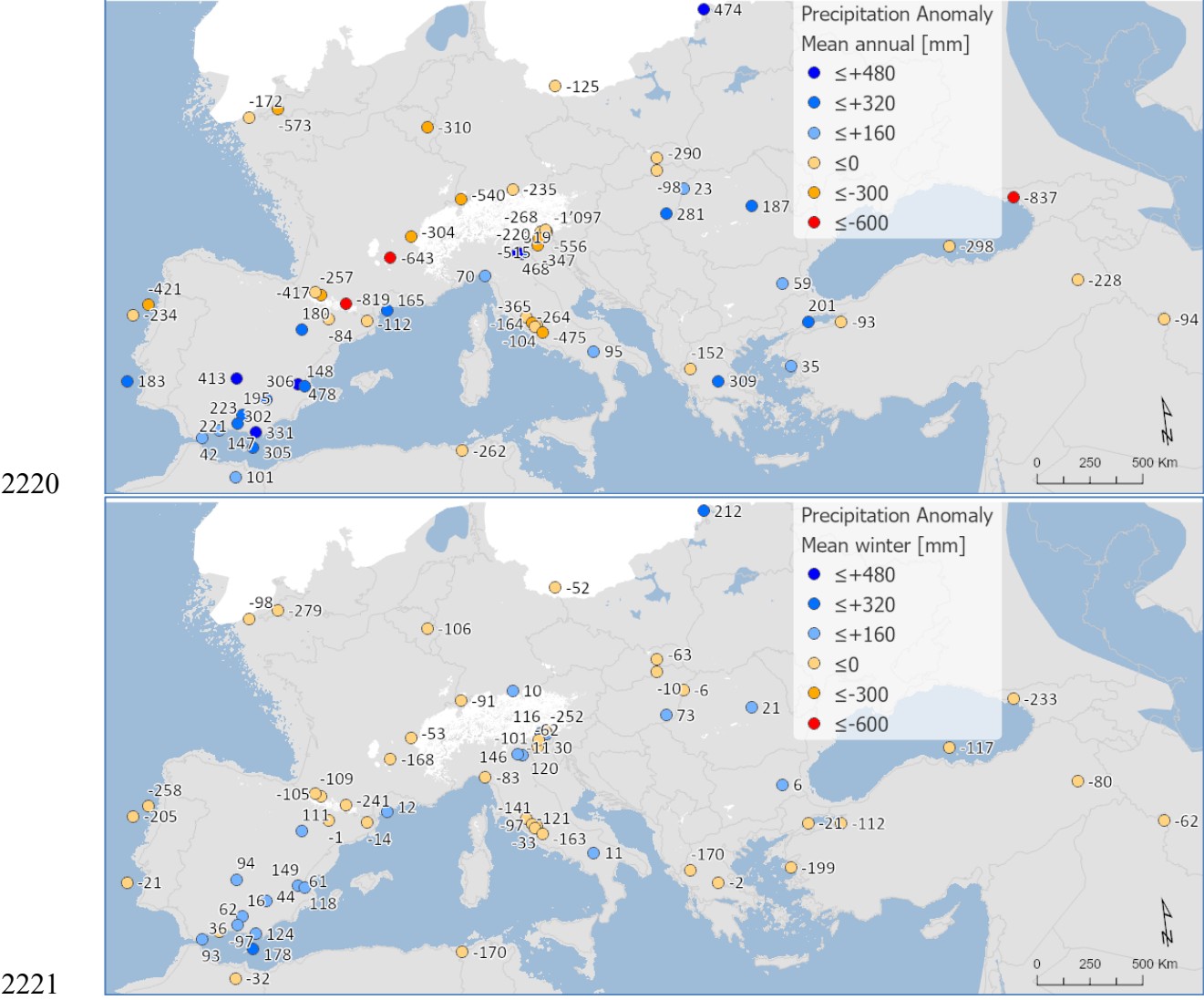



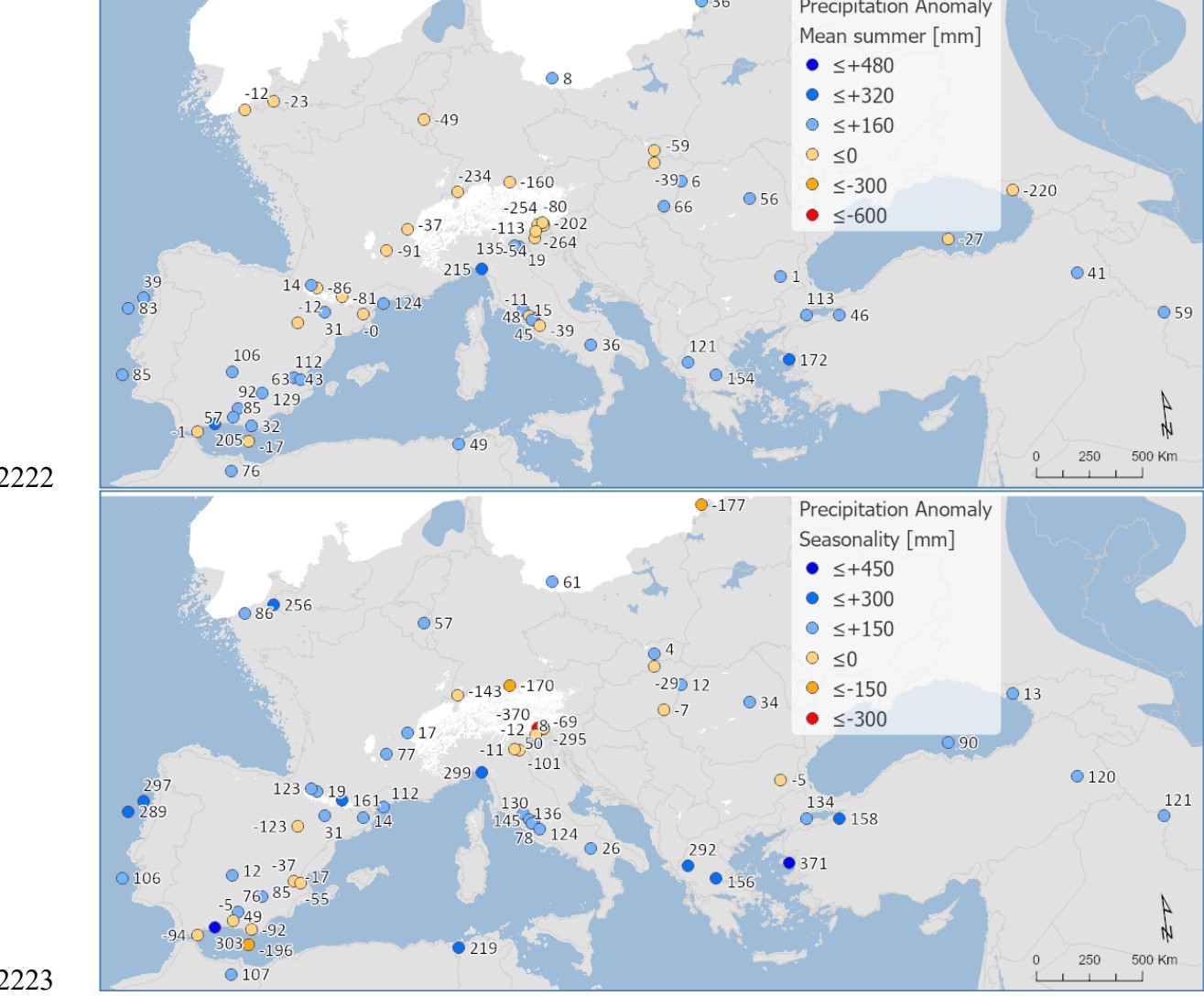

Figure 8. Maps of pollen-based MAT reconstructions for LGM annual, winter and summer precipitation anomalies (as shown in figure 11). Seasonality represents the difference in precipitation between summer and winter, with positive anomalies indicating an increase in summer precipitation compared to winter. All values are expressed as anomalies compared with the present day.


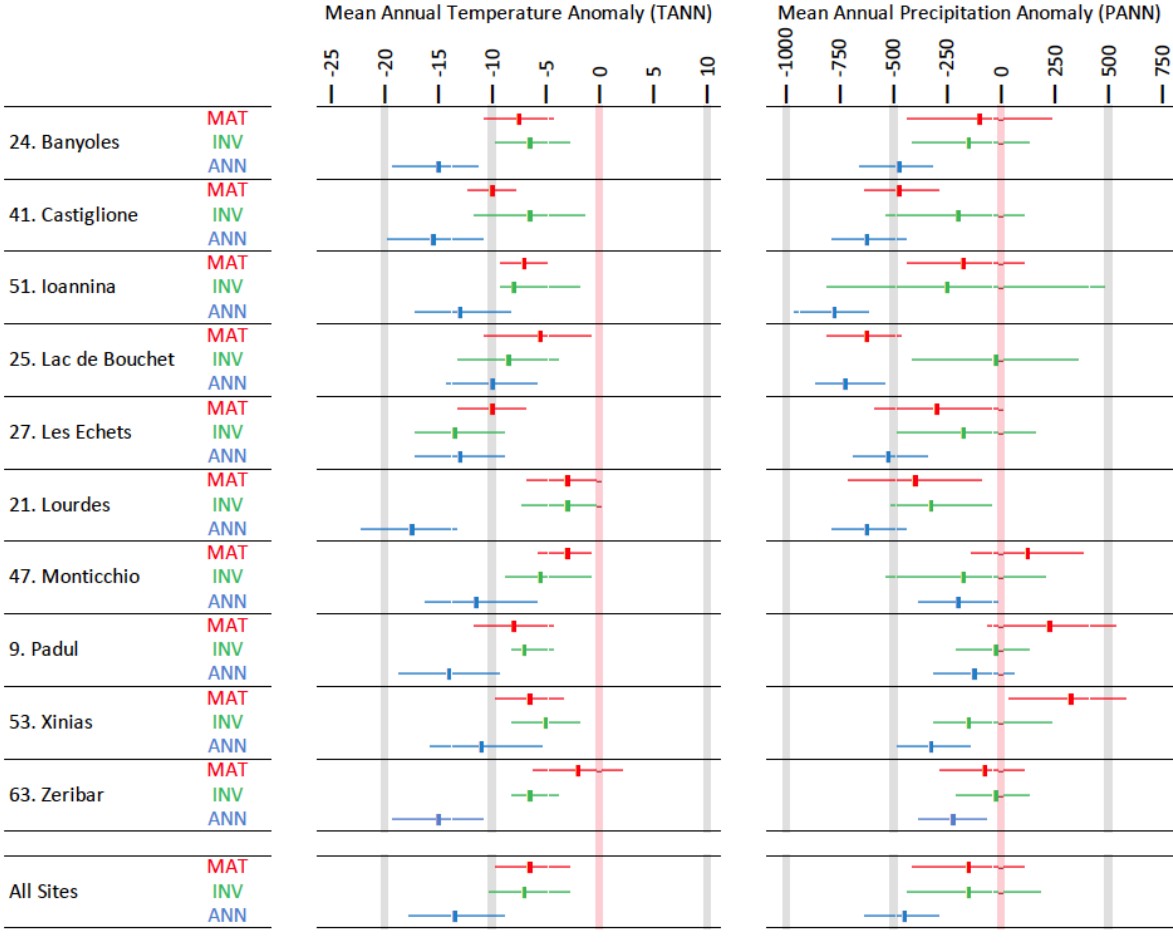

Figure 9. A site-by-site comparison between LGM pollen-climate reconstructions based on
Modern Analogue Technique MAT (this study), neural-networks ANN (Peyron et al., 1998),
and Inverse Modelling INV (Wu et al., 2007). The results show that MAT and INV give
similar climate reconstructions, but ANN is significantly cooler/drier.

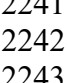


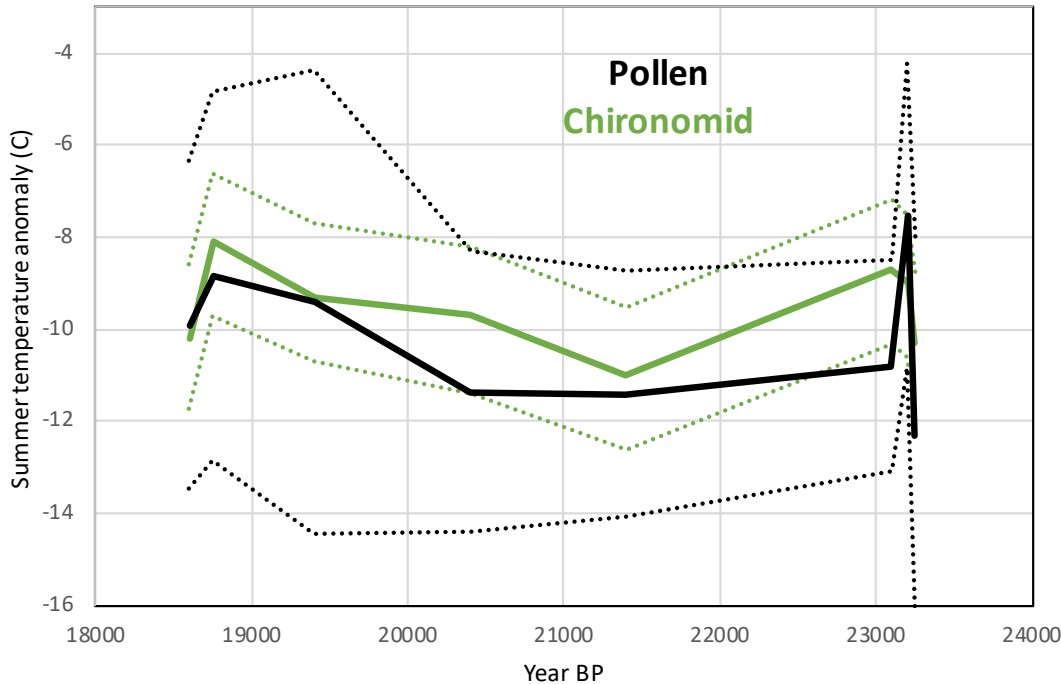

Figure 10. Comparison between LGM pollen-climate MAT and chironomid summer
temperature reconstructions at Lago della Costa, Italy (chironomid reconstruction and pollen
data from Samartin et al., 2016). Dash lines show uncertainties.

**Appendix**

| Site | Site Name | COHMAP Quality | < 17k 18k 19k 20k 21k 22k 23k 24k 25k > | Upper 14C | Upper Cal. BP | Lower 14C | Lower Cal. BP |
|---|---|---|---|---|---|---|---|
| 1 | MD95-2039 (M) | 3C | | 14830±80 | 18166±269 | 19950±210 | 23883±374 |
| 2 | SU81-18 (M) | 2C | | 17510±270 | 20952±404 | 21250±280 | 25420±441 |
| 3 | MD99-2331 (M) | 2C | | 16170±130 | 19325±303 | 19770±170 | 23682±336 |
| 4 | Carn Morval | 4C | | | 18600±3700 | 21500±890/800 | 25867±1127 |
| 5 | Gorham Cave | 4D | | | 18440±160 | | 22055±341 |
| 6 | Dozmary Pool | 2C | | 14568±129 | 17569±523 | 18325±216 | 21769±602 |
| 7 | Bajondillo | 1C | | | 18701±2154 | | |
| 8 | Laguna del maar de Fuentillejo | 5D | | 16540±90 | 19847±308 | | |
| 9 | Padul-1 | 3D | | 18300±300 | 21821±412 | 19100±160 | 22922±308 |
| 10 | Padul-2 | 1D | | | 17450±539 | | 21082±539 |
| 11 | Cova di Carihuela | 2C | | 15700±220 | 18958±280 | 21430±130 | 25659±226 |
| 12 | Ifri El Baroud | 2D | | 17296±87 | 20761±293 | | |
| 13 | MD95-2043 (M) | 2C | | 15440±90 | 18533±294 | 18260±120 | 21951±335 |
| 14 | San Rafael | 3D | | 9980±60 | 11464±133 | 16860±120 | 20083±292 |
| 15 | Siles | 2D | | 17030±80 | 20345±351 | | |
| 16 | Torrecilla de Valmadrid | 2D | | 17100±85 | 20456±366 | | |
| 17 | Navarrés-1 | 4D | | 18360±195 | 22001±353 | 20700±295 | 24664±411 |
| 18 | Navarrés-2 | 5D | | 5150±50 | 5881±85 | 16000± | 19144± |
| 19 | Tourbiere de l'Estarres | 1C | | 17150±250 | 20522±470 | 18970±160 | 22847±317 |
| 20 | Cova de les Malladetes | 5D | | 16300±1500 | 19686±1723 | | |
| 21 | Lourdes | 4D | | 18510±130 | 22112±130 | 20025±175 | 23952±355 |
| 22 | Lake Estanya | 5D | | | 9498±50 | | 19184±251 |
| 23 | Freychinede | 3C | | 14800±800 | 17912±856 | 21300±760 | 25615±1030 |
| 24 | Banyoles | 4C | | | 19878±100 | | 27862±3000 |
| 25 | Lac du Bouchet B5 | 2C | | 15350±350 | 18513±435 | 19200±300 | 23006±384 |
| 26 | MD99-2348 (103) (M) | 1D | | 17660±60 | 21065±310 | 19350±90 | 23111±271 |
| 27 | Les Echets G | 1C | | 17530±270 | 20970±407 | 18030±250 | 21704±473 |
| 28 | La Grotte Walou | 1D | | | | | 21200±700 |
| 29 | Bergsee | 2D | | | | 17780±90 | 21244±306 |
| 30 | Garaat El-Ouez | 2C | | 16010±320 | 19200±801 | | |
| 31 | Pian del Lago | 2D | | | | | 21260±320 |
| 32 | Pilsensee | 6D | | 15860±250 | 19073±290 | | |
| 33 | Orgiano | 2D | | 17760±160 | 21221±373 | 19290±520 | 23141±621 |
| 34 | Lago della Costa | 2C | | 15400±150 | 18484±330 | 19285±160 | 23052±302 |
| 35 | Lagaccione | 2C | | 16080±450 | 19369±527 | 20615±940 | 24746±1201 |
| 36 | Lago Vico | 3C | | 14385±140 | 17541±272 | 20500±230 | 24430±376 |
| 37 | Stracciacappa | 4C | | 12060±130 | 14093±281 | 19745±820 | 22675±955 |
| 38 | Lago di Monterosi | 2D | | 17040±350 | 20398±544 | | |
| 39 | Venice | 5D | | | | 18640±100 | 22277±336 |
| 40 | Azzano Decimo | 2D | | 18000±300 | 21637±529 | 21025±245 | 25179±449 |
| 41 | Valle di Castiglione | 3C | | 14220±145 | 17443±270 | 20300±700 | 24266±842 |
| 42 | Travesio | 5D | | | | 18780±200 | 22483±406 |
| 43 | Orvenco | 2D | | 17760±160 | 21221±373 | 19290±520 | 23141±621 |
| 44 | Rio Doidis | 5D | | | | 18860±190 | 22390±373 |
| 45 | Billerio | 3D | | | | 18165±200 | 21872±382 |
| 46 | Kersdorf-Briesen | 1D | | | | 17622±94 | 21183±356 |
| 47 | Lago Grande di Monticchio | 2C | | | 20204± | | 24014± |
| 48 | Nagymohos | 2C | | 14246±144 | 17361±425 | 18159±247 | 21735±622 |
| 49 | Safarka | 3D | | | | 18287±1512 | 21912±1781 |
| 50 | Feher Lake | 1D | | 17715±250 | 21190±463 | 19911±81 | 23841±313 |
| 51 | Ioannina | 3C | | 15330±140 | 18420±312 | 20760±230 | 24748±330 |
| 52 | Kokad | 5D | | 14326±63 | 17433±443 | 16280±90 | 19685±538 |
| 53 | Lake Xinias | 6C | | 11150±130 | 13049±160 | 21390±430 | 25671±648 |
| 54 | Mickunai | 1D | | | 21000±2200 | | |
| 55 | Lake Sfanta Anna | 1D | | 17626±96 | 20955±432 | | |
| 56 | Megali Limni | 6D | | 19072±237 | 22906±340 | | |
| 57 | Straldzha | 6C | | 14696±65 | 18022±364 | 23653±114 | 28580±390 |
| 58 | MD01-2430 (M) | 4C | | 12050±75 | 14904±324 | 18310±380 | 21746±968 |
| 59 | Lake Iznik | 7D | | 16910±100 | 19515±115 | | |
| 60 | M72/5 628-1 (M) | 2C | | 16835±85 | 18490± | 19495±90 | 21280± |
| 61 | Dziguta | 4C | | 12990±160 | 15839±483 | 20560±880 | 24666±1126 |
| 62 | Lake Van LG | 2C | | | 18590±62 | | 23290±596 |
| 63 | Lake Zeribar | 4C | | 13650±160 | 16610±399 | 22000±500 | 26462±880 |

COHMAP chronological quality classification:
1C: Bracketing dates within 2000 14C (2360 Cal.) yr interval about the time being assessed
2C: Bracketing dates, one within 2000 14C (2360 Cal.) yr and the second within 4000 14C (4682 Cal.) yr of the time being assessed
3C: Bracketing dates within 4000 14C (4682 Cal.) yr interval about the time being assessed
4C: Bracketing dates, one being within 4000 14C (4682 Cal.) yr and the second being within 6000 14C (7490 Cal.) yr of the time being assessed
5C: Bracketing dates within 6000 14C (7490 Cal.) yr interval about the time being assessed
6C: Bracketing dates, one within 6000 14C (7490 Cal.) yr and the second within 8000 14C (9681 Cal.) yr of the time being assessed
7C: Poorly dated
1D: Date within 250 14C (206 Cal.) yr of the time being assessed
2D: Date within 500 14C (684 Cal.) yr of the time being assessed
3D: Date within 750 14C (975 Cal.) yr of the time being assessed
4D: Date within 1000 14C (1123 Cal.) yr of the time being assessed
5D: Date within 1500 14C (1881 Cal.) yr of the time being assessed
6D: Date within 2000 14C (2360 Cal.) yr of the time being assessed
7D: Poorly dated

**Table A1**. Chronological control

| Site Number | Site Name | Site Type | TANN | TDJF | TJJA | PANN | PDJF | PJJA |
|---|---|---|---|---|---|---|---|---|
| 1 | MD95-2039 (M) | Marine | 15.7 | 10.7 | 20.8 | 1047 | 427 | 70 |
| 2 | SU81-18 (M) | Marine | 20.8 | 15.3 | 26.5 | 629 | 282 | 25 |
| 3 | MD99-2331 (M) | Marine | 14.6 | 9.8 | 19.4 | 1239 | 507 | 88 |
| 4 | Carn Morval | Lake | 12.5 | 8.7 | 16.9 | 1183 | 392 | 206 |
| 5 | Gorham Cave | Cave | 18.3 | 13.4 | 23.7 | 740 | 336 | 25 |
| 6 | Dozmary Pool | Lake | 10.3 | 6.0 | 15.2 | 1271 | 422 | 236 |
| 7 | Bajondillo | Cave | 16.6 | 10.5 | 23.4 | 542 | 223 | 27 |
| 8 | Laguna del maar de Fuentillejo | Lake | 16.1 | 8.1 | 25.4 | 474 | 156 | 47 |
| 9 | Padul-1 | Peat Bog | 16.6 | 9.6 | 24.9 | 417 | 157 | 23 |
| 10 | Padul-2 | Peat Bog | 16.6 | 9.6 | 24.9 | 417 | 157 | 23 |
| 11 | Cova di Carihuela | Cave | 15.7 | 8.1 | 25.1 | 551 | 187 | 57 |
| 12 | Ifri El Baroud | Cave | 16.9 | 10.7 | 24.0 | 457 | 184 | 22 |
| 13 | MD95-2043 (M) | Marine | 17.9 | 12.4 | 24.0 | 214.2 | 37 | 72 |
| 14 | San Rafael | Peat Bog | 18.1 | 11.9 | 24.9 | 243 | 87 | 14 |
| 15 | Siles | Lake | 14.4 | 6.8 | 23.4 | 658 | 195 | 92 |
| 16 | Torrecilla de Valmadrid | Colluvium | 14.2 | 6.6 | 22.5 | 390 | 75 | 82 |
| 17 | Navarrés-1 | Peat Bog | 17.0 | 10.9 | 23.8 | 421 | 96 | 51 |
| 18 | Navarrés-2 | Peat Bog | 17.0 | 10.9 | 23.8 | 421 | 96 | 51 |
| 19 | Tourbiere de l'Estarres | Lake | 13.0 | 6.1 | 20.4 | 1045 | 272 | 217 |
| 20 | Cova de les Malladetes | Cave | 18.1 | 12.1 | 24.8 | 478 | 117 | 60 |
| 21 | Lourdes | Lake | 12.6 | 5.5 | 20.1 | 1002 | 256 | 212 |
| 22 | Lake Estanya | Lake | 12.8 | 5.1 | 21.0 | 641 | 125 | 152 |
| 23 | Freychinede | Lake | 10.8 | 3.9 | 19.0 | 1128 | 257 | 277 |
| 24 | Banyoles | Lake | 14.3 | 7.7 | 21.9 | 698 | 157 | 139 |
| 25 | Lac du Bouchet B5 | Lake | 8.2 | 1.3 | 15.9 | 1070 | 251 | 221 |
| 26 | MD99-2348 (103) (M) | Marine | 14.6 | 8.0 | 21.9 | 618 | 158 | 95 |
| 27 | Les Echets G | Peat Bog | 11.4 | 3.6 | 19.6 | 876 | 175 | 215 |
| 28 | La Grotte Walou | Cave | 10.3 | 3.2 | 17.0 | 903 | 215 | 249 |
| 29 | Bergsee | Lake | 9.6 | 1.4 | 17.6 | 1048 | 189 | 387 |
| 30 | Garaat El-Ouez | Peat Bog | 17.3 | 11.0 | 24.3 | 830 | 360 | 33 |
| 31 | Pian del Lago | Lake | 12.4 | 5.1 | 20.0 | 995 | 266 | 149 |
| 32 | Pilsensee | Lake | 9.3 | 0.6 | 17.7 | 947 | 151 | 374 |
| 33 | Orgiano | Peat Bog | 13.0 | 3.3 | 22.3 | 907 | 200 | 228 |
| 34 | Lago della Costa | Lake | 12.9 | 3.3 | 22.1 | 888 | 196 | 224 |
| 35 | Lagaccione | Lake | 14.2 | 7.2 | 21.7 | 705 | 203 | 109 |
| 36 | Lago Vico | Lake | 13.7 | 6.4 | 21.5 | 870 | 258 | 132 |
| 37 | Stracciacappa | Lake | 14.6 | 7.3 | 22.4 | 867 | 266 | 115 |
| 38 | Lago di Monterosi | Lake | 15.0 | 7.7 | 22.9 | 837 | 248 | 115 |
| 39 | Venice | Peat Bog | 13.4 | 4.5 | 22.1 | 1050 | 221 | 277 |
| 40 | Azzano Decimo | Alluvial Fan | 13.3 | 4.4 | 22.1 | 1170 | 241 | 311 |
| 41 | Valle di Castiglione | Lake | 16.3 | 9.1 | 24.0 | 988 | 294 | 144 |
| 42 | Travesio | Lake | 12.6 | 3.7 | 21.3 | 1415 | 281 | 375 |
| 43 | Orvenco | Alluvial Fan | 13.0 | 3.3 | 22.3 | 907 | 200 | 228 |
| 44 | Rio Doidis | Lake | 12.8 | 4.1 | 21.2 | 1529 | 315 | 392 |
| 45 | Billerio | Lake | 12.8 | 4.1 | 21.2 | 1529 | 315 | 392 |
| 46 | Kersdorf-Briesen | Lake | 8.8 | -1.0 | 17.9 | 538 | 110 | 175 |
| 47 | Lago Grande di Monticchio | Lake | 11.5 | 4.1 | 19.8 | 518 | 154 | 76 |
| 48 | Nagymohos | Peat Bog | 9.5 | -1.5 | 19.1 | 616 | 103 | 230 |
| 49 | Safarka | Peat Bog | 7.0 | -3.2 | 16.0 | 755 | 119 | 280 |
| 50 | Feher Lake | Lake | 11.0 | -0.1 | 20.7 | 546 | 112 | 185 |
| 51 | Ioannina | Peat Bog | 14.7 | 6.5 | 23.3 | 1000 | 364 | 98 |
| 52 | Kokad | Peat Bog | 10.2 | -0.9 | 19.8 | 601 | 130 | 204 |
| 53 | Lake Xinias | Lake | 15.6 | 7.5 | 24.1 | 563 | 211 | 47 |
| 54 | Mickunai | Lake | 6.0 | -5.0 | 16.3 | 682 | 131 | 230 |
| 55 | Lake Sfanta Anna | Lake | 11.6 | 5.2 | 18.4 | 867 | 253 | 172 |
| 56 | Megali Limni | Lake | 15.5 | 8.2 | 23.4 | 684 | 357 | 28 |
| 57 | Straldzha | Peat Bog | 12.5 | 2.6 | 21.8 | 591 | 158 | 135 |
| 58 | MD01-2430 (M) | Marine | 18.0 | 8.7 | 27.5 | 595 | 219 | 75 |
| 59 | Lake Iznik | Lake | 13.9 | 6.1 | 21.8 | 677 | 250 | 85 |
| 60 | M72/5 628-1 (M) | Marine | 14.5 | 8.0 | 21.6 | 857 | 251 | 156 |
| 61 | Dziguta | Peat Bog | 14.1 | 6.6 | 21.7 | 1549 | 409 | 373 |
| 62 | Lake Van LG | Lake | 12.0 | 0.9 | 23.1 | 635 | 201 | 34 |
| 63 | Lake Zeribar | Lake | 17.1 | 5.0 | 29.0 | 427 | 167 | 6 |

**Table A2.** Modern climate values for each site used in the calculation of anomalies (taken
from WorldClim 2, Fick & Hijmans 2017)


| Biome | Control | Change in Biome compared to the Control | | | | | | | |
|---|---|---|---|---|---|---|---|---|---|
| | | 0 Pinaceae | +5% Pinaceae | +10% Pinaceae | +20% Pinaceae | +50% Pinaceae | +100% Pinaceae | +200% Pinaceae | +400% Pinaceae |
| CLDE | 25 | 454 | 0 | 0 | 0 | -1 | -1 | -4 | -4 |
| TAIG | 1489 | -1430 | 16 | 38 | 74 | 192 | 337 | 554 | 914 |
| CLMX | 70 | 108 | 1 | 2 | 3 | -6 | -4 | 4 | 6 |
| COCO | 388 | -388 | 0 | -1 | 3 | 6 | 25 | 50 | 74 |
| TEDE | 33 | 16 | 1 | 1 | 1 | -1 | -2 | -8 | -5 |
| COMX | 2952 | -761 | 1 | 8 | 14 | -4 | -42 | -101 | -284 |
| WAMX | 418 | -28 | -1 | 0 | -1 | -6 | -11 | -29 | -62 |
| XERO | 699 | -323 | 3 | 4 | 12 | 45 | 68 | 113 | 180 |
| DESE | 0 | 0 | 0 | 0 | 0 | 0 | 0 | 0 | 0 |
| STEP | 1752 | 1388 | -14 | -39 | -83 | -173 | -296 | -468 | -663 |
| TUND | 387 | 964 | -7 | -13 | -23 | -52 | -74 | -111 | -156 |
| *Total* | *8213* | *5860* | *44* | *106* | *214* | *486* | *860* | *1442* | *2348* |


**Table A3.** This shows the results of experiment to test the sensitivity of pollen Biomes to
changes in the amount of Pinaceae in the pollen assemblage using 8213 modern pollen samples
from the EMPD2. Pinaceae can be over-represented in marine samples, and it has been
proposed that removing all Pinaceae from these samples is better than leaving the Pinaceae in
the pollen assemblage. The 'Control' column on the left shows the number of samples that were
classified for each Biome without changing the amount of Pinaceae (ie using the original pollen
assemblage). The other 8 columns to the right show the number of samples where the Biome
changed relative to the number shown in the control column as a result of either removal of all
Pinaceae ('0 Pinaceae'), or by artificially increasing the amount of Pinaceae respectively from
5 to 400% of the original count (+5% Pinaceae' to '+400% Pinaceae'). For instance, for the
CLDE (Cold Deciduous) Biome, 25 pollen samples were classified as CLDE without any
change in Pinaceae ('Control'), but 454 more samples were classified as CLDE when all
Pinaceae was removed ('0 Pinaceae') compared to 4 fewer samples that were classified as
CLDE when Pinaceae was increased by as much as 400% ('+400% Pinaceae'). The totals along
the bottom show that out of the 8213 pollen samples included in the experiment, 5860 biomes
changed when all Pinaceae was removed, compared to up to 2348 when Pinaceae was
artificially increased by up to 400%.

**Temperature Anomlay**

| Site Name | Site Number | TANN delta | | TDJF delta | | TJJA delta | | PANN delta | | PDJF delta | | PJJA delta | |
|---|---|---|---|---|---|---|---|---|---|---|---|---|---|
| | | Pinaceae | No Pinaceae | Pinaceae | No Pinaceae | Pinaceae | No Pinaceae | Pinaceae | No Pinaceae | Pinaceae | No Pinaceae | Pinaceae | No Pinaceae |
| MD95-2039 (M) | 1 | -10.5 | -12.3 | -14.6 | -17.9 | -5.7 | -5.9 | -234.3 | -236.3 | -205.4 | -196.4 | 83.1 | 63.0 |
| SU81-18 (M) | 2 | -13.3 | -21.4 | -15.2 | -23.0 | -10.4 | -17.7 | 183.3 | 703.2 | -21.1 | 124.4 | 85.1 | 167.7 |
| MD99-2331 (M) | 3 | -5.3 | -4.8 | -7.5 | -7.0 | -2.3 | -1.4 | -420.6 | -435.6 | -257.7 | -251.1 | 39.4 | 19.0 |
| MD95-2043 (M) | 13 | -7.0 | -6.0 | -11.0 | -9.9 | -3.6 | -2.7 | 304.6 | 332.5 | 178.4 | 201.9 | -17.3 | -22.9 |
| MD99-2348 (103) (M) | 26 | -7.8 | -8.8 | -10.0 | -11.5 | -6.5 | -7.3 | 164.7 | 218.0 | 12.1 | 7.6 | 124.0 | 179.5 |
| MD01-2430 (M) | 58 | -11.1 | -13.5 | -11.3 | -14.5 | -11.0 | -12.8 | 200.6 | 349.1 | -20.8 | 31.4 | 113.1 | 127.9 |
| M72/5 628-1 (M) | 60 | -2.3 | -0.5 | -4.9 | -3.0 | -0.2 | 1.7 | -298.0 | -311.1 | -116.8 | -100.1 | -27.0 | -51.7 |
| Site Average | | -8.2 | -9.6 | -10.6 | -12.4 | -5.7 | -6.6 | -14.2 | 88.5 | -61.6 | -26.0 | 57.2 | 68.9 |

**Standard Deviation**

| Site Name | Site Number | TANN STDEV | | TDJF STDEV | | TJJA STDEV | | PANN STDEV | | PDJF STDEV | | PJJA STDEV | |
|---|---|---|---|---|---|---|---|---|---|---|---|---|---|
| | | Pinaceae | No Pinaceae | Pinaceae | No Pinaceae | Pinaceae | No Pinaceae | Pinaceae | No Pinaceae | Pinaceae | No Pinaceae | Pinaceae | No Pinaceae |
| MD95-2039 (M) | 1 | 4.6 | 3.7 | 7.5 | 4.0 | 2.9 | 4.0 | 330.6 | 268.9 | 110.5 | 96.7 | 53.6 | 55.6 |
| SU81-18 (M) | 2 | 3.1 | 4.0 | 4.0 | 3.4 | 2.8 | 4.8 | 297.1 | 149.3 | 126.6 | 58.0 | 47.3 | 28.4 |
| MD99-2331 (M) | 3 | 2.9 | 4.0 | 3.4 | 3.8 | 2.8 | 5.0 | 302.6 | 368.6 | 103.5 | 134.2 | 57.6 | 64.3 |
| MD95-2043 (M) | 13 | 2.0 | 4.7 | 2.4 | 5.6 | 2.1 | 4.1 | 115.9 | 121.5 | 59.3 | 72.1 | 36.2 | 25.2 |
| MD99-2348 (103) (M) | 26 | 2.4 | 3.8 | 3.0 | 4.2 | 2.7 | 4.6 | 192.7 | 242.9 | 75.8 | 68.2 | 58.9 | 52.1 |
| MD01-2430 (M) | 58 | 5.1 | 2.3 | 8.1 | 2.0 | 3.9 | 2.8 | 218.7 | 182.8 | 78.9 | 58.3 | 53.4 | 45.0 |
| M72/5 628-1 (M) | 60 | 3.2 | 3.9 | 3.8 | 4.0 | 3.5 | 4.6 | 149.0 | 171.9 | 67.1 | 48.2 | 56.5 | 61.8 |
| Site Average | | 3.3 | 3.8 | 4.6 | 3.9 | 3.0 | 4.3 | 229.5 | 215.1 | 88.8 | 76.5 | 51.9 | 47.5 |

**Table A4.** A comparison of the LGM reconstructed climate for marine sites showing the effect of excluding Pinaceae (shaded) from the pollen assemblage, compared to the results of including Pinacea (unshaded, also presented in Figures 6-8). It has been proposed that because of the potential for over-representation of Pinaceae in marine pollen samples, it is better to exclude Pinaceae completely from marine pollen samples. Comparing the two approaches, temperatures are generally 1-2C cooler, and precipitation slightly higher when Pinaceae is excluded. The differences for both temperature and precipitation are significantly less than the standard deviation of their uncertainties.

| | All surface samples | | Steppe only | |
|---|---|---|---|---|
| | RMSE | R2 | RMSE | R2 |
| TANN | 2.28 | 0.9 | 2.51 | 0.87 |
| TDJF | 3.35 | 0.91 | 3.26 | 0.88 |
| TJJA | 2.21 | 0.81 | 2.49 | 0.82 |
| PANN | 224.94 | 0.69 | 185.7 | 0.71 |
| PDJF | 78.51 | 0.69 | 66.5 | 0.66 |
| PJJA | 52.49 | 0.75 | 43.8 | 0.79 |

**Table A5.** A comparison of MAT performance statistics based on the modern pollen sample training set using all surface samples from the EMPD2 used in the LGM reconstruction (as shown in Table 3), and a subset of 1588 samples from the EMPD2 that were classified as steppe. The results show little difference between the two different types of samples. The table includes Mean Annual Temperature and Precipitation (TANN and PANN), Mean Winter Temperature and Precipitation (TDJF and PDJF) and Mean Summer Temperature and Precipitation (TJJA and PJJA).

| Site Name | Site# | Pollen Biome | Modern Analogue Biome | Modern Analogue Ecoregion |
|---|---|---|---|---|
| MD95-2039 | 1 | XERO | Mediterranean Forests, woodlands and scrubs | Iberian conifer forests |
| SU81-18 | 2 | COMX | Mediterranean Forests, woodlands and scrubs | Iberian conifer forests |
| MD99-2331 | 3 | STEP | Mediterranean Forests, woodlands and scrubs | Alps conifer and mixed forests |
| Carn Morval | 4 | STEP | Temperate broadleaf and mixed forests | North Atlantic moist mixed forests |
| Gorham Cave | 5 | STEP | Mediterranean Forests, woodlands and scrubs | Cyprus Mediterranean forests |
| Dozmary Pool | 6 | STEP | Temperate Coniferous Forest | Alps conifer and mixed forests |
| Bajondillo | 7 | STEP | Temperate broadleaf and mixed forests | Central European mixed forests |
| Laguna del maar de Fuentillejo | 8 | COMX | Mediterranean Forests, woodlands and scrubs | Northwest Iberian montane forests |
| Padul | 9 | STEP | Mediterranean Forests, woodlands and scrubs | Central Anatolian steppe |
| Padul-15-05 | 10 | WAMX | Mediterranean Forests, woodlands and scrubs | Iberian sclerophyllous and semi-deciduous forests |
| Cova di Carihuela | 11 | STEP | Deserts and xeric shrublands | Azerbaijan shrub desert and steppe |
| Ifri El Baroud | 12 | STEP | Mediterranean Forests, woodlands and scrubs | Iberian sclerophyllous and semi-deciduous forests |
| MD95-2043 | 13 | CLMX | Mediterranean Forests, woodlands and scrubs | Southern Anatolian montane conifer and deciduous forests |
| San Rafael | 14 | XERO | Mediterranean Forests, woodlands and scrubs | Tyrrhenian-Adriatic Sclerophyllous and mixed forests |
| Siles | 15 | XERO | Mediterranean Forests, woodlands and scrubs | Northwest Iberian montane forests |
| Torrecilla de Valmadrid | 16 | STEP | Mediterranean Forests, woodlands and scrubs | Southern Anatolian montane conifer and deciduous forests |
| Navarres | 17 | XERO | Mediterranean Forests, woodlands and scrubs | Iberian sclerophyllous and semi-deciduous forests |
| Navarres | 18 | STEP | Temperate broadleaf and mixed forests | Pyrenees conifer and mixed forests |
| Tourbiere de lEstarres | 19 | STEP | Temperate grasslands, savannas and shrublands | Eastern Anatolian montane steppe |
| Cova de les Malladetes | 20 | XERO | Mediterranean Forests, woodlands and scrubs | Pyrenees conifer and mixed forests |
| Lourdes | 21 | STEP | Temperate broadleaf and mixed forests | Gissaro-Alai open woodlands |
| Estanya | 22 | XERO | Temperate broadleaf and mixed forests | Western Siberian hemiboreal forests |
| Freychinede | 23 | STEP | Temperate grasslands, savannas and shrublands | Mongolian-Manchurian grassland |
| Lake Banyoles | 24 | STEP | Temperate grasslands, savannas and shrublands | Gissaro-Alai open woodlands |
| Lac du Bouchet B5 | 25 | STEP | Temperate grasslands, savannas and shrublands | Gissaro-Alai open woodlands |
| MD99-2348-103 | 26 | COMX | Temperate broadleaf and mixed forests | Rodope montane mixed forests |
| Les Echets G - DIGI | 27 | STEP | Temperate broadleaf and mixed forests | Western Siberian hemiboreal forests |
| La Grotte Walou | 28 | STEP | Temperate broadleaf and mixed forests | Kazakh forest steppe |
| Bergsee | 29 | STEP | Temperate broadleaf and mixed forests | Kazakh forest steppe |
| Garaat El-Ouez | 30 | STEP | Mediterranean Forests, woodlands and scrubs | Anatolian conifer and deciduous mixed forests |
| Pian del Lago | 31 | COMX | Temperate broadleaf and mixed forests | Western European broadleaf forests |
| Pilsensee | 32 | TAIG | Tundra | Kola Peninsula tundra |
| Orgiano | 33 | COMX | Temperate broadleaf and mixed forests | Western European broadleaf forests |
| Lago della Costa | 34 | COMX | Temperate Coniferous Forest | Alps conifer and mixed forests |
| Lagaccione | 35 | STEP | Temperate grasslands, savannas and shrublands | Gissaro-Alai open woodlands |
| Lago Vico | 36 | STEP | Temperate grasslands, savannas and shrublands | Gissaro-Alai open woodlands |
| Stracciacappa | 37 | STEP | Mediterranean Forests, woodlands and scrubs | Western European broadleaf forests |
| Lago di Monterosi | 38 | STEP | Temperate grasslands, savannas and shrublands | Northwest Iberian montane forests |
| Venice | 39 | XERO | Tundra | Scandinavian Montane Birch forest and grasslands |
| Azzano Decimo | 40 | XERO | Temperate broadleaf and mixed forests | Scandinavian Montane Birch forest and grasslands |
| Valle di Castiglione | 41 | STEP | Temperate broadleaf and mixed forests | Tian Shan montane steppe and meadows |
| Travesio | 42 | XERO | Mediterranean Forests, woodlands and scrubs | Iberian conifer forests |
| Orvenco | 43 | TAIG | Temperate broadleaf and mixed forests | Western Siberian hemiboreal forests |
| Rio Doidis | 44 | XERO | Mediterranean Forests, woodlands and scrubs | Cyprus Mediterranean forests |
| Billerio | 45 | TAIG | Temperate broadleaf and mixed forests | Western Siberian hemiboreal forests |
| Kersdorf-Briesen | 46 | TAIG | Temperate broadleaf and mixed forests | Western Siberian hemiboreal forests |
| Lago Grande di Monticchio | 47 | STEP | Temperate broadleaf and mixed forests | Tian Shan montane steppe and meadows |
| Nagymohos Pleistocene | 48 | STEP | Tundra | Sarmatic mixed forests |
| Safarka | 49 | TAIG | Boreal forests / Taiga | Ural montane forests and tundra |
| Feher-to | 50 | COMX | Temperate Coniferous Forest | Alps conifer and mixed forests |
| Ioannina | 51 | STEP | Temperate broadleaf and mixed forests | Central European mixed forests |
| Kokad | 52 | STEP | Temperate broadleaf and mixed forests | East European forest steppe |
| Lake Xinias | 53 | STEP | Temperate broadleaf and mixed forests | Western European broadleaf forests |
| Mickunai | 54 | COCO | Tundra | Scandinavian Montane Birch forest and grasslands |
| Lake Sfanta Anna | 55 | COMX | Temperate Coniferous Forest | Alps conifer and mixed forests |
| Lesvos ML01 Megali Limni | 56 | STEP | Temperate broadleaf and mixed forests | Rodope montane mixed forests |
| Straldzha | 57 | STEP | Temperate broadleaf and mixed forests | Aegean and Western Turkey sclerophyllous and mixed forests |
| MD01-2430 | 58 | STEP | Temperate broadleaf and mixed forests | Euxine-Colchic broadleaf forests |
| Lake Iznik | 59 | STEP | Temperate broadleaf and mixed forests | Tian Shan montane steppe and meadows |
| M72/5 628-1 | 60 | STEP | Deserts and xeric shrublands | Azerbaijan shrub desert and steppe |
| Dziguta Core 1 | 61 | CLMX | Temperate broadleaf and mixed forests | Northeastern Spain and Southern France Mediterranean forests |
| Lake Van LG | 62 | STEP | Mediterranean Forests, woodlands and scrubs | Aegean and Western Turkey sclerophyllous and mixed forests |
| Lake Zeribar | 63 | STEP | Temperate grasslands, savannas and shrublands | Pontic steppe |

*Notes: Modern analogue Biomes and Ecoregions were calculated as the most commonly occuring amongst all 6 best modern analogue pollen samples in all LGM samples for each pollen site/record. These are taken from the EMPD2 (Davis et al 2020), using the classification of Olsen et al 2001.*

**Table A6**. The biome and ecoregion of the modern surface samples used as analogues in the pollen-climate reconstructions.



**Figures**



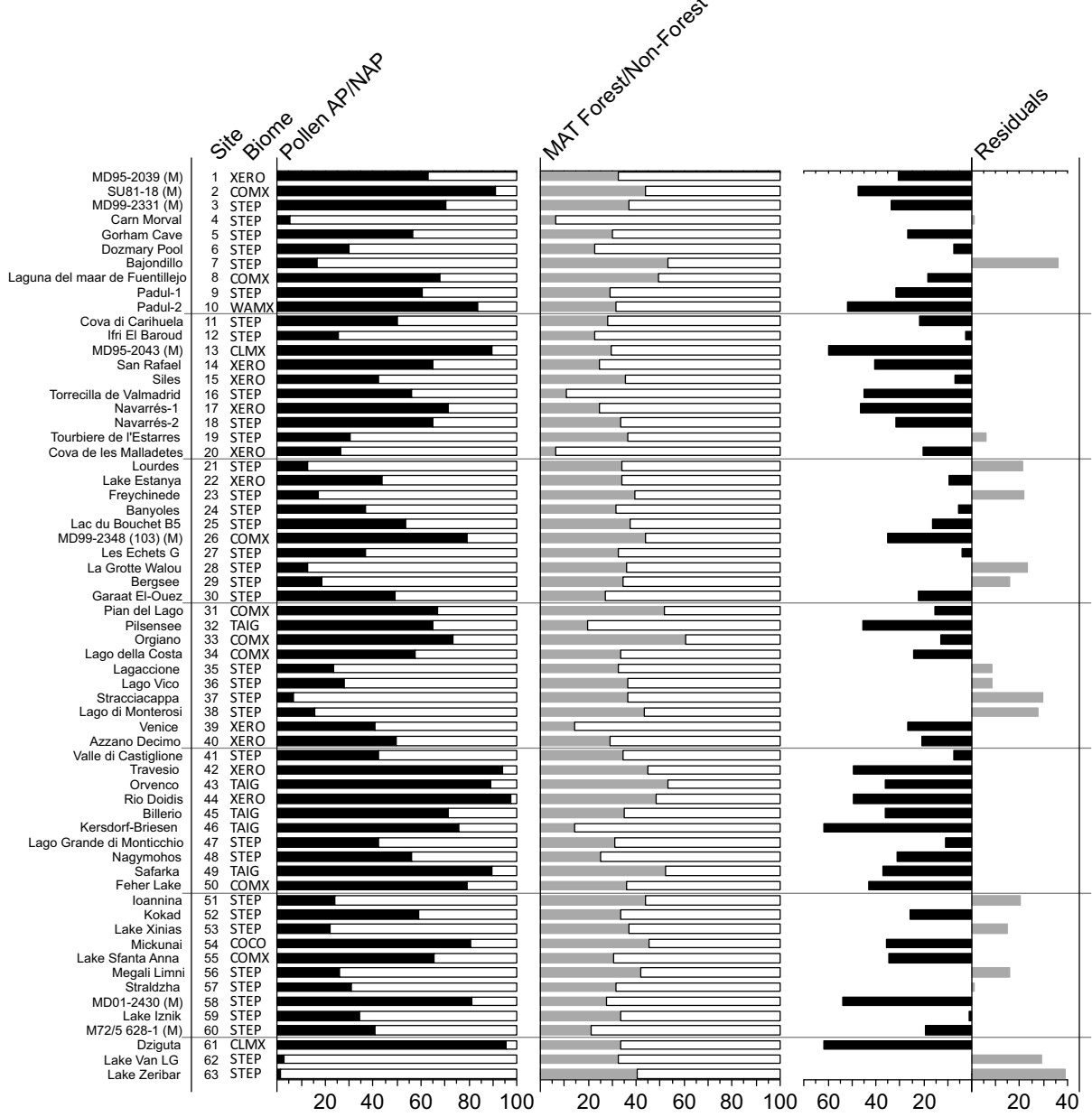



Figure A1. Pollen biomes (see figure 2 for key), Arboreal Pollen (AP) % forest cover, MAT
% forest cover and residuals (AP % compared to MAT Forest %)

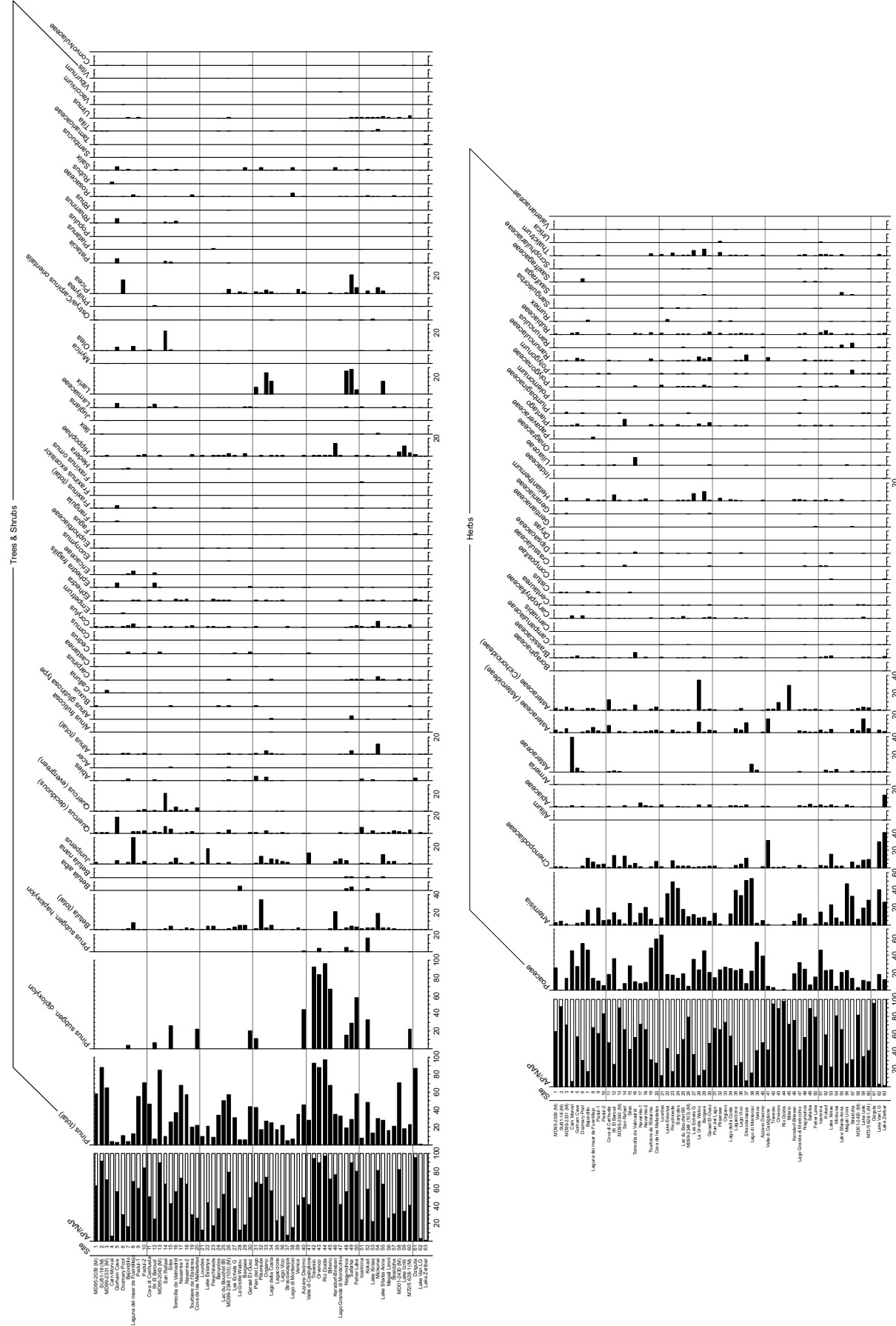

Figure A2. Pollen taxa percentages for all LGM sites/records

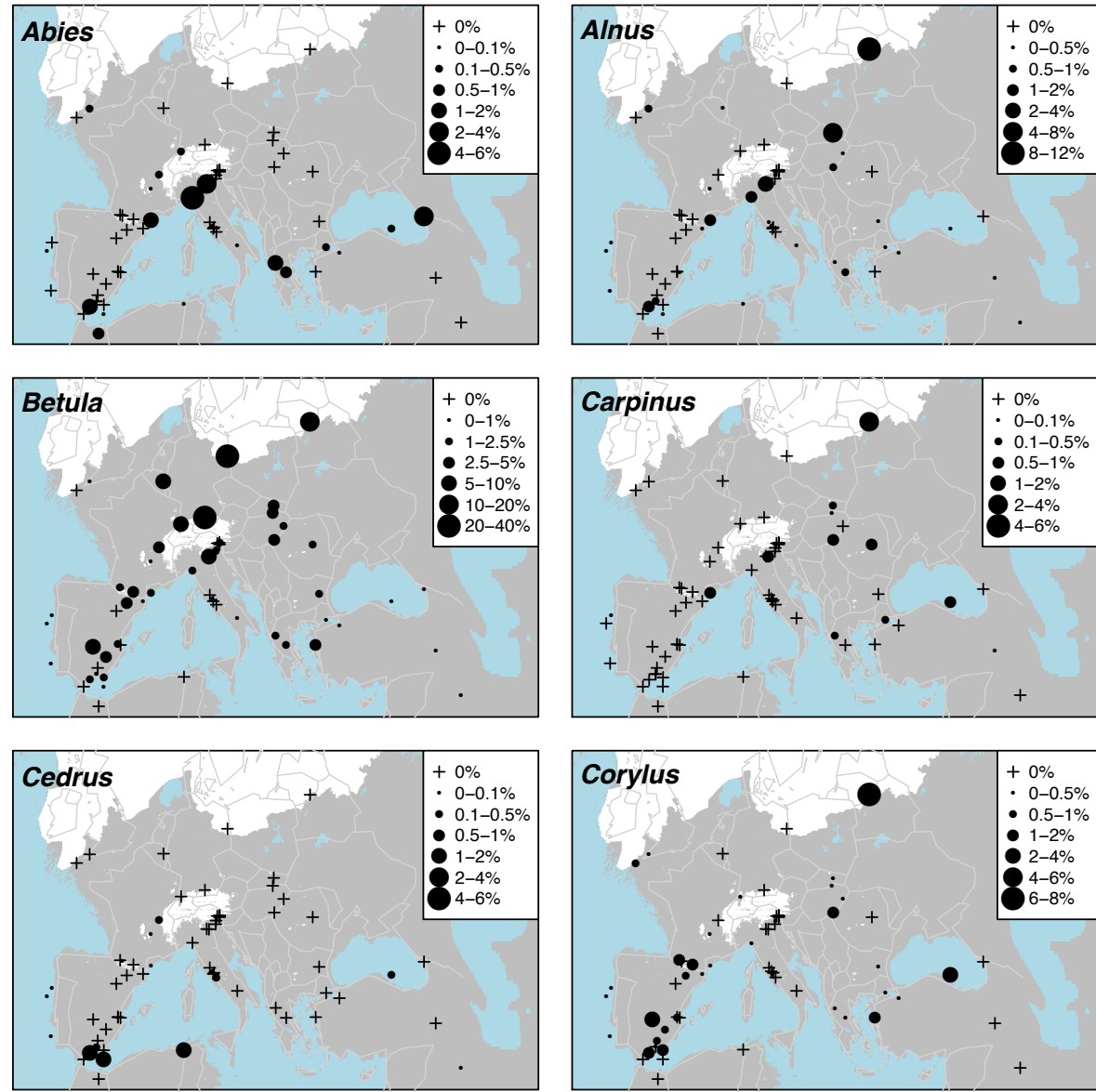

Figure A3a. Percentage maps of *Abies*, *Alnus*, *Betula*, *CarPinus*, *Cedrus* and *Corylus*

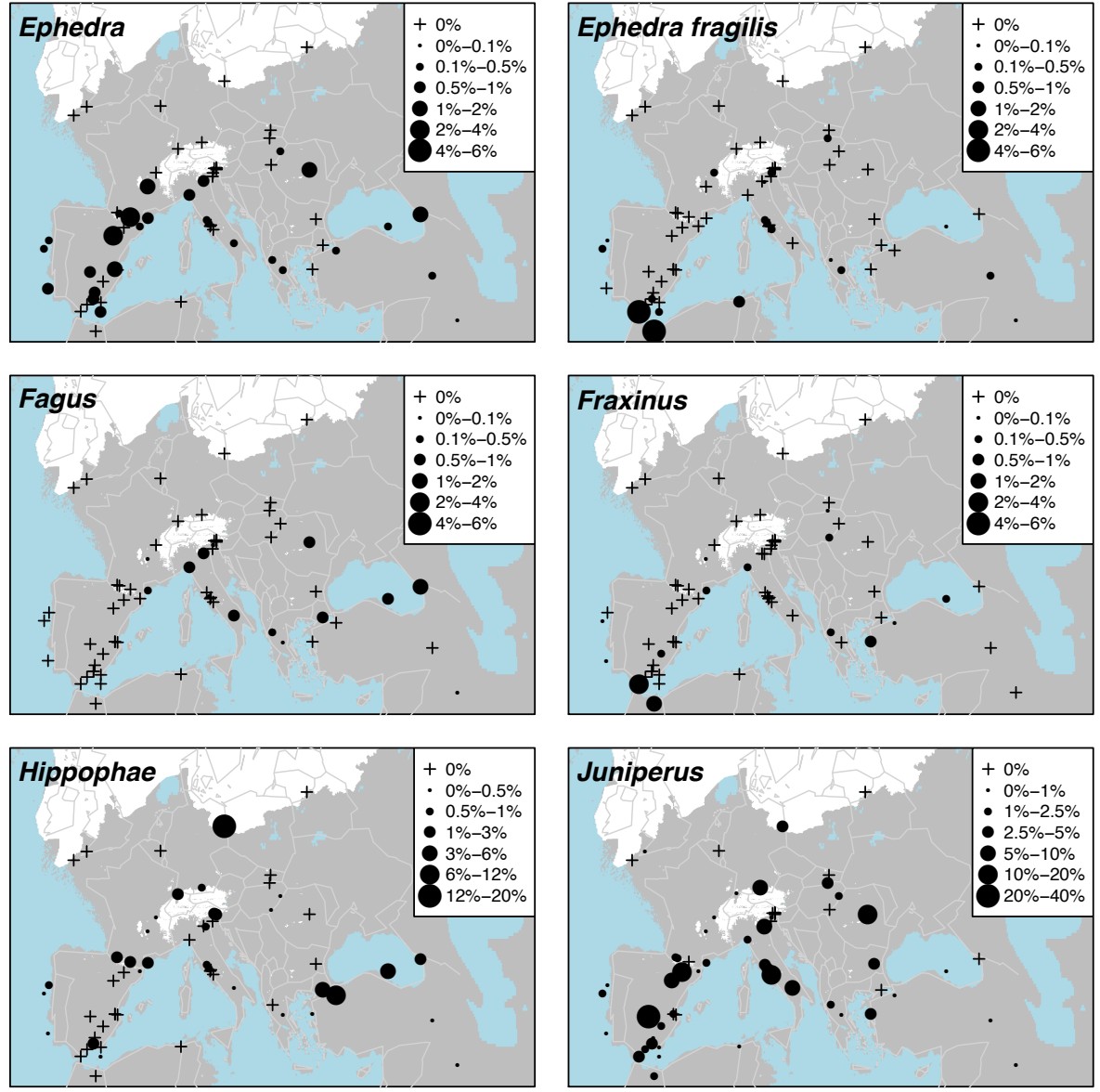

Figure A3b. Percentage maps of *Ephedra*, *Ephedra fragilis*, *Fagus*, *Fraxinus*, *Hippophae* and
*Juniperus*

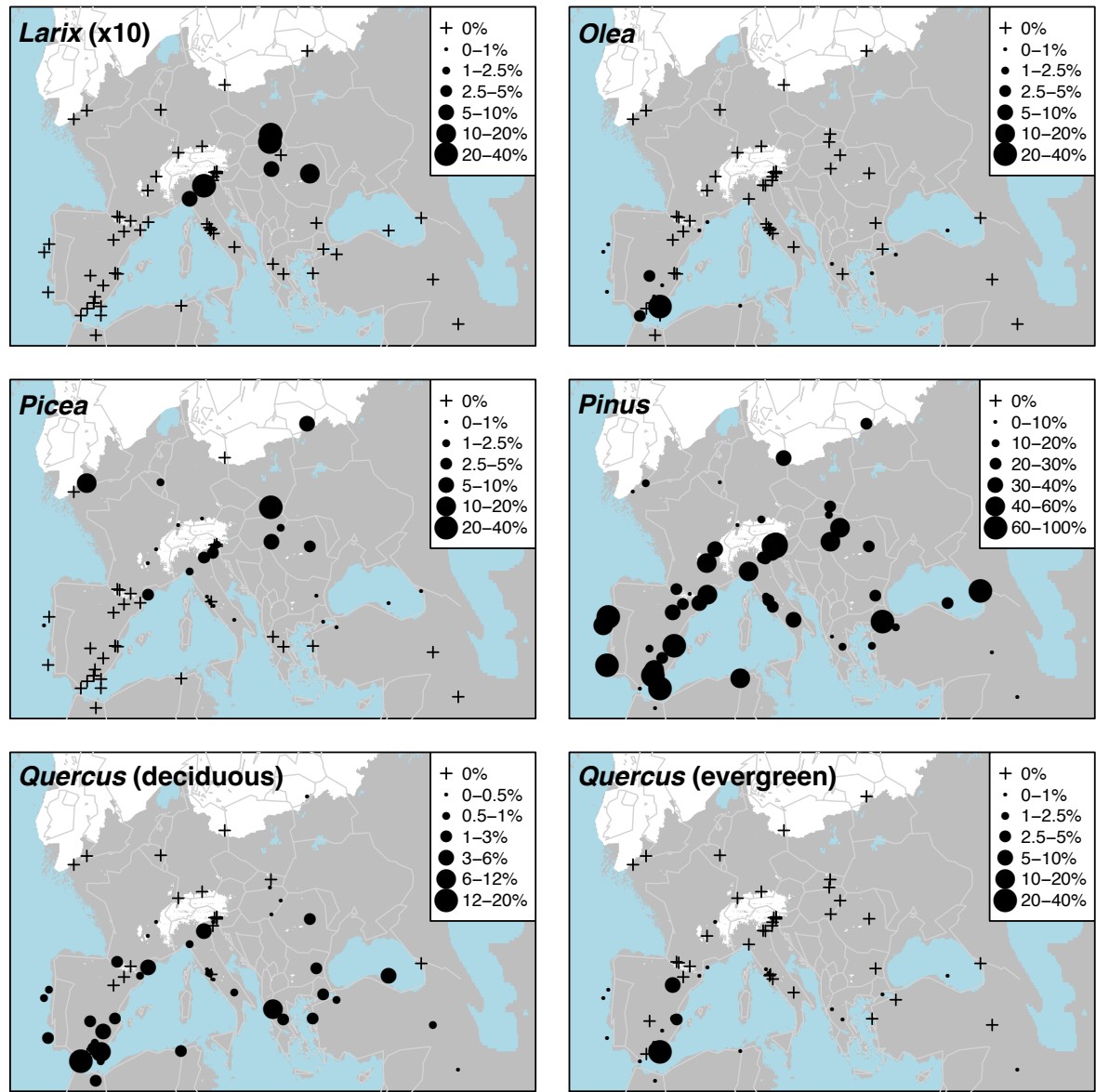



Figure A3c. Percentage maps of *Larix* (x10 exaggeration), *Olea*, *Picea*, *Pinus*, *Quercus*
(deciduous) and *Quercus* (evergreen)

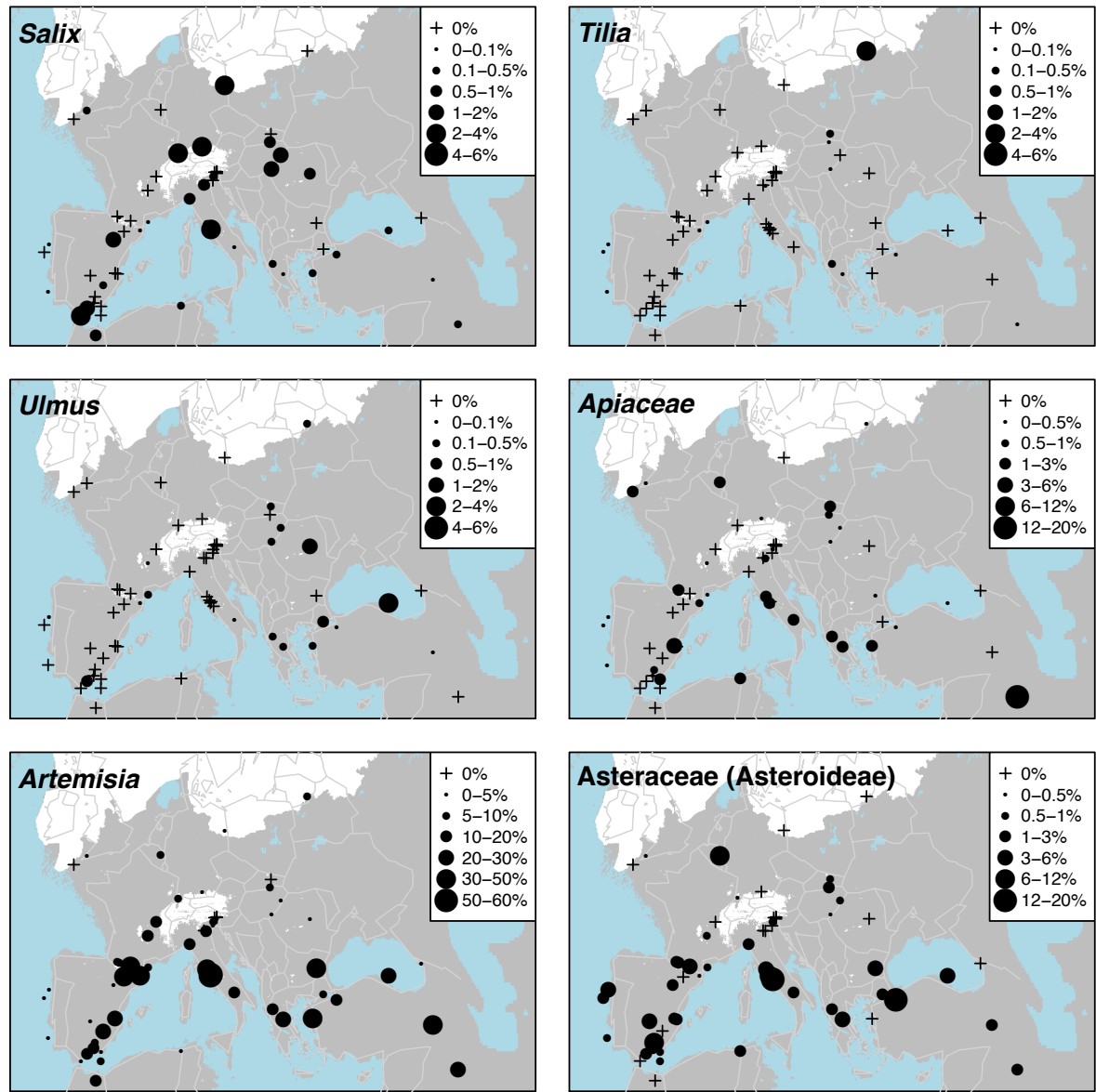

Figure A3d. Percentage maps of *Salix*, *Tilia*, *Ulmus*, Apiaceae, *Artemisia* and Asteraceae
(Asteroideae)

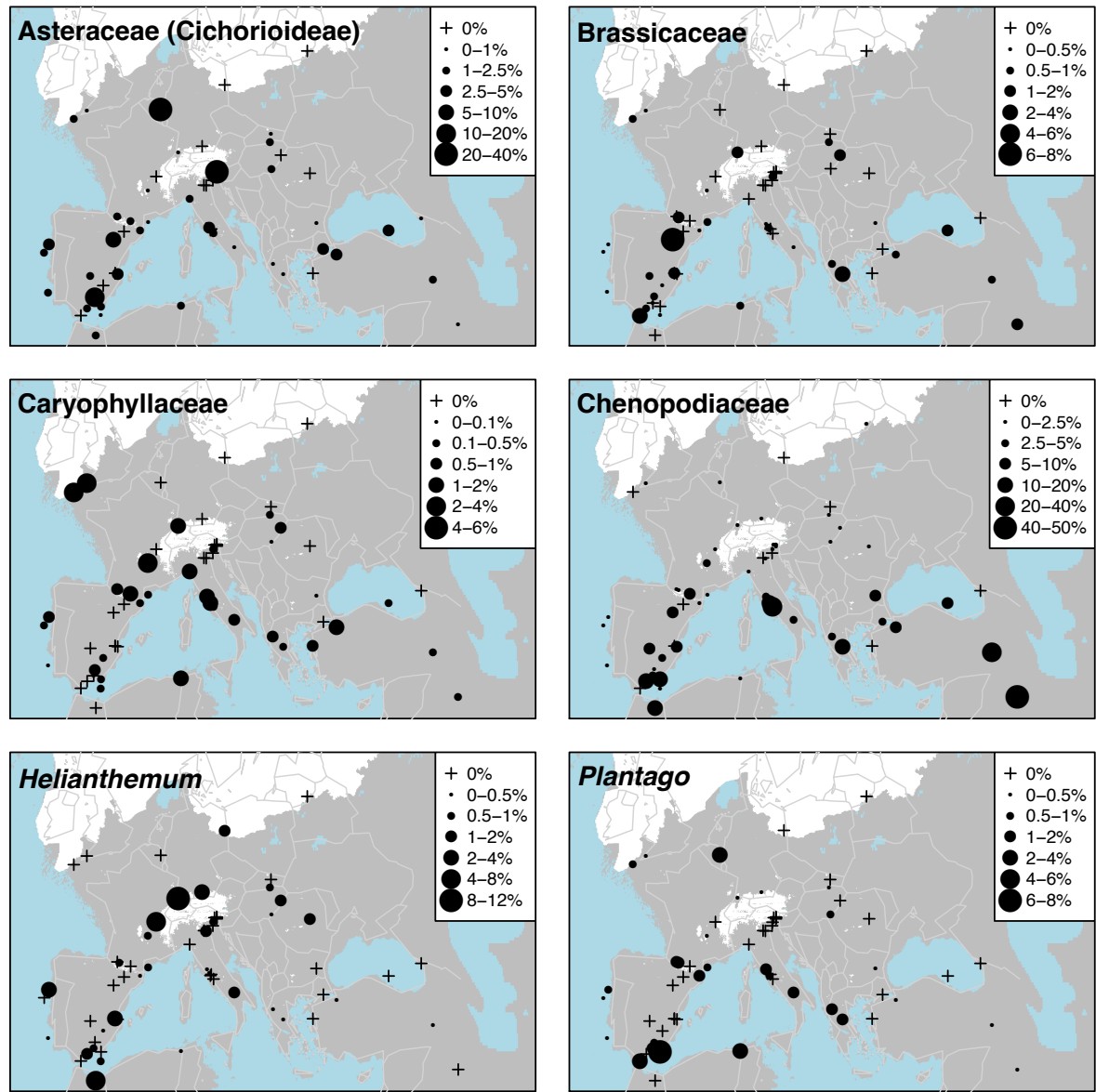

Figure A3e. Percentage maps of Asteraceae (Cichorioideae), Brassicaceae, Caryophyllaceae,
Chenopodiaceae, Helianthemum and *Plantago*

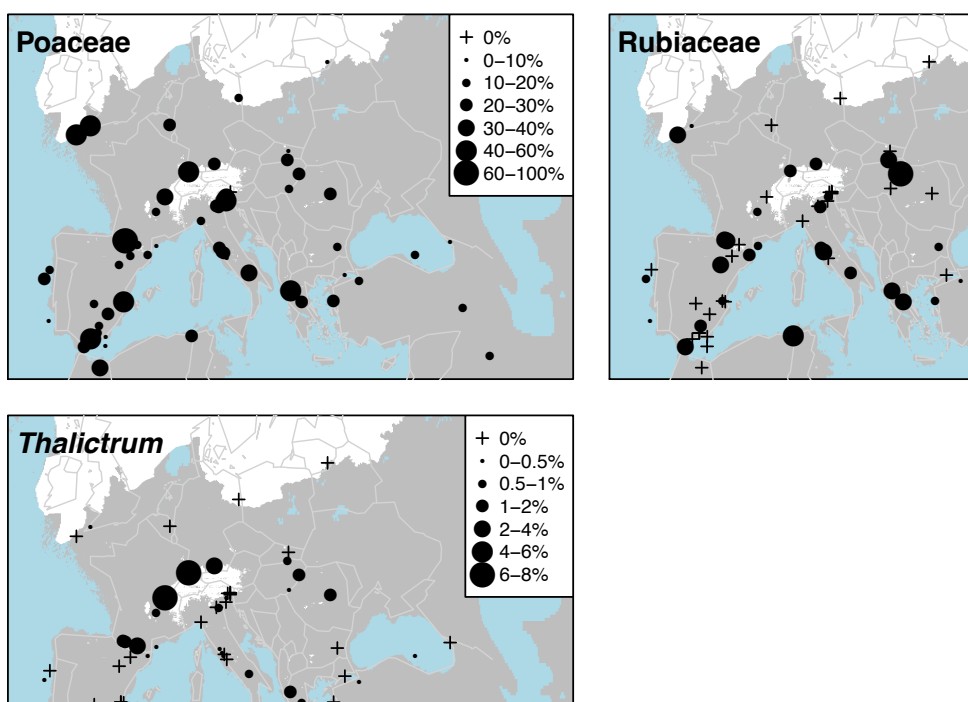

Figure A3f. Percentage maps of Poaceae, Rubiaceae and *Thalictrum*


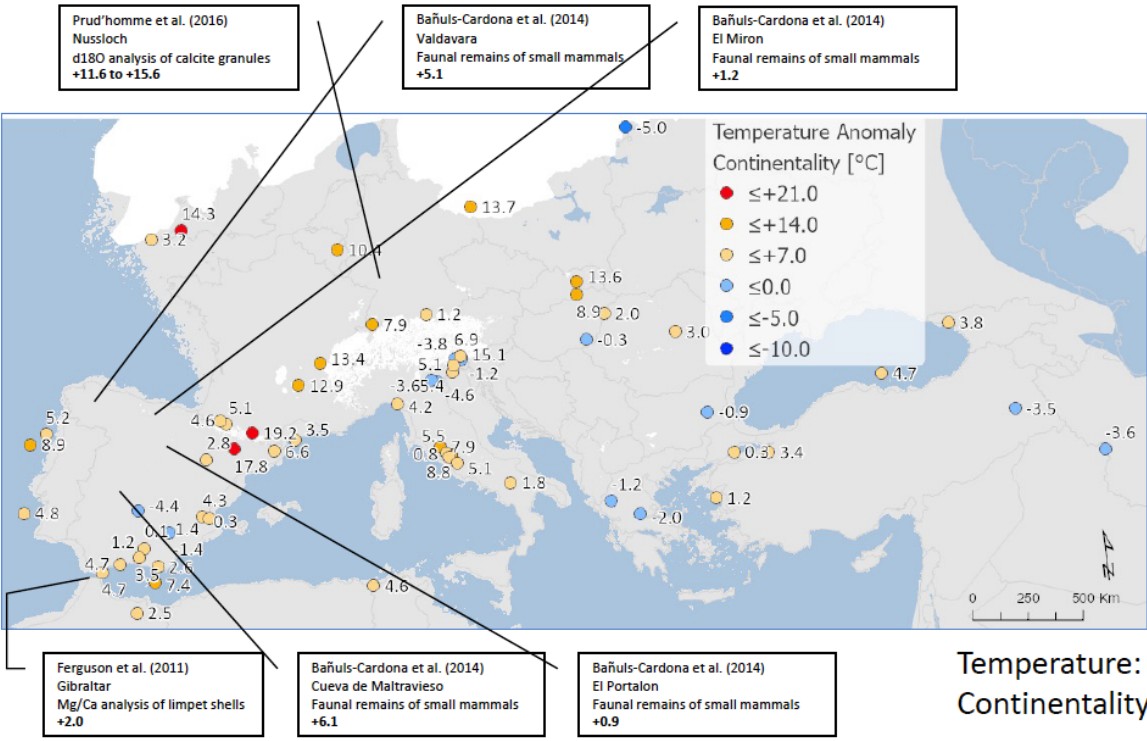

Prud'homme et al. (2016)
Nussloch
d18O analysis of calcite granules
**+11.6 to +15.6**

Bañuls-Cardona et al. (2014)
Valdavara
Faunal remains of small mammals
**+5.1**

Bañuls-Cardona et al. (2014)
El Miron
Faunal remains of small mammals
**+1.2**

Temperature Anomaly
Continentality [°C]
- ≤+21.0
- ≤+14.0
- ≤+7.0
- ≤0.0
- ≤-5.0
- ≤-10.0

Ferguson et al. (2011)
Gibraltar
Mg/Ca analysis of limpet shells
**+2.0**

Bañuls-Cardona et al. (2014)
Cueva de Maltravieso
Faunal remains of small mammals
**+6.1**

Bañuls-Cardona et al. (2014)
El Portalon
Faunal remains of small mammals
**+0.9**

Temperature:
Continentality


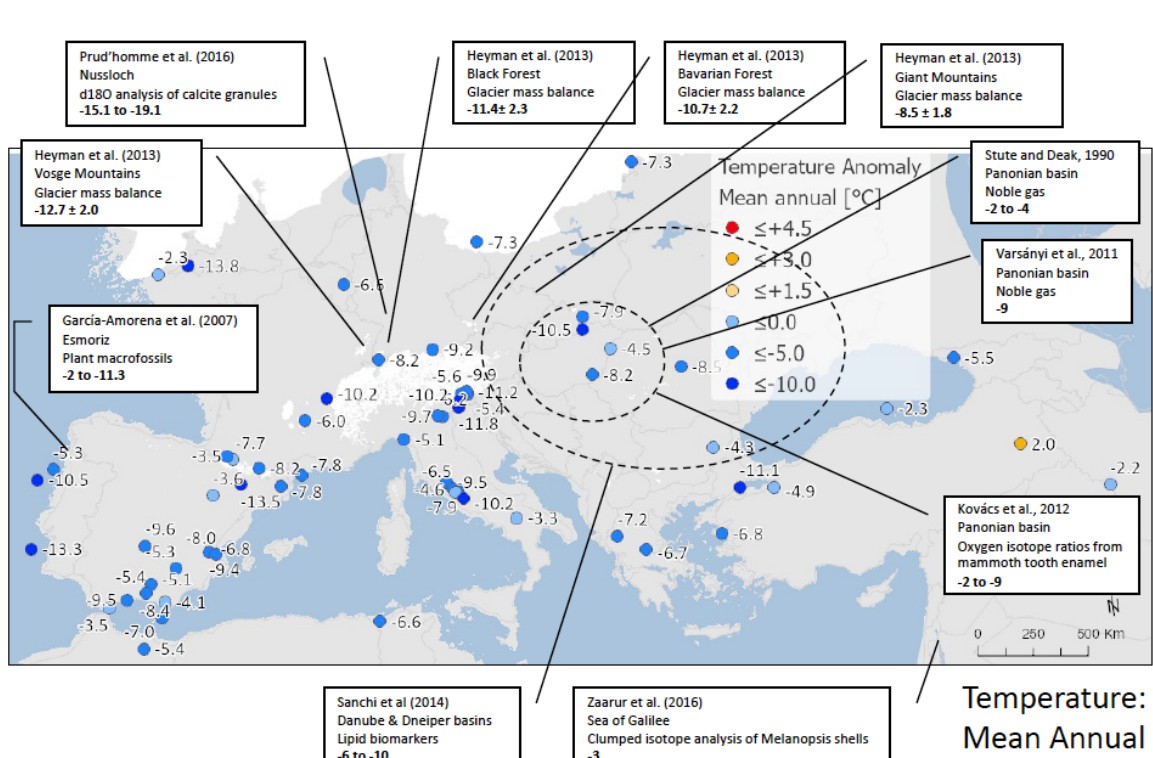

Prud'homme et al. (2016)
Nussloch
d18O analysis of calcite granules
**-15.1 to -19.1**

Heyman et al. (2013)
Black Forest
Glacier mass balance
**-11.4± 2.3**

Heyman et al. (2013)
Bavarian Forest
Glacier mass balance
**-10.7± 2.2**

Heyman et al. (2013)
Giant Mountains
Glacier mass balance
**-8.5 ± 1.8**

Heyman et al. (2013)
Vosge Mountains
Glacier mass balance
**-12.7 ± 2.0**

Stute and Deak, 1990
Panonian basin
Noble gas
**-2 to -4**

Varsányi et al., 2011
Panonian basin
Noble gas
**-9**

García-Amorena et al. (2007)
Esmoriz
Plant macrofossils
**-2 to -11.3**

Temperature Anomaly
Mean annual [°C]
- ≤+4.5
- ≤+3.0
- ≤+1.5
- ≤0.0
- ≤-5.0
- ≤-10.0

Kovács et al., 2012
Panonian basin
Oxygen isotope ratios from
mammoth tooth enamel
**-2 to -9**

Sanchi et al (2014)
Danube & Dneiper basins
Lipid biomarkers
**-6 to -10**

Zaarur et al. (2016)
Sea of Galilee
Clumped isotope analysis of Melanopsis shells
**-3**

Temperature:
Mean Annual


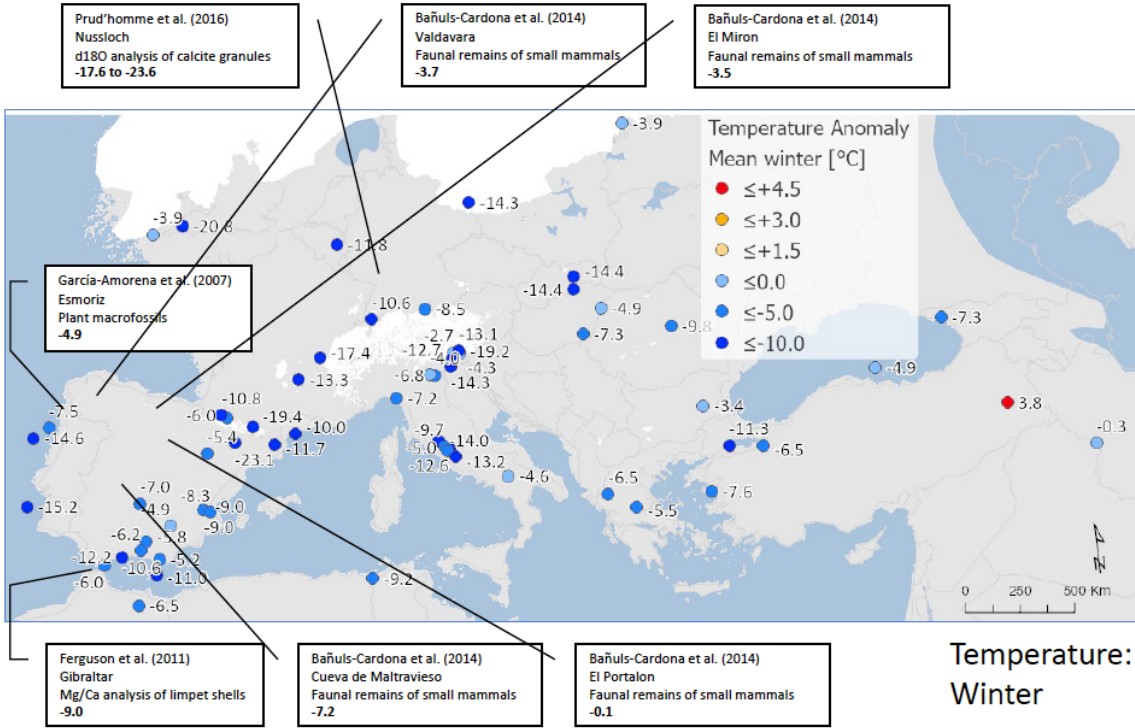

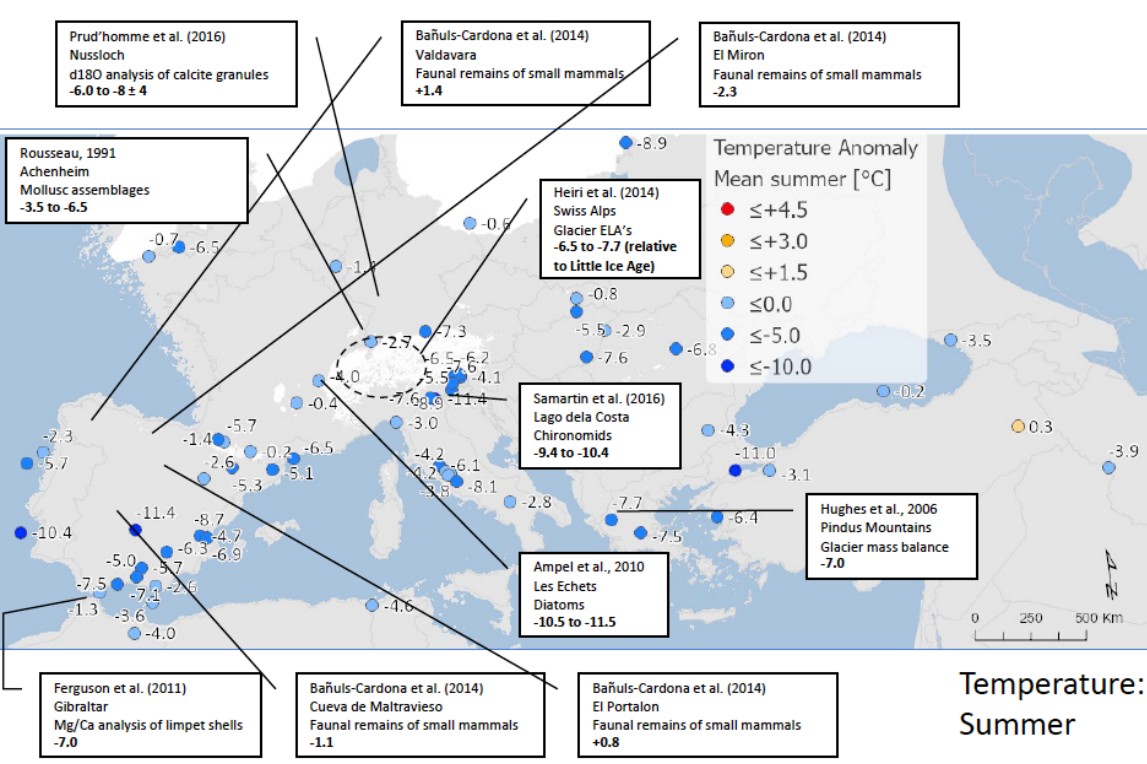



Figure A4. Maps of pollen-based MAT reconstructions for LGM annual, winter and summer
temperature anomalies (as shown in figure 10), shown together with the results of other
published studies. Continentality represents the difference in temperature between summer
and winter, with positive anomalies indicating an increase in the temperature difference
between summer and winter. All values are expressed as anomalies compared with the
present day unless otherwise indicated.

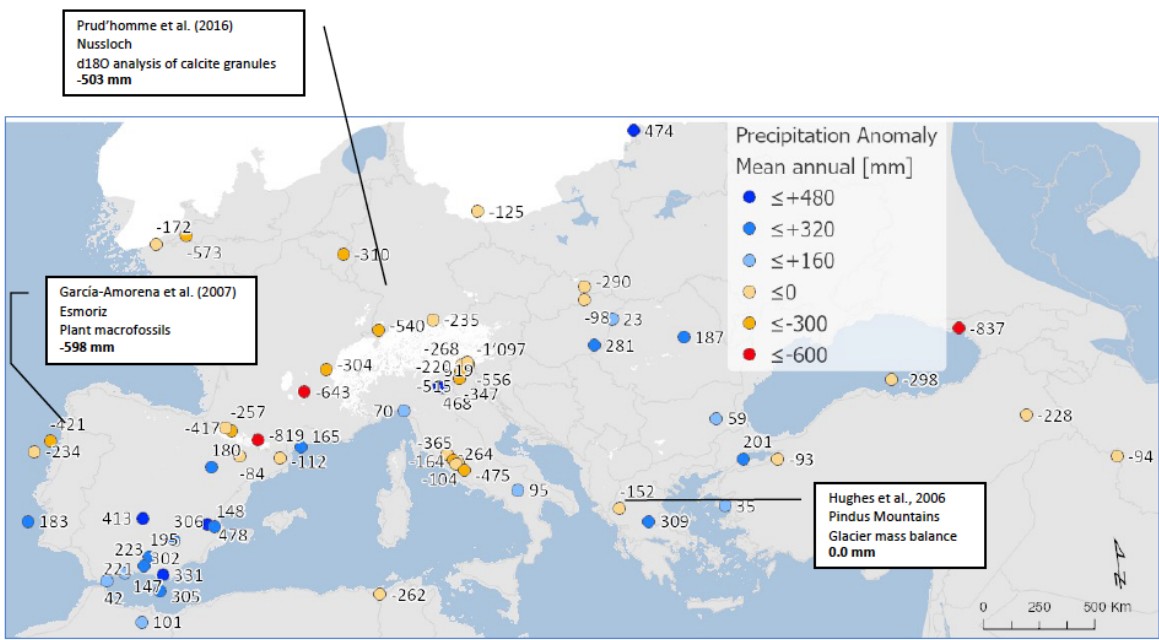

Precipitation:
Mean Annual

Figure A5. Maps of pollen-based MAT reconstructions for LGM annual precipitation
anomalies (as shown in figure 12), shown together with the results of other published studies.
All values are expressed as anomalies compared with the present day.
