# Peer review of "Middle East during the Last Glacial Maximum"

_Climate of the Past, 2022_

## Referee Comment (RC2)

Thank you for the opportunity to review for Climate of the Past the manuscript entitled "The climate and vegetation of Europe, North Africa and the Middle East during the Last Glacial Maximum (21,000 years BP) based on pollen data" by Davis B. and coauthors.

The Last Glacial Maximum or LGM is a key past period selected since last decades to provide proxies-model comparisons in the framework of PMIP and following projects. This paper follows different papers which aimed to reconstruct quantitatively the climate conditions in Europe during the LGM from pollen data (eg Peyron et al., 1998, Tarasov et al. 2000, Guiot et al., 1999, Jost et al., 2005; Wu et al., 2007; Bartlein et al., 2011, Cleator et al., 2020). All these papers are based on the same fossil pollen dataset, but on different methods : (1) PFT method with ANN calibration (Peyron et al., 1998; Tarasov et al., 2000, Jost et al., 2005, Bartlein et al., 2011), and (2) methods which take into account the effect of the CO2 on plants as Inverse Modelling (Guiot et al., 1999; Wu et al. 2007) or recent algorithms developed by Cleator et al. (2020) and used in Pini et al., (2022) and Wei et al., (2021).

The objective of this paper is to update these studies and to propose a new synthesis on the climate in Europe during the LGM based on the modern analogue method (MAT). The positive point of this paper is that it's based on new pollen datasets. It is based on the MPD dataset (8000 modern pollen spectra instead of 800 to 1500 for previous studies), and it also proposes a new fossil dataset (63 instead of 18), which allows to strongly increase the spatial resolution of the results and to better understand the climatic patterns during the LGM from proxies data.

For these two reasons, I think that the paper presents interesting findings in terms of results to be published in Climate of the Past ; however, I also think that it cannot be published in its current version.

My first point concerns the choice of the method to reconstruct LGM climate changes. You have selected the MAT: why? It's a key point because previous studies have evidenced that it's not easy to find reliable modern analogues for the LGM vegetation, and that assuming past CO2 equivalent to modern one may induce biases in climate reconstruction (Guiot et al., 2009; Prentice et al., 2017). So why do you use the MAT on your data instead of the IM or the Cleator method which are the only methods to take into account the CO2 changes? I like the MAT for Holocene but I think that the MAT, as used here for LGM, is not appropriate for several reasons.

The first one is that the vegetation of the LGM is mostly steppe, and there is, with the MAT, a possible confusion between warm and cold steppes, which can lead to a bias to the reconstruction of too warm climate conditions for the LGM. There is a method that take into account this bias by distinguishing warm and cold steppes (Tarasov et al 1998, JQS) , but this is not what was used here. Here, the MAT is applied directly to the PFT scores of the undifferentiated steppe biome. I think that the fact of not differentiating the steppes can lead to an important bias in the results obtained in this study. You should add a figure (with the basic statistical tests R2, RMSE) plotting the climate parameters estimated/observed for the modern samples of the steppe biome and see if we have no deviation. We should also add a figure (supp mat ?) with the location of the modern analogues chosen for each of the fossil spectra classified in steppe.

The second one is that CO2 is not really taken into account in this paper. You compare the results obtained here with the MAT with the already published results of Wu et al (2007) based on IM developed by Guiot et al. 2000 (Guiot et al., 2009). I consider that it is insufficient because the datasets used (modern and fossil) are different and therefore hardly comparable. It would be necessary to compare your results with the recent results of Cleator et al (2020, values are available in supplementary mat). The solution that I recommend is to apply the Inverse modelling developed by Guiot on your new datasets presented here, or the algorithm developed by Cleator et al (2020), cf in Pini et al (2022).

If this is not possible, one of the solutions would be to apply to your data a multi-methods approach - WA-PLS or machine learning methods (Random Forest, Boosted Regression Trees)- as often applied now (Salonen et al.,2014; Brewer et al., 2008; Peyron et al., 2013; Robles et al. 2022 ...) to be sure of your results.

Once concerns are addressed, I feel the manuscript will be much closer to being an outstanding contribution to knowledge in this time period, and a key paper to validate model outputs.

Minor points :

Abstract, line 28 "Previous pollen-based climate reconstructions based on MAT show…": which ones? The MAT has not been often used to reconstruct LGM climate, and the PFTs method cf Peyron et al., Tarasov et al and others references IS not a variant of the MAT, so correct it.

Introduction

     -lines 40 to 52 : more references are needed

     -lines 62-34: "the pollen-based reconstructions that show the greatest disagreement with climate models have themselves been criticized for not considering the possible effect of low atmospheric CO2 on the physiological relationship between plants and climate (Ramstein et al., 2007)". The significant bias of CO2 in climate reconstructions for glacial periods must be further explained here, as well as the developed methods that take it into account: inverse modelling by Guiot et al., 2000, 2009; the recent algorithm of Prentice et al., 2017 and Cleator et al., 2020.

     -line 65 Inverse modelling, the ref is missing; please add Guiot et al, 2000 (Guiot, J., et al Inverse vegetation modeling by Monte Carlo samgpling to reconstruct paleoclimate under changed precipitation seasonality and CO2 conditions: application to glacial climate in Mediterranean region, Ecol. Model., 1, 119–140, 2000.) and Guiot et al 2009.

     -line 100-103 the chronology of the LGM needs to be further explained here as the LGM time window is very close to the Heinrich stadial 1 (17.7 ka) and 2 (23.7 ka).

     -line 126: other proxies: which ones? Speleothems?

Methods

-line 177 "more recent studies": which ones?,

"although the exact record (EPD site #Entity) ": ???

-lines 178-180 " We estimate that we have excluded 16 of the 17 European sites used by Binney et al. (2017) , 5 of the 6 European sites used by Allen et al. (2010), 28 of the 33 sites used by Cao et al. (2019) and 27 of the 71 sites used by Kaplan et al. (2016)". So finally, how large is your dataset?  How many marine cores ?  How is the spatial coverage of these new sites?

-line 194 « The count of *Larix* was amplified by a factor of 10 due to its low pollen representation (Binney et al., 2017)": why only *Larix*? Other taxa are under or over represented: how do you manage that?

-line 213" we did not apply this additional procedure and present only the merged steppe biome": I disagree with that (see my major point) because a possible confusion between warm and cold steppes can lead to a bias in the climate reconstruction to too warm climate conditions for the LGM.

-line 220 "to match fossil samples with modern calibration pollen samples": the MAT is an assemblage approach which require no statistical calibration, so correct it (the modern pollen samples dataset is not a calibration dataset as it's the case for the WAPLS for example).

-line 221-223 "This is a similar approach to that used by Peyron et al. (1998) and Jost et al. (2005) who also applied pollen PFT scores to reconstruct LGM climate from pollen data, but who used a neural network technique which is a variant of the standard MAT (Chevalier et al., 2020)". I disagree with that, there is a confusion here in the principle of each method. The Artificial neural networks used by peyron et al  and others studies IS NOT a variant of the MAT. It's a method close to machine-learning methods, with a real calibration dataset and not easy to check because similar to a black box; in contrast the MAT is very simple, based on an dissimilarity calculation. The only common point is that both methods use PFTs scores to overcome problems associated with the lack of modern analogue but that is all.

-line 242 "The size and distribution of the modern training set in climate and vegetation space is important": yes, I strongly agree with that, the role of the modern dataset is a key one see papers of Turner et al., 2021; Salonen et al; Dugerdil et al., 2021 for example. I think that the differences in the different climate reconstructions evidenced here are mainly due to the size of the modern dataset.

-line 259 "It was therefore decided not to apply this filter", so how to you take into account the autocorrelation in your data?

-line 263 A part on the climate parameters reconstructed here is lacking, as statistical tests to be sure that these climate parameters are not autocorrelated; how is calculated the error bars?

-lines 267-272: refs are missing;

-line 312 "Similarly, quantitative climate methods have been applied to individual marine pollen records (Combourieu Nebout et al., 2009; Fletcher et al., 2010)": some key references are missing, as the MF Sanchez Goni team.

-line 331 "In this study we have taken the closest point on land as the modern climate for the calculation of anomalies": better to take a regional temperature range

-lines 337-347 "we did not adjust the pollen assemblage for the over-representation of Pinus in the marine pollen samples" This poses the problem of *Pinus* transport over very long distances in open environments as the LGM vegetation; this is particularly true for marine cores but it is also true for some terrestrial sites. So the question of excluding or keeping *Pinus* needs to be more investigated and tested may be on a site-by-site basis.

-line 363 "such as [site #3] and [site #58]" ; better to give the name of the sites

-lines 377-380 "The main arboreal biomes found at the LGM include Taiga (TAIG), Cool Mixed Forest (COMX), Cool Conifer Forest (COCO) and Xerophytic Scrub (XERO), with just a single occurrence of Cold Mixed Forest (COMX) and Warm Mixed Forest (WAMX). We do not record any Temperate Deciduous Forest (TEDE), Tundra (TUND) or Desert (DESE) biomes at any site at the LGM." Could you explain more the location of the differents biome patterns?

- in the text, many taxa are not in italic: please correct it

- lines 441-443 "The first test was to compare our MAT results with previous pollen-climate reconstructions based on the same LGM sites but using different methods. These previous reconstructions include the neural-network methodology of Peyron et al. (1998) and Jost et al. (2005)". I don't agree, it's not a validation test: not the same method, not the same surface datasets, so we cannot really compare the results. Moreover, the LGM spectra used in previous studies and here are probably not the same, that too can bias the results. OK for me in the discussion but not in this part as a validation test. Same for Wu et al, 2007.

-lines 443-444 "the neural-network methodology of Peyron et al. (1998) and Jost et al. (2005) which we call MAT-NN, as well as the Inverse Modelling approach by Wu et al. (2007) which we call INV." First, the neural networks methodology of peyron et al. is NOT a MAT method, so you cannot call it MAT-NN, it's a non-sense. Second, could you use the name of the method given in the reference papers? Please check, I guess it's the PFT method for Peyron et al and I.M. for Wu et al. which are correct.

-line 472 "We compare the chironomid record with our MAT reconstruction…": you don't compare the chironomid record, you compare the temperature inferred from the chironomid record, please correct it

Samartin et al (2016) not Samaratin et al (2016)

-lines 510-512 « The second consequence of lower seas levels is that terrestrial pollen sites were located further from the moderating effect of the ocean than they are today, resulting in a localised modification of the climate experienced by the site irrespective of regional or global changes.": a ref is lacking

-lines 531-538: "In terms of regional climate, the major ice sheets would have provided significant barriers to westerly atmospheric circulation, or even north-south circulation in the case of the Alps and Pyrenees. As well as representing a physical obstruction, the thermodynamic response of the atmosphere to these high, cold obstructions would have been to encourage the formation of areas of semi-permanent high pressure, similar to those found today for instance over the Greenland ice sheet. In addition, the Laurentide ice sheet located over North America would have generated downstream effects over Europe. These physical and thermodynamic effects would have affected the direction of storm tracks, as well as more local climatic effects commonly associated with ice sheets such as strong katabatic winds.": refs are lacking

-Line 563: "despite arboreal pollen forming 70-80% of the pollen assemblage": a significant part of the arboreal pollen is due to *Pinus* which is clearly overestimated in LGM pollen assemblages due to long distance transport in open areas as during the LGM.

-lines 615-616: "expected, areas of forest reconstruct similar or increased precipitation compared to today, and areas of steppe indicate deceased precipitation (see next section)." The $CO_2$ effect on climate reconstruction (see recent papers by Cleator et al. and Prentice et al) is not discussed, please add a part on this point.

-line 618 correct "archaezoological"

-line 669 PMIP = Paleoclimate Modelling Intercomparison Project, not "Palaeo-model Intercomparison Project", correct it; many key refs on PMIP project are missing: Jost et al., 2005; Tarasov et al …

-line 372: "suffer from the same problems of dating control, unclear provenance and a potentially limited taxa assemblages." I don't agree with that, you kept a lot of them for your study.

- line 677: "and the Neural Networks method which is a version of MAT (MAT-NN) ": the method developed by Peyron et al and Tarasov et al is named the PFT method and IS NOT a version of the MAT. It's a method based on Artificial neural networks close to machine-learning methods, with a real calibration dataset and similar to a black box; both methods use PFTs scores to overcome problems associated with the lack of modern analogue but that is all.

-lines 678-690: see my major concern; I think that the fact of not differentiating the steppes can lead to the warm temperatures reconstructed here with the MAT; please check.

-line 721: diatom not Diatom

-line 730 check  "Hughes et al (Hughes et al., 2006)"

-line 755  "19.1 ºC" or -19.1 ºC ?

-line 763 "This compares with -7.2 ºC for our 63 pollen sites": not sure it makes sense to calculate the mean for 63 sites given the regional climate patterns

-lines 778-784: Good to add a comparsion with the brGDDTs temperature record from Padul (Rodrigo-Gámiz et al., 2022).

-line 806: I think a part on the comparison of these results with LGM model outputs is lacking.

-lines 856-857 "Nevertheless, one of the most consistent signals in our dataset is for an increase in summer precipitation over many areas of Southern Europe and the Mediterranean". In south Spain, the reconstructed biomes is steppe or xerophytic, with a lot of *Artemisia* and chenopodiaceae: these taxa are characteristic of dry environments (semi-desert), so how do you explain the wetter than today conditions reconstructed?

-check your reference list : Allen et al., 2008 a and b, two refs for Peyron et al 1998 ..

I realize the authors may find my comments difficult to approach, but I sincerely hope they accept them as well-intentioned guidance. It should not be difficult to address them. Once concerns are addressed, I feel the manuscript will be much closer to being an outstanding contribution to knowledge in this time period.

---

## Author Comment (AC2)

We would like to thank the reviewers for the time and effort that they have put into their reviews of our manuscript. Their comments and suggested changes have greatly improved the manuscript.

We respond to the reviewers comments line-by-line below (their comments are highlighted in yellow). We include both our response and details of the relevant action that we have taken. Additions to the original text are also highlighted in the revised manuscript. We start with reviewer 2 who had two major comments:

**Reviewer 2**

*My first point concerns the choice of the method to reconstruct LGM climate changes. You have selected the MAT: why? It's a key point because previous studies have evidenced that it's not easy to find reliable modern analogues for the LGM vegetation, and that assuming past CO2 equivalent to modern one may induce biases in climate reconstruction (Guiot et al., 2009; Prentice et al., 2017). So why do you use the MAT on your data instead of the IM or the Cleator method which are the only methods to take into account the CO2 changes? I like the MAT for Holocene but I think that the MAT, as used here for LGM, is not appropriate for several reasons.*

*1) The first one is that the vegetation of the LGM is mostly steppe, and there is, with the MAT, a possible confusion between warm and cold steppes, which can lead to a bias to the reconstruction of too warm climate conditions for the LGM. There is a method that take into account this bias by distinguishing warm and cold steppes (Tarasov et al 1998, JQS) , but this is not what was used here. Here, the MAT is applied directly to the PFT scores of the undifferentiated steppe biome. I think that the fact of not differentiating the steppes can lead to an important bias in the results obtained in this study. You should add a figure (with the basic statistical tests R2, RMSE) plotting the climate parameters estimated/observed for the modern samples of the steppe biome and see if we have no deviation. We should also add a figure (supp mat ?) with the location of the modern analogues chosen for each of the fossil spectra classified in steppe.*

**Response:** The reviewer suggests that our MAT-based reconstruction does not take into account the difference between warm and cold steppe. This is not true. As stated in the methods section we reconstruct climate using the PFT classification of Tarasov et al 1998 and Peyron et al. 1998. This means that our analysis includes both the warm grass steppe (wgs) and cool grass steppe (cgs) PFT's that are used by both authors to distinguish between warm and cold steppe biomes.

The reviewer mentions the Tarasov et al (1998) method to better differentiate between cold and warm steppe pollen biomes using the Prentice et al (1996) pollen biomisation algorithm. The method of Tarasov et al (1998) is designed to overcome a problem at the biome level, and not the PFT level used in our MAT method. It is applied after the calculation of the PFT scores. The method essentially works by using the presence or absence of thermophilous arboreal taxa to re-assign steppe and desert PFT's into ONLY either warm or cold varieties. As such it artificially exaggerates the difference between the PFT's that make up the warm or cold steppe biomes. This artificial separation acts to

increase the biome score of either the warm or cold steppe biome relative to each other, as well as relative to other competing biomes, making it more likely that one of the steppe biomes will become the dominant biome (the dominant biome is the one with the highest cumulative PFT score of all the PFT's within that biome). However, using these re-assigned PFT scores for MAT makes no sense because it undermines the basic assumption that PFT scores (and the underlying vegetation that they represent) vary in a consistent and uniform manner with climate.

At the reviewer's suggestion we have undertaken an analysis of the performance of our MAT transfer function based on a sub-set of modern pollen samples from steppe environments. For this, we selected 1588 samples from the Eurasian Modern Pollen Database (EMPD) (Davis et al 2020) that were classified as belonging to the steppe pollen biome (the pollen biome is included in the metadata for each sample in the database). The results show little difference between the steppe samples and the performance using the complete dataset. They do not indicate any specific weakness as suggested by the reviewer.

| | All surface samples | | Steppe only | |
|---|---|---|---|---|
| | RMSE | R2 | RMSE | R2 |
| TANN | 2.28 | 0.9 | 2.51 | 0.87 |
| TDJF | 3.35 | 0.91 | 3.26 | 0.88 |
| TJJA | 2.21 | 0.81 | 2.49 | 0.82 |
| PANN | 224.94 | 0.69 | 185.7 | 0.71 |
| PDJF | 78.51 | 0.69 | 66.5 | 0.66 |
| PJJA | 52.49 | 0.75 | 43.8 | 0.79 |

The reviewer mentions problems with finding modern analogues for LGM pollen samples, especially steppe. This was often mentioned in early studies but this was probably because they used particularly small modern surface sample datasets, as well as both modern and fossil datasets that were often digitized or from secondary sources that did not include the full pollen assemblage. We show in the paper, as others have shown before us (Pini et al 2021, Magyari et al 2014a), that there are in fact many available analogues for LGM pollen samples in the new bigger and more spatially extensive modern surface sample datasets such as the EMPD2. We also use PFT scores rather than individual taxa which increases the potential to find modern analogues. Based on the square-chord distance measurement we did not find any fossil sample where we could not find 6 close modern analogues in the EMPD2 (a 'close' analogue being defined as a chord distance <0.3, as suggested by Huntley 1990).

The reviewer suggests providing maps showing the location of the analogues. This would require generating a lot of maps (the dataset includes 524 samples from 63 sites) which might not be very helpful. We already intended to include the list of 6 analogues for each fossil sample in the supplementary files, which would allow anyone to investigate the location and nature of the analogues in detail. Instead, we have included a table in the appendix (Table A4) which shows the main ecoregion where most of the analogues originated for each site. This probably represents a more accessible summary of the modern location and vegetation landscape of the analogues being used.

**Action:** The analysis of steppe samples is now mentioned in the text (lines 550-555) and shown in the appendix table A3. A summary table is provided in the appendix (A4) showing the ecoregion from where most of the modern analogues originated for each site. This is also mentioned in the text (lines 683-685).

==2) The second one is that CO2 is not really taken into account in this paper. **2.1)** You compare the results obtained here with the MAT with the already published results of Wu et al (2007) based on IM developed by Guiot et al. 2000 (Guiot et al., 2009). I consider that it is insufficient because the datasets used (modern and fossil) are different and therefore hardly comparable.==

**Response:** We justify the use of the MAT method for the LGM by showing that our MAT reconstruction produces results that are essentially indistinguishable from an Inverse Modelling (IM) reconstruction by Wu et al 2007 from the same fossil dataset of 10 sites. We cannot be absolutely sure we used exactly the same fossil samples as Wu et al, since the dataset used by Wu et al is poorly documented, but we can say that it is from the same site, the same pollen record, from the same time period, and has the same reconstructed pollen-biome. We therefore consider the fossil datasets and therefore the results, to be comparable. We are not sure what the reviewer means by suggesting that the MAT and IM modern pollen datasets are not comparable, since the IM method does not require use of a modern pollen dataset.

**Action:** None

==**2.2)** It would be necessary to compare your results with the recent results of Cleator et al (2020, values are available in supplementary mat).==

**Response:** As far as we can see, the results of both Cleator et al. 2020a and 2020b are only available as a gridded dataset in the supplementary material to these papers. Unfortunately, the authors do not provide the site data that would allow us to make a comparison.

**Action:** None

==**2.3)** The solution that I recommend is to apply the Inverse modelling developed by Guiot on your new datasets presented here, or the algorithm developed by Cleator et al (2020), cf in Pini et al (2022).==

**Response:** As we mentioned in reply to reviewer Response **2.1**, we show in our analysis that our MAT method produces almost exactly the same results as the inverse modelling method for essentially the same samples from the same 10 sites (Fig 7). It is not therefore clear why undertaking an Inverse modelling analysis, as the reviewer proposes, would substantially change these results.

Given the reviewers concerns about the CO2 problem, we think that it is important here to place the problem in perspective. Firstly, it is now generally considered that glacial-level atmospheric CO2 concentrations mainly affects inferred precipitation or moisture balance in pollen-based paleoclimate reconstructions as opposed to temperature. This is evidenced by

the fact that in both of the papers by Cleator et al. (2020) and Pini et al. (2022) cited by the reviewer, the CO2 correction algorithm is applied only to precipitation reconstructions, since they do not consider the the effects of low CO2 to be sufficiently important to apply to temperature variables "*Low [CO2] will not impact reconstructions of temperature, but has a large impact on moisture-related variables*" (Pini et al., 2022).

Secondly, it is important to ask "IF there really is a problem with reconstructing precipitation, exactly how big a problem is it?", especially in relation to other uncertainties.

As a demonstration of this point we show below the effects of the Cleator et al (2020) correction algorithm as applied by Pini et al (2022) on a Modern Analogue Technique (MAT) reconstruction of mean annual precipitation (Pann) at the Lake Fimon site in Northern Italy. Here we include the authors' uncertainty bounds of their MAT reconstruction because it provides perspective when viewing the CO2 correction (Pini et al 2022, figure 6F and 7G). As can be seen in the figure below, the correction '**b**' during the LGM (21K +/-2k highlighted) is in fact very minor (roughly ~22mm on average for the 23 samples over this time period). This compares with the uncertainties '**a**' of the MAT reconstruction itself (+/-200mm) which are approximately an order of magnitude greater. In other words, Pini et al 2022 show pretty much the same result as we do, that any CO2 effect is essentially indistinguishable from the overall uncertainties of the MAT reconstruction.

[Figure]

a wMAT Pann uncertainty ± 200 mm
b Correction for CO2 effect around LGM <30mm

In addition, Pini et al (2022) do not provide uncertainties for the CO2 correction itself, and this does not appear to be discussed in any of the Cleator et al papers or is included in the code (as shown in Wei et al 2021 supplementary). Pini et al 2022 explain that the correction algorithm is based on inputs of growing season temperature, cloud cover and insolation. We must therefore assume that at least the first two of these variables are themselves estimates with their own uncertainties. Pini et al 2022 say that they undertook a sensitivity analysis of the role of these different variables but unfortunately, they do not show the results of this in their paper, and only mention that cloud cover appeared to explain most of the variance during the glacial period. In any case it seems surprising that the size of the CO2 correction throughout the record does not appear to be closely related to the actual atmospheric CO2 concentration, with a correction close to zero for a number of samples around the LGM period when CO2 was at its lowest compared to the present day.

**Action:** We have amended the text to include reference to the Pini et al (2022) paper in support of our conclusions about the $CO_2$ effect (lines 852-869).

*2.4) If this is not possible, one of the solutions would be to apply to your data a multi-methods approach - WA-PLS or machine learning methods (Random Forest, Boosted Regression Trees)- as often applied now (Salonen et al.,2014; Brewer et al., 2008; Peyron et al., 2013; Robles et al. 2022 ...) to be sure of your results.*

**Response:** We are not sure why any of these methods would be better at addressing the $CO_2$ problem, since all of them use a modern calibration dataset and therefore rely on the same assumptions about modern analogues as MAT. Also, we are not sure how a multi-method approach helps one to be 'sure' of the results? This would suggest that some kind of combination of methods is better than one, although none of the publications cited appear to provide a clear scientific justification for this more complex approach. The assumption appears to be that if more methods agree then the reconstruction is somehow more robust, but they may just as well be agreeing on being wrong. In some cases some of these pollen-climate methods may be quite inappropriate. For instance, machine learning methods can suffer from over-tuning and by 'black boxing' provide poor analytical insight, while WA-PLS only really works well when the fossil samples and their modern analogues are regionally well constrained, something that can quickly breakdown during the LGM when the best analogues are to be found on the other side of continents (e.g. Siberia, Mongolia).

Instead we prefer to evaluate our reconstruction in the light of the $CO_2$ problem in 3 key ways 1) we compare on a site-by-site basis with a pollen-based INV method (ie the results presented by the Wu et al 2008 study) that is designed specifically to account for the $CO_2$ effect, we 2) compare on a sample-by-sample basis with a chironomid-based method which represents an entirely different proxy (fauna not flora), and 3) we undertake an extensive discussion that compares our reconstruction with a wide variety of records from a wide variety of proxies from across Europe.

**Action:** We emphasise all of the above points in our revised manuscript text
* * *
Minor points :

*3) Abstract, line 28 "Previous pollen-based climate reconstructions based on MAT show...": which ones? The MAT has not been often used to reconstruct LGM climate, and the PFTs method cf Peyron et al., Tarasov et al and others references IS not a variant of the MAT, so correct it.*

**Response:** The comparison of ANN and MAT is really in reference to their common use of a modern surface sample dataset, and therefore both are dependent on 'modern analogues'. In this sense ANN is more similar to MAT than Inverse Modelling. However, we agree it is confusing, so have corrected this throughout.

**Action:** MAT-NN has been changed to ANN throughout. Abstract, line 28 has been changed from "*Previous pollen-based climate reconstructions based on MAT show a much colder and*

*drier climate for the LGM than both Inverse Modelling and climate model simulations*" to
"*Previous pollen-based climate reconstructions **using modern pollen calibration datasets**
show a much colder and drier climate for the LGM than both Inverse Modelling and climate
model simulations*" (line 33).

*4) Introduction*
*-lines 40 to 52 : more references are needed*

**Response:** Ok

**Action:** The following references have been added- Ehlers et al. 2011, Arslanov et al. 2007,
Lehmkuhl et al. 2021, Grichuk 1992 (lines 51-62)

*5) -lines 62-34: "the pollen-based reconstructions that show the greatest disagreement with
climate models have themselves been criticized for not considering the possible effect of low
atmospheric CO2 on the physiological relationship between plants and climate (Ramstein et
al., 2007)". The significant bias of CO2 in climate reconstructions for glacial periods must be
further explained here, as well as the developed methods that take it into account: inverse
modelling by Guiot et al., 2000, 2009; the recent algorithm of Prentice et al., 2017 and
Cleator et al., 2020.*

**Response:** This is similar to a comment shared by reviewer 1

**Action:** The following section has been added to the introduction: "***Methods that use
modern pollen samples for calibration purposes are based on the assumption that the
relationship between vegetation and climate remains the same through time, and that
this is independent of change in CO2 concentration. Studies have shown however that
plant growth processes and plant resilience are sensitive to CO2 concentration, and
particularly water-use efficiency which would make plants more drought sensitive in low
CO2 environments (Cowling & Sykes 1999). Atmospheric CO2 during the LGM was around
190 ppm, some 100 ppm lower than the pre-industrial period, and 200 ppm lower than the
levels experienced in the last 50 years. Concerns about the effects of lower CO2 during the
LGM has directly led to the development of pollen-climate reconstruction methods that
can take account of CO2 effects, either through use of a process-based vegetation model
run in inverse mode (Guiot et al. 2000, Guiot et al. 2009), or through the use of a
correction algorithm (Prentice et al. 2017).***" (lines 77-89)

*6) -line 65 Inverse modelling, the ref is missing; please add Guiot et al, 2000 (Guiot, J., et al
Inverse vegetation modeling by Monte Carlo samgpling to reconstruct paleoclimate under
changed precipitation seasonality and CO2 conditions: application to glacial climate in
Mediterranean region, Ecol. Model., 1, 119–140, 2000.) and Guiot et al 2009.*

**Response:** OK.

**Action:** The two references (Guiot et al. 2000, 2009) have been added, see response to
previous comment. (lines 87-88)

*-line 100-103 the chronology of the LGM needs to be further explained here as the LGM time window is very close to the Heinrich stadial 1 (17.7 ka) and 2 (23.7 ka).*

**Response:** Ok

**Action:** The text has been changed from "*This is particularly important because the 21 Å} 2.0 ka time slice commonly used to represent the LGM period in PMIP data-model comparisons and other synthesis studies (MARGO members, 2009; Bartlein et al., 2011) occurs immediately after the glacial maxima in the Alps, which occurs around 26-23 ka (Heiri et al., 2014; Spotl et al., 2021), and is therefore likely to be represented by a different vegetation and climate.*" to "*This is particularly important because the 21 ± 2.0 ka time slice commonly used to represent the LGM period in PMIP data-model comparisons and other synthesis studies (MARGO members, 2009; Bartlein et al., 2011) occurs immediately after the glacial maxima in the Alps **around 26-23 ka (Heiri et al., 2014; Spötl et al., 2021) and Heinrich stadial HS-2 (24.3-26.5), whilst also being closely followed by Heinrich stadial HS-1 (15.6-18.0 ka) (Sanchez-Goñi & Harrison, 2010. These closely associated time periods can therefore be expected to represent both a different vegetation and climate than the LGM itself.***" (lines 135-138)

*-line 126: other proxies: which ones? Speleothems?*

**Response:** It seems unnecessary and distracting to list in the introduction all the proxies that we mention in the discussion. Such a list would include for example chironomids, oxygen isotopes from molluscs shells and soil calcites, macrofossils, mammal bone assemblages, tree leaf lipids, sedimentary lipids, molluscs, glacial modelling, diatoms, alkenones, foraminifera, Mg/Ca etc

**Action:** None

*Methods*
*-line 177 "more recent studies": which ones?,*
*"although the exact record (EPD site #Entity) ": ???*

**Response:** The answer is contained in the sentence following the one that the reviewer refers to where we list the studies and estimate the number of sites/entities involved.

**Action:** The text has been changed from "*(EPD site #Entity)*" to "*(EPD Entity number)*" (line 228). The 'more recent studies' are cited in lines 229-231

*-lines 178-180 " We estimate that we have excluded 16 of the 17 European sites used by Binney et al. (2017) , 5 of the 6 European sites used by Allen et al. (2010), 28 of the 33 sites used by Cao et al. (2019) and 27 of the 71 sites used by Kaplan et al. (2016)". So finally, how large is your dataset? How many marine cores ? How is the spatial coverage of these new sites?*

**Response:** OK.

**Action:** *So finally, how large is your dataset?* The text has been changed from *"The distribution of sites included in our study"* to *"The distribution of **the 63** sites included in our study"* (line 182), *How many marine cores ?* The has been changed from *"For completeness, we also include marine records"* to *"For completeness, we also include **7** marine records"* (line 196) *How is the spatial coverage of these new sites?* We have added the following text ***"Nevertheless, our dataset includes sites from this region, as well as North Africa and eastern Central Europe through to Iran, although most sites are located in an arc across eastern Spain, the Alps, and Italy**."* (lines 186-188)

-line 194 « The count of *Larix* was amplified by a factor of 10 due to its low pollen representation (Binney et al., 2017)": why only *Larix*? Other taxa are under or over represented: how do you manage that?

**Response:** We apply the correction for Larix in common with many other authors. It is an important forest forming boreal tree indicative of a particular climate, and it has a particularly low pollen dispersal compared to other trees that occur in the LGM dataset.

**Action:** As well as Binney et al 2017, we have now added the following authors who have also applied a similar correction for Larix pollen: **Edwards et al. 2000, Bigelow et al. 2003, Tarasov et al. 1998, 2000, 2013** (lines 248-249)

-line 213" we did not apply this additional procedure and present only the merged steppe biome": I disagree with that (see my major point) because a possible confusion between warm and cold steppes can lead to a bias in the climate reconstruction to too warm climate conditions for the LGM.

**Response:** Please see the response to the earlier comment 1. The climate reconstruction is based on pft's, not biomes, therefore it differentiates between warm and cold steppe.

**Action:** None

-line 220 "to match fossil samples with modern calibration pollen samples": the MAT is an assemblage approach which require no statistical calibration, so correct it (the modern pollen samples dataset is not a calibration dataset as it's the case for the WAPLS for example).

**Response:** The term calibration is widely used with respect to MAT in the literature. See Simpson (2007) *"The modern analogue technique, described below, is an inverse multivariate **calibration** approach."* Or Juggins & Birks (2012) for instance figure 14.3, part of which is shown below.

[Figure]

[Figure]

**Fig. 14.3** Conceptual diagram illustrating the different approaches to multivariate calibration. Sp are biological taxa in the modern training-set and Env is the environmental variable of interest in the modern data. C are components

Simpson (2007) *Analogue Methods in Palaeoecology: Using the analogue Package* doi: 10.18637/jss.v022.i02
Juggins & Birks (2012) doi:10.1007/978-94-007-2745-8_14,

**Action:** None.

*-line 221-223 "This is a similar approach to that used by Peyron et al. (1998) and Jost et al. (2005) who also applied pollen PFT scores to reconstruct LGM climate from pollen data, but who used a neural network technique which is a variant of the standard MAT (Chevalier et al., 2020)". I disagree with that, there is a confusion here in the principle of each method. The Artificial neural networks used by peyron et al and others studies IS NOT a variant of the MAT. It's a method close to machine-learning methods, with a real calibration dataset and not easy to check because similar to a black box; in contrast the MAT is very simple, based on an dissimilarity calculation. The only common point is that both methods use PFTs scores to overcome problems associated with the lack of modern analogue but that is all.*

**Response:** Already agreed, see answer to earlier comment 3.

**Action:** See answer to comment 3.

*-line 242 "The size and distribution of the modern training set in climate and vegetation space is important": yes, I strongly agree with that, the role of the modern dataset is a key one see papers of Turner et al., 2021; Salonen et al; Dugerdil et al., 2021 for example. I think that the differences in the different climate reconstructions evidenced here are mainly due to the size of the modern dataset.*

**Response:** Agreed, we make the same point.

**Action:** The additional references have been added (lines 307-308)

*-line 259 "It was therefore decided not to apply this filter", so how to you take into account the autocorrelation in your data?*

**Response:** In common with many studies that use MAT, we do not take account of the effects of autocorrelation. We highlight to the reader the autocorrelation problem, and why we do not apply the h-block method developed to reduce the effects of autocorrelation, mainly because it creates as many problems as it solves.

**Action:** We explain our reasoning in lines 310-326.

*-line 263 A part on the climate parameters reconstructed here is lacking, as statistical tests to be sure that these climate parameters are not autocorrelated; how is calculated the error bars?*

**Response:** We are not quite sure what the reviewer is wanting here. Almost all of the common climate parameters used for reconstructions in the palaeo sciences are correlated to some extent with each other, it's the nature of the climate system. It is almost impossible to distinguish for instance whether a proxy, and especially a biological proxy, is responding to degree days, frost days, absolute minimum or mean monthly temperature etc. This is a problem inherent in climate reconstruction and is beyond the scope of this paper (see e.g. the discussion in section 3.4 of Chevalier et al 2021). We calculate error bars using the standard MAT method, we can add a description of this.

**Action:** The following text has been added (lines 328-332) to describe the calculation of uncertainties: "***Uncertainties for the pollen-climate reconstructions were calculated using the standard method for MAT (Juggins 2020), that is, as a function of the spread of the climates associated with the best modern pollen analogues used for each fossil sample. The closer the climates of the best modern pollen analogues (6 in the case of this study) then the smaller are the uncertainties assigned to the reconstructed climate of the fossil pollen sample***."

-lines 267-272: refs are missing;

**Response:** OK.

**Action:** The following references have been added: ***Davis 1963, Gaillard et al. 2010, Zanon et al. 2018*** (lines 343-344)

-line 312 "Similarly, quantitative climate methods have been applied to individual marine pollen records (Combourieu Nebout et al., 2009; Fletcher et al., 2010)": some key references are missing, as the MF Sanchez Goni team.

**Response:** Unfortunately, the reviewer does not provide any details of the key references that are supposed to be missing. While MF Sanchez Goni and her team have published many important papers, we cannot find any that involve quantitative reconstructions of climate based on pollen, which is the subject of the sentence.

**Action:** None

*-line 331 "In this study we have taken the closest point on land as the modern climate for the calculation of anomalies": better to take a regional temperature range*

**Response:** The reviewer suggests taking the climate of a region, but then the problem becomes, what region? There is no easy answer. The region represented by the source area is one of the key problems for interpreting pollen from marine cores, which is why we make specific reference to this problem in the text. Should it be weighted for distance from the core site? What about the land that is now covered by the sea but which would have contributed pollen when sea levels were lower during the LGM? Many studies show that pollen discharged by rivers close to marine core sites can be a significant source of pollen, should this also be taken into account? We agree that the option we have chosen is somewhat unsatisfactory, but then it would appear that every solution seems unsatisfactory. We suggest instead to include the modern climate values in the appendix so that the reader can adjust the anomalies as they see fit.

**Action:** We have included the modern climate values for all 63 sites in our dataset the appendix, Table A2. The problems (and advantages) associated with marine sites are discussed in lines 400-427.

*-lines 337-347 "we did not adjust the pollen assemblage for the over-representation of Pinus in the marine pollen samples" This poses the problem of Pinus transport over very long distances in open environments as the LGM vegetation; this is particularly true for marine cores but it is also true for some terrestrial sites. So the question of excluding or keeping Pinus needs to be more investigated and tested may be on a site-by-site basis.*

**Response**: Agreed, but the problem of over (or under) representation due to differential transport is a problem that is intrinsic to the science of palynology with no straight-forward answer. Fundamental to this is the fact that although the risk of under/over representation can be acknowledged, it is generally very difficult to detect and correct in any detail. One of the closest attempts at this can be seen in the use of MAT methods to reconstruct tree cover, which we apply in our study. This 'black box' approach at least makes some attempt to take into account the potential over-representation of *Pinus* in both terrestrial and marine environments, at least where this problem is also found in the modern analogue samples that are matched with the fossil samples.

From the point of view of the marine pollen records, we find it more appropriate to include rather than exclude *Pinus* because it is a key forest forming tree in the coastal regions close to the marine sites and to remove it completely would create an artificially arid assemblage that would certainly undermine the ability of the transfer function to reconstruct precipitation. Reconstructions of temperature would be less affected because *Pinus* is a generalist found in both hot and cold regions and so carries only a weak temperature signal compared to the rest of the assemblage. We can add this clarification to the text.

**Action:** The following text has been added to clarify the problem for the reader (lines 435-439) "***Removing Pinus from the assemblage would almost certainly create an artificially arid assemblage in these circumstances, undermining the ability of the transfer function to***

*reconstruct precipitation, although temperature would likely be less affected since Pinus is a generalist found in both hot and cold temperature regions*.".

-line 363 "such as [site #3] and [site #58]" ; better to give the name of the sites

**Response:** OK.

**Action:** Site names have been added throughout the text.

-lines 377-380 "The main arboreal biomes found at the LGM include Taiga (TAIG), Cool Mixed Forest (COMX), Cool Conifer Forest (COCO) and Xerophytic Scrub (XERO), with just a single occurrence of Cold Mixed Forest (COMX) and Warm Mixed Forest (WAMX). We do not record any Temperate Deciduous Forest (TEDE), Tundra (TUND) or Desert (DESE) biomes at any site at the LGM." Could you explain more the location of the differents biome patterns?

**Response:**  Ok

**Action:** The text has been changed from: "*The main arboreal biomes found at the LGM include Taiga (TAIG), Cool Mixed Forest (COMX), Cool Conifer Forest (COCO) and Xerophytic Scrub (XERO), with just a single occurrence of Cold Mixed Forest (COMX) and Warm Mixed Forest (WAMX). We do not record any Temperate Deciduous Forest (TEDE), Tundra (TUND) or Desert (DESE) biomes at any site at the LGM.*" to "***Of the main arboreal biomes, Taiga (TAIG) is the dominant biome at 3 sites at the eastern end of the Alpine ice sheet, as well as at a site just to the north in northern Germany and a site in Slovakia, while Cool Conifer Forest (COCO) is found at 1 site close to the Scandinavian ice sheet in Lithuania. Cool Mixed Forest (COMX) is found much more widely at 8 sites south of the Alps from south-west Iberia to Romania, with Xerophytic Scrub (XERO) occurring at 8 sites with a similar distribution but not as far east or west. Cold Mixed Forest (CLMX) occurs at just two sites in Georgia and the Alboran Sea at the far east and west of the study area, while Warm Mixed Forest (WAMX) is the dominant biome at just 1 site in Southern Spain. We do not record Temperate Deciduous Forest (TEDE), Tundra (TUND) or Desert (DESE) as the dominant biome at any site at the LGM, although they do occur as lesser biomes***." (lines 477-487)

- in the text, many taxa are not in italic: please correct it

**Response:** This was also mentioned by Reviewer 1

**Action:** Taxa names are now italicised (where appropriate) throughout the text.

- lines 441-443 "The first test was to compare our MAT results with previous pollen-climate reconstructions based on the same LGM sites but using different methods. These previous reconstructions include the neural-network methodology of Peyron et al. (1998) and Jost et al. (2005)". I don't agree, it's not a validation test: not the same method, not the same surface datasets, so we cannot really compare the results. Moreover, the LGM spectra used

in previous studies and here are probably not the same, that too can bias the results. OK for me in the discussion but not in this part as a validation test. Same for Wu et al, 2007.
-lines 443-444 "the neural-network methodology of Peyron et al. (1998) and Jost et al. (2005) which we call MAT-NN, as well as the Inverse Modelling approach by Wu et al. (2007) which we call INV." First, the neural networks methodology of peyron et al. is NOT a MAT method, so you cannot call it MAT-NN, it's a non-sense. Second, could you use the name of the method given in the reference papers? Please check, I guess it's the PFT method for Peyron et al and I.M. for Wu et al. which are correct.

**Response**:  The evaluation/comparison section has been moved to the discussion. We intended our method acronyms to be as self-explanatory as possible. 'PFT' is not the defining feature of the Peyron et al 1998 method, since the use of PFT scores can, and has, been used in other methods such as MAT. We therefore prefer to use the acronym 'ANN' for Artificial Neural Network (as used by Chevalier et al 2019).

**Action:**  The section mentioned by the reviewer has been moved to the discussion as they suggest. MAT-NN has been changed to ANN throughout.

*-line 472 "We compare the chironomid record with our MAT reconstruction…": you don't compare the chironomid record, you compare the temperature inferred from the chironomid record, please correct it*

**Response:** Ok.

**Action:** This section of text has been re-written as part of the move to the discussion section. See lines 876-888.

*Samartin et al (2016) not Samaratin et al (2016)*

**Response:** Ok

**Action:** Corrected

*-lines 510-512 « The second consequence of lower seas levels is that terrestrial pollen sites were located further from the moderating effect of the ocean than they are today, resulting in a localised modification of the climate experienced by the site irrespective of regional or global changes.": a ref is lacking*

**Response:** OK.

**Action:** The following reference has been added: ***Geiger, R.: The climate near the ground. Cambridge: Blue Hill Met. Observ. Harvard University, 1960***. (line 602)

*-lines 531-538: "In terms of regional climate, the major ice sheets would have provided significant barriers to westerly atmospheric circulation, or even north-south circulation in the case of the Alps and Pyrenees. As well as representing a physical obstruction, the thermodynamic response of the atmosphere to these high, cold obstructions would have*

*been to encourage the formation of areas of semi-permanent high pressure, similar to those found today for instance over the Greenland ice sheet. In addition, the Laurentide ice sheet located over North America would have generated downstream effects over Europe. These physical and thermodynamic effects would have affected the direction of storm tracks, as well as more local climatic effects commonly associated with ice sheets such as strong katabatic winds.": refs are lacking*

**Response:** OK.

**Action:** We have added the following references: COHMAP (1988), Kageyama,et al. 2021, Velasquez et al 2021, Luetscher et al 2015, Lefort et al 2019  (lines 633-634)

*-Line 563: "despite arboreal pollen forming 70-80% of the pollen assemblage": a significant part of the arboreal pollen is due to Pinus which is clearly overestimated in LGM pollen assemblages due to long distance transport in open areas as during the LGM.*

**Response:** The point being made in the sentence is that the biomisation algorithm is indicating that steppe is the dominant biome, even when arboreal pollen forms 70-80% of the pollen assemblage. This problem is not caused by high levels of arboreal pollen from long-distance transport but is simply a quirk of the biomisation algorithm. However, the reviewer is right in that some samples may be affected by long distance transport of Pine in the open environments of the LGM. However there also appear to be plenty of samples with low or even very low (<20%) arboreal percentages, so not all sites in open areas may be affected by long-distance transport of *Pinus* in the same way. Again, this is one of the reasons why we have applied the MAT tree-cover reconstruction rather than rely on % arboreal pollen.

**Action:** We acknowledge the points above in our revised manuscript (lines 671-674)

*-lines 615-616: "expected, areas of forest reconstruct similar or increased precipitation compared to today, and areas of steppe indicate deceased precipitation (see next section)." The CO2 effect on climate reconstruction (see recent papers by Cleator et al. and Prentice et al) is not discussed, please add a part on this point.*

**Response:** Ok

**Action:** The CO2 problem is revisited in the discussion (lines 852-869)

*-line 618 correct "archaezoological"*

**Response:** Ok

**Action:** changed to "archae**o**zoological" (line 735)

*-line 669 PMIP = Paleoclimate Modelling Intercomparison Project, not "Palaeo-model Intercomparison Project", correct it; many key refs on PMIP project are missing: Jost et al., 2005; Tarasov et al …*

**Response:** Ok. We are not sure what the Tarasov et al reference is though.

**Action:** "*Palaeo-model Intercomparison Project*" has been corrected to "***Paleoclimate Modelling Intercomparison Project***" **Jost et al 2005** and **Kageyama et al 2021** have also been added. (line 800, 803)

-line 372: "suffer from the same problems of dating control, unclear provenance and a potentially limited taxa assemblages." I don't agree with that, you kept a lot of them for your study.

**Response:** We reject 16 out of 26 records used in PMIP studies, which is a lot of sites on which previous conclusions will have been based.

**Action:** This text has been removed.

- line 677: "and the Neural Networks method which is a version of MAT (MAT-NN) ": the method developed by Peyron et al and Tarasov et al is named the PFT method and IS NOT a version of the MAT. It's a method based on Artificial neural networks close to machine-learning methods, with a real calibration dataset and similar to a black box; both methods use PFTs scores to overcome problems associated with the lack of modern analogue but that is all.

**Response:** Agreed, this has been corrected

**Action:** See earlier comments

-lines 678-690: see my major concern; I think that the fact of not differentiating the steppes can lead to the warm temperatures reconstructed here with the MAT; please check.

**Response:** See earlier comments, there appears to be some confusion between biomes and pft's. Warm and cold steppe is differentiated at the PFT level used in the MAT reconstructions

**Action:** None

-line 721: diatom not Diatom

**Response:** Ok

**Action:** "*Diatom*" changed to "***diatom***" (line 927)

-line 730 check "Hughes et al (Hughes et al., 2006)"

**Response:** Agreed

**Action:** Corrected to *Hughes et al (2006*) (line 937)

-line 755 "19.1 ºC" or -19.1 ºC ?

**Response:** Agreed

**Action:** Corrected to *-19.1* (line 970)

-line 763 "This compares with -7.2 ºC for our 63 pollen sites": not sure it makes sense to calculate the mean for 63 sites given the regional climate patterns

**Response:** We agree with the reviewer, but in this case we are comparing with Allen et al (2008) who undertook a similar calculation.

**Action:** None

-lines 778-784: Good to add a comparsion with the brGDDTs temperature record from Padul (Rodrigo-Gámiz et al., 2022).

**Response:** We are reluctant to include this study by Rodrigo-Gámiz et al. 2022 because this record looks quite odd. In particular, it appears warmer than the present day for much of the glacial period and has a long-term trend very similar to pH. This is important because the brGDDT proxy has been criticised for being influenced by pH as well as temperature, although this potential bias does not appear to be mentioned in the paper. We do not think that excluding the study would make any significant difference to the conclusions of the paper.

**Action:** None

-line 806: I think a part on the comparison of these results with LGM model outputs is lacking.

**Response:** We agree, but including a comprehensive data-model comparison would greatly extend the paper. We have a different paper in preparation which addresses this (Russo et al.), and of course the results will be made available for the whole community as soon as our manuscript is accepted for publication.

**Action:** None

-lines 856-857 "Nevertheless, one of the most consistent signals in our dataset is for an increase in summer precipitation over many areas of Southern Europe and the Mediterranean". In south Spain, the reconstructed biomes is steppe or xerophytic, with a lot of Artemisia and chenopodiaceae: these taxa are characteristic of dry environments (semi-desert), so how do you explain the wetter than today conditions reconstructed?

**Response:** It may seem a little counter-intuitive, but it is still possible to have quite a large change in climate without radically changing the vegetation, especially the pollen biome. For

instance, a semi-arid climate ranges from 250-500mm rainfall a year, so we could expect a semi-arid vegetation to be dominant even if the rainfall increases 250mm. Even beyond 500mm per year, you can still find Artemisia and Chenopodiaceae in the landscape where edaphic conditions are favourable, for instance with a saline geology in the Mediterranean, or even somewhere like the heathlands of northern Germany.

Action: The following text has been added (lines 1098-1104): ). "*It may seem counter-intuitive to see an increase in reconstructed precipitation in the same regions where we also find a preponderance of steppe or xerophytic biomes and taxa, including Artemisia and Chenopodiaceae. This is attributable to the fact that climate can change quite markedly with necessarily invoking a major change in vegetation, and especially the pollen biome. For instance, a semi-arid climate ranges from 250-500mm rainfall a year, so we could expect a semi-arid vegetation to be dominant even if the rainfall increases 250mm (100%).*"

==-check your reference list : Allen et al., 2008 a and b, two refs for Peyron et al 1998 ..==

**Response:** Ok

**Action:** These have been corrected, the duplicate Peyron et al 1998 has been removed and the Allen et al 2008 a and 2008b references have been cited in the text at the appropriate point.
* * *
**Reviewer 1**

*==One of the main places to improve the paper is the graphical representation of the findings. There is a detailed comparison of results from this study with other published records of vegetation, faunistic (zoological remains), and climate. I wonder if it is possible to show some of these values /comparisons on the figures. Otherwise, there are pages of text in the manuscript with no possibility of seeing this visually, which is a pity, as this would significantly improve the paper's impact.==*

**Response:** We agree with the reviewer, but we were worried about over-crowding the figures.

**Action:** We have added this comparison to the figures and show them in appendix figures A4 and A5.

*==More specific Response:==*

*==40 to 52, a nice overview; please add some references to support these statements. Here and in other places in the text, please see a very recent book describing the landforms of the European glacial landscapes:==*

*https://www.sciencedirect.com/book/9780323918992/european-glacial-landscapes#book-description*

**Response:** Agreed, see response to the same comment by Reviewer 2. Unfortunately the book is behind a paywall but we have added some other references.

**Action:** The following references have been added: ***Ehlers et al. 2011, Arslanov et al. 2007, Lehmkuhl et al. 2021, Grichuk 1992*** (lines 51-62)

*64-67 Please extend the relationship between climate CO2 and vegetation slightly.*

**Response:** Agreed, see response to the same comment by Reviewer 2

**Action:** See earlier comment by reviewer 2, the paragraph has been extended to include this information (lines 77-89)

*89 perhaps also add the rates of plant expansion; generally, these are very high assuming the postglacial expansion from southern refugia, and generally, this does not fit modeling results (for example, Nogués-Bravo et al. 2018; TREE, 33, 765-76; Feurdean et al., 2013, Plos One, 26, 8 71797, etc ).*

Response: Agreed, a good addition

Action: The following sentence has been added: "***Modelling have shown difficulty in supporting the very high rates of postglacial expansion that would be necessary for southern refugia (Feurdean et al., 2013, Nogués-Bravo et al. 2018).***" (lines 118-120)

*138, so there were 63 records, 27 with raw counts, and 35 digitized? Please re-write this sentence to make these numbers more transparent.*

**Response:** Ok

**Action:** The following sentence has been changed from "*Overall, 35 out of 63 records were digitized, while the rest of the data consisted of raw pollen counts*" to "***Overall we have included 63 records in our study, of which 35 were digitized and 28 consisted of the original pollen counts (Table 1).*** " (lines 178-180)

*L172-180 may consider moving these levels of detail at the SI*

**Response:** We include this information in the main text because we think it is important to place the current study in the context of previous work. The quality control criteria is one of the key innovations of the study, and the exclusion of records that have been included in previous studies shows the impact of applying this quality control criteria.

**Action:** None

*248-251 is too long and a complicated sentence, please rephrase*

Response: Ok

Action: The following sentence has been changed from : "*To reduce this problem it is possible to systematically exclude closely located modern samples from the analogue matching process, for instance, by excluding samples that fall within a certain spatial range (h-block filter) (Telford and Birks, 2009).*" To "**To reduce this problem it is possible to exclude closely located samples from the analogue matching process using a filter based on a set distance (h-block filter) (Telford and Birks, 2009)**" (lines 312-314)

*l.261  What exactly is meant here by modern climate?*

Response: Ok

Action: The following sentence has been changed from: "*These have been calculated with respect to modern climate at each core site location using WorldClim 2*" to "*These have been calculated with respect to modern climate **(1970-2000 average)** at each core site location using WorldClim 2*" (lines 334-337)

*l.267-272, these lines should be supported by a ref*

**Response:** Agreed, this was also a comment from reviewer 2

**Action:** The following references have been added: **Davis 1963, Gaillard et al. 2010, Zanon et al. 2018** (lines 343-344)

*The names of taxa (Pinus, pine, birch, to name a few) appear wrongly written everywhere I guess it is due to the software conversion; please amend.*

**Response:** Agreed.

**Action:** Taxa names have been italicised where appropriate

*Results. I think one should avoid comparisons/ references to other studies in the Results and should be placed in the discussion*

**Response:** This is similar to a comment by reviewer 2, this section has been modified and a large part moved to the discussion

**Action:** See response to earlier comment

*418 I am surprised to see the low percentages of Chenopodiaceae, Asteraceae, and Artemisia, over most of Europe*

**Response:** The values are still high at some sites (40%+), but it is true, they are not high at many sites. This is one of our main conclusions, that there was more diversity in the

vegetation landscape across Europe at the LGM than has previously been suggested. It wasn't all cold steppe.

**Action:** This is discussed in section 4.1, lines 638 onwards

*502 ff Chapter 4.0 also, please see the new book 2022 European glacial landscape: the last deglaciation shttps://www.sciencedirect.com/book/9780323918992/european-glacial-landscapes#book-description*

**Response:** This looks like a nice book but unfortunately it is behind a paywall.

**Action:** None

*626 ff see also Demay et al., 2021 Quaternary International 581-582, 258–289.*

**Response:** Agreed, nice paper.

**Action:** Demay et al 2021 has been added (line 745)

*Conclusions: I found them overall too long, too many details. I think they should provide better summaries of the essential findings, for ex. L.889-891, l.903-904 sound like results, and the overall ending phrase is missing.*

**Response:** Ok

**Action:** The conclusion has been re-written and shortened to better emphasise the main findings

*The number of graphs and figures made the number of illustrations very high and somehow redundant. Better keep the maps and send graphs to SI. This way, one can accommodate a comparative figure with published records described extensively in the discussion.*

**Response:** Ok.

**Action:** The chronology table has been moved to the appendix, table A1. The tree-cover figure and the pollen diagram figure have also been moved to the appendix (figures A1 and A2 respectively). The comparative figures are now included, although we have put these in the appendix and not in the main text because adding the results of the other studies does make them very busy (temperature figure A4, precipitation figure A5).

---

## Referee Report (RR1)

Thank you for the opportunity to review again the manuscript entitled "The climate and vegetation of Europe, North Africa and the Middle East during the Last Glacial Maximum (21,000 years BP) based on pollen data" by Davis B. and coauthors.

I'd like to thank the authors for taking into account some of my comments in the new manuscript: they've tested the reliability of their approach on the steppe biome and added a table with the R2s and RMSE, and the section on CO2 in the discussion has been greatly improved. Even if I don't necessarily agree with the authors on certain points (I really prefer multi-methods, which is better to understand the reliability of the results obtained), I accept their response.

I still have a few comments to make, and as soon as these are taken into account in the next version, I think the paper can be definitively accepted.

*-line 220 "to match fossil samples with modern calibration pollen samples": the MAT is an assemblage approach which require no statistical calibration, so correct it (the modern pollen samples dataset is not a calibration dataset as it's the case for the WAPLS for example).*
**Response:** The term calibration is widely used with respect to MAT in the literature. See Simpson (2007) *"The modern analogue technique, described below, is an inverse multivariate **calibration** approach."* Or Juggins & Birks (2012) for instance figure 14.3, part of which is shown below.

I don't agree with the authors. The MAT is not a calibration approach : methods based on NN and WAPLS are true transfer function and are based on mathematical calibrations, but MAT is based on a comparison between modern and fossil pollen assemblages (or PFT in your case); there is no calibration in this method (see Guiot et al original paper) and the recent paper by Chevalier et al 2020 "MAT is a classification method, classification techniques compare fossil pollen assemblages to collections of assemblages for which climate is known to identify which assemblages are most similar to the fossil ones".
Please remove the term calibration in the text

*--line 221-223 "This is a similar approach to that used by Peyron et al. (1998) and Jost et al. (2005) who also applied pollen PFT scores to reconstruct LGM climate from pollen data, but who used a neural network technique which is a variant of the standard MAT (Chevalier et al., 2020)". I disagree with that, there is a confusion here in the principle of each method. The Artificial neural networks used by peyron et al and others studies IS NOT a variant of the MAT. It's a method close to machine-learning methods, with a real calibration dataset and not easy to check because similar to a black box; in contrast the MAT is very simple, based on an dissimilarity calculation. The only common point is that both methods use PFTs scores to overcome problems associated with the lack of modern analogue but that is all.*
**Response:** Already agreed, see answer to earlier comment 3.
**Action:** See answer to comment 3.
You have corrected the abstract not this part. Please correct it here too. I propose to replace our sentence by "Other methods using PFT scores and artificial neural network techniques have been developed to reconstruct the climate of Europe during the LGM from pollen data (Peyron et al. (1998) and Jost et al (2005).

-line 312 "Similarly, quantitative climate methods have been applied to individual marine pollen records (Combourieu Nebout et al., 2009; Fletcher et al., 2010)": some key references are missing, as the MF Sanchez Goni team.

**Response:** Unfortunately, the reviewer does not provide any details of the key references that are supposed to be missing. While MF Sanchez Goni and her team have published many important papers, we cannot find any that involve quantitative reconstructions of climate based on pollen, which is the subject of the sentence

**Action:** None

Salonen, J. & Sanchez Goñi, Maria & Renssen, Hans & Plikk, Anna. (2021). Contrasting northern and southern European winter climate trends during the Last Interglacial. Geology. 49. 10.1130/G49007.1. Or

Sánchez Goñi, M.F., Loutre, M.F., Crucifix, M., Peyron, O., Santos, L., Duprat, J., Malaizé, B., Turon, J.-L., and Peypouquet, J.-P., 2005, Increasing vegetation and climate gradient in western Europe over the Last Glacial inception (122–110 ka): Data–model comparison: Earth and Plan etary Science Letters, v. 231, p. 111–130, https://doi.org/10.1016/j.epsl.2004.12.010.

*- -lines 337-347 "we did not adjust the pollen assemblage for the over-representation of Pinus in the marine pollen samples" This poses the problem of Pinus transport over very long distances in open environments as the LGM vegetation; this is particularly true for marine cores but it is also true for some terrestrial sites. So the question of excluding or keeping Pinus needs to be more investigated and tested may be on a site-by-site basis.*

**Response**: Agreed, but the problem of over (or under) representation due to differential transport is a problem that is intrinsic to the science of palynology with no straight-forward answer. Fundamental to this is the fact that although the risk of under/over representation can be acknowledged, it is generally very difficult to detect and correct in any detail. One of the closest attempts at this can be seen in the use of MAT methods to reconstruct tree cover, which we apply in our study. This 'black box' approach at least makes some attempt to take into account the potential over-representation of *Pinus* in both terrestrial and marine environments, at least where this problem is also found in the modern analogue samples that are matched with the fossil samples.

From the point of view of the marine pollen records, we find it more appropriate to include rather than exclude *Pinus* because it is a key forest forming tree in the coastal regions close to the marine sites and to remove it completely would create an artificially arid assemblage that would certainly undermine the ability of the transfer function to reconstruct precipitation. Reconstructions of temperature would be less affected because *Pinus* is a generalist found in both hot and cold regions and so carries only a weak temperature signal compared to the rest of the assemblage. We can add this clarification to the text.

**Action:** The following text has been added to clarify the problem for the reader (lines 435-439) "***Removing Pinus from the assemblage would almost certainly create an artificially arid assemblage in these circumstances, undermining the ability of the transfer function to reconstruct precipitation, although temperature would likely be less affected since Pinus is a generalist found in both hot and cold temperature regions***.".

I don't agree with the fact that you keep Pinus in the marine records for the climate reconstruction: it could change the biome!
Palynologists working on marine records exclude it of the pollen sum, and its particularly true for open environment as LGM. A solution may be you to test your method with and without pinus and check the incidence of removing Pinus on the climate reconstruction by comparing the results with terrestrial close records.
It's an important point for me, but I leave the editor take his decision.

-lines 443-444 "the neural-network methodology of Peyron et al. (1998) and Jost et al. (2005) which we call MAT-NN, as well as the Inverse Modelling approach by Wu et al. (2007) which we

call INV." First, the neural networks methodology of peyron et al. is NOT a MAT method, so you cannot call it MAT-NN, it's a non-sense. Second, could you use the name of the method given in the reference papers? Please check, I guess it's the PFT method for Peyron et al and I.M. for Wu et al. which are correct.

**Response**: We intended our method acronyms to be as self-explanatory as possible. 'PFT' is not the defining feature of the Peyron et al 1998 method, since the use of PFT scores can, and has, been used in other methods such as MAT. We therefore prefer to use the acronym 'ANN' for Artificial Neural Network (as used by Chevalier et al 2019).

**Action:** The section mentioned by the reviewer has been moved to the discussion as they suggest. MAT-NN has been changed to ANN throughout.

Better to use PFT-ANN for the study of Peyron et al and IM for the study of Wu et al (not INV, I dont understand why you have changed it): please correct in the text

-lines 615-616: *"expected, areas of forest reconstruct similar or increased precipitation compared to today, and areas of steppe indicate deceased precipitation (see next section)." The CO2 effect on climate reconstruction (see recent papers by Cleator et al. and Prentice et al) is not discussed, please add a part on this point.*

**Response: Ok**

**Action:** The CO2 problem is revisited in the discussion (lines 852-869)

Ok, perfect

-lines 778-784: *Good to add a comparsion with the brGDDTs temperature record from Padul (Rodrigo-Gámiz et al., 2022).*

**Response:** We are reluctant to include this study by Rodrigo-Gámiz et al. 2022 because this record looks quite odd. In particular, it appears warmer than the present day for much of the glacial period and has a long-term trend very similar to pH. This is important because the brGDDT proxy has been criticised for being influenced by pH as well as temperature, although this potential bias does not appear to be mentioned in the paper. We do not think that excluding the study would make any significant difference to the conclusions of the paper.

**Action:** None

It's often the case with BRGDGTs studies: the temperature values are depending on the calibration used (here the Martinez-Sosa et al one), and are often too high. I think that the most important is to look at the climate patterns: an important result is that they show that LGM temperature were higher than those reconstructed during Heinrich events. Please cite this paper.

---

## Editor Decision (ED1)

As editor at C. Past, I am succeeding Nathalie Combourieu-Nebout to finalize the revision process of the paper entitled "The climate and vegetation of Europe, North Africa and the Middle East during the Last Glacial Maximum (21,000 years BP) based on pollen data."

First, many thanks to thanks B. Davis and co-authors for proposing a new version of his paper that takes into account the previous comments of the reviewers and the editor. I find the new version much improved. Therefore I decide to definitively accept it after a few minor corrections listed as follows :

Abstract, line 28 "The reconstructions are based on the modern analogue technique (MAT) with a modern calibration pollen dataset taken from the latest Eurasian Modern Pollen Database (~8000 samples)." **You don't use the standard MAT (Guiot et al papers) as the MAT is based on pollen counts; if you calculate a dissimilarity index not on raw counts but on PFTs scores as you did in Davis et al 2003, you need to clarify it. I suggest " The reconstructions are based on the modern analogue technique (MAT) adapted with PFT scores…".**

- Abstract, lines 36-37 "Differences between our latest MAT reconstruction and those in earlier studies can be largely attributed to bias in the small modern calibration dataset previously used". **I agree  but differences can also been explained by the method itself (see Brewer et al 2008 or Salonen et al 2019 for multi-method approaches). I suggest "… to bias in the small modern calibration dataset previously used and also to the method itself (Brewer et al 2008, Salonen et al 2019)".**

- Introduction, line 70, **a key reference is missing, please add it. Braconnot, P., Harrison, S.P., Kageyama, M., Bartlein, P.J., Masson-Delmotte, V., Abe-Ouchi, A., Otto-Bliesner, B., and Zhao, Y.: Evaluation of climate models using palaeoclimatic data, Nat. Clim. Change, 2, 417–424, https://doi.org/10.1038/nclimate1456, 2012.**

- Introduction, line 88 "through the use of a correction algorithm (Prentice et al. 2017)"**Here, the recent papers by Cleator et al 2020 is missing, please add it.**

- Introduction, line 91 "Pollen-climate reconstructions based on inverse modelling that account for these low CO2 effects show less cooling and drying and consequently greater agreement with climate models (Ramstein et al., 2007; Wu et al., 2007)." **Key recent papers using the INV for the LGM are missing, please add it.**
> **- Izumi and Bartlein, 2016:  North American paleoclimate reconstructions for the Last Glacial Maximum using an inverse modeling through iterative forward modeling approach applied to pollen data, https://doi.org/10.1002/2016GL070152**
> **- Wu et al 2019. Quantitative climatic reconstruction of the Last Glacial Maximum in China. Sci. China Earth Sci. 62, 1269–1278 (2019). https://doi.org/10.1007/s11430-018-9338-3**

- Introduction, line 161 "In addition, … using the Modern Analogue Technique (MAT)…"  and  Methods lines 277-279 "We reconstructed climate from pollen data based on a standard Modern Analogue Technique (MAT) that used PFT scores to match fossil samples with modern pollen samples (as used by Davis et al., 2003)." **same as in the abstract: you don't use the standard version of the MAT, so avoid the term standard and replace it by "a modified version of the standard MAT (Guiot et al 1989)…**

- Methods 2.6 Marine pollen records: **thanks so much for this new part! You provide very interesting new results which have never been discussed before. I just suggest to put it in 2.4 before the part on the vegetation cover**

- Results line 576 *Pinus* **in italic**

- Results line 607-610 "comparisons between studies can only be made with caution because results are often heavily dependent on the nature of the modern pollen dataset used as the training set, which is not the same in all studies (Juggins, 2013)." I **agree with you; the choice of the method is also very important, so I suggest to add "and results also largely depend on the method used (Salonen et al., 2019; Brewer et al., 2008; Peyron et al., 2013)"**

- Discussion line 690 : **tundra not Tundra ,** and line 821 **boreal not Boreal**

- Discussion line 835: (see figure 6 in Velasquez et al., 2021); **just cite the ref Velasquez et al., 2021);**

- Discussion line 848: (Kageyama et al., 2021,Bartlein et al., 2011; Harrison et al., 2015; Kageyama et al., 2006). **References are missing : Braconnot et al 2007 ; Braconnot et al 2012, Cleator et al 2020**

- Discussion line 855: « but instead uses a process-based vegetation model run in inverse mode." **The ref Guiot et al 2000 is missing, please add it**

- Discussion line 857 : «  but in inverse mode the model is reconfigured to generate climate as an output given a particular vegetation (pollen) assemblage as an input." **not true: in the inverse modelling developed by Guiot and updated by Wu et al, 2007, 2019 input data are climate data (and CO2) and output data are PFTs scores simulated by the vegetation model; these PFTs scores are compared to the pollen-inferred PFT scores following an iterative process: the climate value is selected as the most probable (when the error between the simulated and pollen-based PFTs score is the lowest). Please correct**

- Discussion line 981 : **Lago della Costa just after in the text, please check**

- Discussion line 1011 : "A number of additional proxies have also been used to reconstruct LGM mean annual temperature**". a recent ref is missing for your comparison: Last glacial maximum cooling of 9 °C in continental Europe from a 40 kyr-long noble gas paleothermometry record Bekaert, D.V .et al 2023 Quaternary Science Reviews, 310, 108123**

- Discussion line 1014 : **correct to the Vosges Mountains**

- Discussion line 1027 : "This compares with -7.2 C for our 63 pollen 1028 sites.": **not sure to understand which is compared**

- Discussion line 1046: "Further south and west…"**A reference is missing for your comparison : Rodrigo-Gamiz et al 2022 : Padul new record, lipids biomarkers, temperature close to the current ones.**

- Discussion line 1085: "Few proxies apart from pollen provide quantitative reconstructions of precipitation during the LGM." **A comparison with the paper by García-Alix et al 2021 is missing García-Alix et al 2021. Paleohydrological dynamics in the Western Mediterranean during the last glacial cycle, Global and Planetary Change, 202, 2021,103527, https://doi.org/10.1016/j.gloplacha.2021.103527.**

- Conclusion lines 1196 and 1200**: boreal not Boreal;** line 1221**: comparison not comparsons**

-Figure3:  **in the figure correct arborel pollen**
-Figure 5 and 7, **the name of the sites is too small, the title of each curve also, please correct it**
- figure A3: **this figure`s panels are included separately with their own captions. This will be unacceptable in the final paper**

**Odile Peyron, for Climate of the Past**

---

## Author Response (AR2)

*CP-2022-59 Davis et al. The climate and vegetation of Europe, North Africa and the Middle East during the Last Glacial Maximum (21,000 years BP) based on pollen data*

Dear Editor, Reviewer #1 appears to ask for no further revisions to the manuscript. Reviewer #2 makes 6 comments, which I respond to below in RED:

**Referee 2**

Thank you for the opportunity to review again the manuscript entitled "The climate and vegetation of Europe, North Africa and the Middle East during the Last Glacial Maximum (21,000 years BP) based on pollen data" by Davis B. and coauthors.

I'd like to thank the authors for taking into account some of my comments in the new manuscript: they've tested the reliability of their approach on the steppe biome and added a table with the R2s and RMSE, and the section on CO2 in the discussion has been greatly improved. Even if I don't necessarily agree with the authors on certain points (I really prefer multi-methods, which is better to understand the reliability of the results obtained), I accept their response.

I still have a few comments to make, and as soon as these are taken into account in the next version, I think the paper can be definitively accepted.

*-line 220 "to match fossil samples with modern calibration pollen samples": the MAT is an assemblage approach which require no statistical calibration, so correct it (the modern pollen samples dataset is not a calibration dataset as it's the case for the WAPLS for example).*
**Response:** The term calibration is widely used with respect to MAT in the literature. See Simpson (2007) *"The modern analogue technique, described below, is an inverse multivariate **calibration** approach."* Or Juggins & Birks (2012) for instance figure 14.3, part of which is shown below.

1. I don't agree with the authors. The MAT is not a calibration approach : methods based on NN and WAPLS are true transfer function and are based on mathematical calibrations, but MAT is based on a comparison between modern and fossil pollen assemblages (or PFT in your case); there is no calibration in this method (see Guiot et al original paper) and the recent paper by Chevalier et al 2020 "MAT is a classification method, classification techniques compare fossil pollen assemblages to collections of assemblages for which climate is known to identify which assemblages are most similar to the fossil ones".
Please remove the term calibration in the text

Response: The referee is making an argument about semantics. They may be correct in a purely mathematical sense, but words (including scientific terminology) can and do have different meanings in different contexts. The referee may 'disagree' with the use of the word 'calibration' by Simpson (2007) and Juggins & Birks (2012) but the term 'calibration' is widely used by the scientific community when talking about MAT, and especially when talking about the 'calibration dataset' and not 'calibration' as a process. Even in Chavalier et al 2020 in the specific section (5.3.1) on MAT to which the referee refers, the word calibration is used at least twice "..*selecting more and more analogues will progressively include drier samples from the rest of the climate space represented in the **calibration** data, thus inducing an undesired dry bias on the reconstruction (Gajewski, 2015; Viau et al., 2008).* As well as; "*However, including more analogues also increases the risk of false positive matches, especially when the **calibration** dataset encompasses wide spatial areas where the low taxonomic resolution of pollen data can..*".

I am not sure which paper by Guiot "Guiot et al original paper" that the referee is talking about. As with previous comments by the referee, it would be helpful to provide precise information. Certainly, all of the original papers by Joel Guiot from the 1980's that reference an 'analogue' method make reference to calibration, for instance Guiot 1987:

[Figure]

Although the method used by Guiot at this time is not the 'analogue' method that we use, which is closer to Overpeck et al 1985 (in which the term calibration is used at least 4 times).

Action: None

--line 221-223 *"This is a similar approach to that used by Peyron et al. (1998) and Jost et al. (2005) who also applied pollen PFT scores to reconstruct LGM climate from pollen data, but who used a neural network technique which is a variant of the standard MAT (Chevalier et al., 2020)".* I disagree with that, there is a confusion here in the principle of each method. The Artificial neural networks used by peyron et al and others studies IS NOT a variant of the MAT. It's a method close to machine-learning methods, with a real calibration dataset and not easy to check because similar to a black box; in contrast the MAT is very simple, based on an dissimilarity calculation. The only common point is that both methods use PFTs scores to overcome problems associated with the lack of modern analogue but that is all.
**Response:** Already agreed, see answer to earlier comment 3.
**Action:** See answer to comment 3.
2. You have corrected the abstract not this part. Please correct it here too. I propose to replace our sen- tence by "Other methods using PFT scores and artificial neural network techniques have been devel- oped to reconstruct the climate of Europe during the LGM from pollen data (Peyron et al. (1998) and Jost et al (2005).

Response: I am not sure what the reviewer is referring to. The text and line numbers shown above are from the first draft of the manuscript, not the revised version. This section WAS changed in the latest version 3 of the manuscript. These changes were also shown in the response to the reviewer's comments. The text has already been corrected according to the referee's instructions.

Action: None

-line 312 "Similarly, quantitative climate methods have been applied to individual marine pollen records (Combourieu Nebout et al., 2009; Fletcher et al., 2010)": some key references are missing, as the MF Sanchez Goni team. **Response:** Unfortunately, the reviewer does not provide any details of the key references that are supposed to be missing. While MF Sanchez Goni and her team have published many im- portant papers, we cannot find any that involve quantitative reconstructions of climate based on pollen, which is the subject of the sentence
**Action:** None
3. Salonen, J. & Sanchez Goñi, Maria & Renssen, Hans & Plikk, Anna. (2021). Contrasting northern and southern European winter climate trends during the Last Interglacial. Geology. 49. 10.1130/G49007.1. Or
Sánchez Goñi, M.F., Loutre, M.F., Crucifix, M., Peyron, O., Santos, L., Duprat, J., Malaizé, B., Turon, J.-L., and Peypouquet, J.-P., 2005, Increasing vegetation and climate gradient in western Europe over the Last Glacial inception (122–110 ka): Data–model comparison: Earth and Plan etary Science Letters, v. 231, p. 111–130, https://doi.org/10.1016/j.epsl.2004.12.010.

Response: Ok

Action: The 2 references have been added

*- -lines 337-347 "we did not adjust the pollen assemblage for the over-representation of Pinus in the marine pollen samples" This poses the problem of Pinus transport over very long distances in open environments as the LGM vegetation; this is particularly true for marine cores but it is also true for some terrestrial sites. So the question of excluding or keeping Pinus needs to be more investigated and tested may be on a site-by-site basis.*

**Response**: Agreed, but the problem of over (or under) representation due to differential transport is a problem that is intrinsic to the science of palynology with no straight-forward answer. Fundamental to this is the fact that although the risk of under/over representation can be acknowledged, it is generally very difficult to detect and correct in any detail. One of the closest attempts at this can be seen in the use of MAT methods to reconstruct tree cover, which we apply in our study. This 'black box' approach at least makes some attempt to take into account the potential over-representation of *Pinus* in both terrestrial and marine environments, at least where this problem is also found in the modern analogue samples that are matched with the fossil samples.

From the point of view of the marine pollen records, we find it more appropriate to include rather than exclude *Pinus* because it is a key forest forming tree in the coastal regions close to the marine sites and to remove it completely would create an artificially arid assemblage that would certainly undermine the ability of the transfer function to reconstruct precipitation. Reconstructions of temperature would be less affected because *Pinus* is a generalist found in both hot and cold regions and so carries only a weak temperature signal compared to the rest of the assemblage. We can add this clarification to the text.

**Action:** The following text has been added to clarify the problem for the reader (lines 435-439) "***Removing Pinus from the assemblage would almost certainly create an artificially arid assemblage in these circumstances, undermining the ability of the transfer function to reconstruct precipitation, although temperature would likely be less affected since Pinus is a generalist found in both hot and cold temperature regions***.".

4. I don't agree with the fact that you keep Pinus in the marine records for the climate reconstruction: it could change the biome! Palynologists working on marine records exclude it of the pollen sum, and its particularly true for open environment as LGM. A solution may be you to test your method with and without pinus and check the incidence of removing Pinus on the climate reconstruction by comparing the results with terrestrial close records. It's an important point for me, but I leave the editor take his decision.

Response: The referee has not engaged with the argument. Removing *Pinus* can be just as detrimental as keeping *Pinus* (and it can equally "change the biome"), they are two sides of the same problem. In the manuscript we give the example of coastal Portugal where *Pinus* is the dominant forest forming tree. If you remove *Pinus* from the sum then you are removing a key component of the ecosystem, and a key indicator of the climate. It should be enough that the reader is reminded of this problem, which we do in the text. I would note that both Salonen et al 2021 and Sánchez Goñi 2005 highlighted by the referee (see above) don't even mention this problem, and it is not clear that Salonen et al 2021 even removed *Pinus* in their marine-based pollen-climate reconstruction, since the method section simply says that they used the same method as an earlier paper, and that earlier paper only analyzed terrestrial samples where *Pinus* was included in the sum.

Action: None

-lines 443-444 "the neural-network methodology of Peyron et al. (1998) and Jost et al. (2005) which we call MAT-NN, as well as the Inverse Modelling approach by Wu et al. (2007) which we call INV." First, the neural networks methodology of peyron et al. is NOT a MAT method, so you cannot call it MAT-NN, it's a non-sense. Second, could you use the name of the method given in the reference papers? Please check, I guess it's the PFT method for Peyron et al and I.M. for Wu et al. which are correct.

**Response**: We intended our method acronyms to be as self-explanatory as possible. 'PFT' is not the defining feature of the Peyron et al 1998 method, since the use of PFT scores can, and has, been used in other methods such as MAT. We therefore prefer to use the acronym 'ANN' for Artificial Neural Network (as used by Chevalier et al 2019).

**Action:** The section mentioned by the reviewer has been moved to the discussion as they suggest. MAT-NN has been changed to ANN throughout.

5. Better to use PFT-ANN for the study of Peyron et al and IM for the study of Wu et al (not INV, I dont understand why you have changed it): please correct in the text

Response: I don't really see the necessity to add PFT to ANN since we only use the ANN results of Peyron et al, who used PFT scores in their ANN analysis. Peyron et al themselves use just the acronym ANN, as do Chevalier et al 2019. Confusingly Wu et al call the ANN approach of Peyron et al just 'PFT'. Wu et al also call their inverse modelling approach IVM, not IM as the referee suggests. In summary there does not appear to be a consensus on the use of acronyms for these methods, although more so for ANN, so the fact that we use ANN and INV seems reasonable.

Action: None

*-lines 615-616: "expected, areas of forest reconstruct similar or increased precipitation compared to today, and areas of steppe indicate deceased precipitation (see next section)." The CO2 effect on climate reconstruction (see recent papers by Cleator et al. and Prentice et al) is not discussed, please add a part on this point.*

**Response: Ok**

**Action:** The CO2 problem is revisited in the discussion (lines 852-869)

Ok, perfect

*-lines 778-784: Good to add a comparsion with the brGDDTs temperature record from Padul (Rodrigo-Gámiz et al., 2022).*

**Response:** We are reluctant to include this study by Rodrigo-Gámiz et al. 2022 because this record looks quite odd. In particular, it appears warmer than the present day for much of the glacial period and has a long-term trend very similar to pH. This is important because the brGDDT proxy has been criticised for being influenced by pH as well as temperature, although this potential bias does not appear to be mentioned in the paper. We do not think that excluding the study would make any significant difference to the conclusions of the paper.

**Action:** None

6. It's often the case with BRGDGTs studies: the temperature values are depending on the calibration used (here the Martinez-Sosa et al one), and are often too high. I think that the most important is to look at the climate patterns: an important result is that they show that LGM temperature were higher than those reconstructed during Heinrich events. Please cite this paper.

Response: The referee is insisting on us citing a paper that has little or no scientific merit in relation to our manuscript. I do not know if the referee has any connection with this paper, but this appears to us to be highly unethical.

Action: None

---

## Author Response (AR3)

**Cp-2022-59 Davis et al Response to editors comments**

Point 1 regarding the terms
It is important to have a carefull use of the words even if you considered it is only semantic. I ask you to follow the request of the reviewer concerning the use of the term calibration. Please do that before the acceptance of the manuscript. If not, justify in the text the term calibration by add something about that and join the reference in additional data for example. It is essential for the non-specialists of climate reconstruction to know what you have exactly used and to solve this disagreement.

**Response:** Ok

**Action:** The word 'calibration' has been removed from the text
* * *
Point 2 concerning the sentence pointes in line 121 although just at the beginning of section 2.3
Perhaps the reviewer did not note the good lines but the sentence is in your text and I think it will be better to change in accordance with this remark. So adapt your sentence in accordance please.

**Response:** Ok

**Action:** The sentence "" has been removed and replaced by the sentence requested by the reviewer "*Other methods using PFT scores and artificial neural network techniques have been devel- oped to reconstruct the climate of Europe during the LGM from pollen data (Peyron et al. (1998) and Jost et al (2005).*"
* * *
Point 3 the reconstruction and the problem of Pinus
I think that the reviewer point the problem of quantitative reconstruction including Pine when you are working on marine cores. It is not sufficient to just add the référece. I think that more explanation are necessary. The dta published by Sanchez Goni team and Salonen used method excluding Pinus from the counts which is so different from your method. Some precisions have to be included in the text to clarify that. Did you use the raw pollen data from the marine sites (complete, those used by Sanchez Goni) or the values from the reconstruction published in these paper? Or do you do another run of reconstruction with pine included in all cases?

now it is not clear for me.

Combourieu-Nebout et al., Fletcher et al. and Sanchez Goni et al. used reconstructions excluded PInus in any cases.

The responses to these questions are crucial and have to be included in your text in the method and in the discussion. If not we might considered that you compare apple with pears!!

In addition, you're right saying that the vegetation includes Pinus in western Portugal. Nevertheless, keeping this taxa in the reconstructions in marine core induce a very large bias as at 100 km off the coast the % of pine is over-represented up to 80% in a very big proportion that hide the modification in the other association as the reviewer said to you. For this reason the counts are improved to better represent the vegetation. This fact has been demonstrated since a long, long time by many authors (papers are often referenced in the methods of numerous papers on the pollen records from marine cores).

What do you do with the pollen spectra in some series where some taxa are over-represented due to edaphic conditions or specific inputs? Did you conserve them or not?

This is important to note such things in your paper and more explain it in the methods to close the debate and show that you have done the same on all the records.

When you have 80% of Pinus in marine samples, it did not necessary reflect that you have a high representation of Pine in the vegetation, it is not the same as in peatbogs or lakes. Reasoning only on the other taxa drives to a best representation of the vegetation in which we can add Pine afterwards and the reconstructions without Pinus (using samples and database without Pinus) are not so bad especially concerning the precipitation, temperature being sometime too cold (see the tests done in Combourieu-Nebout et al 2009 on top-core and surface samples in Mediterranean area).

At the end, for my part, I remain not convinced by the reconstructions when you conserve Pine in all the marine core as it is not correct to illustrate the vegetation, used for the reconstruction. It did not change a lot your paper if you had presented different runs for the marine and continental records if explaining why by justifying the differen between sediments.

So, Please do something about that in your text especially by developping your choices in the methods parts.

**Response:** I understand that you would like us to clarify more clearly the choices made in the methods section. To try and resolve this problem scientifically, we have now included two additional analyses that investigate the effect of including/excluding Pinaceae on 1) the biomisation process and 2) the pollen-climate reconstruction. In the first test we use the modern pollen dataset to investigate the effect on the biome assignment of varying the amount of Pinaceae.

This shows that removing Pinaceae completely changes the biome in 5860 out of 8213 samples, while increasing the Pinaceae as much as 400% only changes the biome in 2348 samples. This suggests that it is better to include rather than exclude the Pinaceae unless the sample clearly shows signs of over representation (eg 95%+ Pinaceae). In the second test we compare the climate reconstructed from the marine LGM samples when Pinaceae is included/excluded. We find that excluding Pinaceae results in temperatures slightly cooler, and precipitation slightly higher, but the values are a lot smaller than the uncertainties. This suggests that there is little difference between the two approaches, irrespective of which may be more appropriate.

**Action:**  The results of the biome experiment are shown in the new supplementary table S3 and the results of the pollen-climate experiment are shown in the new supplementary table S4. In addition, the following text has been added to the results section:

*Line 449+ " The effect of excluding Pinaceae on the biomisation algorithm and MAT climate reconstruction process has not been widely investigated. We therefore decided to evaluate this problem for 1) biomisation, and 2) pollen-climate reconstruction. In table S3 we show the biomisation results for 8213 modern pollen samples taken from the EMPD2 modern pollen database. Using this as the control, we then artificially varied the amount of Pinaceae (Pinus, Abies and Picea) in the assemblage of each pollen sample and compiled the results (Table S3). This shows quite clearly that removing all of the Pinaceae has a much more profound effect on the biomisation process than artificially inflating the amount of Pinaceae (as might be expected in a marine sample where Pinaceae can be over-represented). Even when Pinaceae was artificially inflated by as much as 400% of the original value, the biomes were changed in only 2348 samples, compared to 5860 samples if all the Pinaceae was removed entirely. In terms of the effects on individual biomes, removing the Pinaceae considerably increased the amount of CLDE, STEP and TUND, whilst greatly reducing the amount of XERO, almost eliminating the amount of TAIG, and completely eliminating the COCO biome. In contrast, the effect of inflating the amount of Pinaceae tended to be more evenly distributed between the biomes, with the biggest increase seen in TUND and biggest decrease in STEP. This suggests that even if the over-representation of Pinaceae was quite extreme in marine pollen samples, the effect on biome classification (and by definition, the underlying PFT scores) is less than removing Pinaceae completely from the pollen assemblage.*

*In a second test, we compared the reconstruction of LGM climate from marine pollen samples when Pinaceae was included, and excluded. The results are shown in table S4 and indicate reconstructed temperatures are generally 1-2C cooler, and precipitation slightly higher when Pinaceae is excluded. The differences between the two methods however are small, and generally less than half of the uncertainties, suggesting that*

*differences are statistically indistinguishable when considered in the context of the overall uncertainties.*

*In summary we find that including Pinaceae in the biomisation process is less likely to lead to miss-assignment of the biome than excluding Pinaceae, except in extreme cases of over-representation. Percentages of Pinaceae in the LGM marine samples range on average between 23-88%, suggesting that while Pinaceae was high at some sites, it does not appear to completely overwhelm the assemblage as might be expected if over-representation was to be a significant problem. We also find that including Pinaceae in the pollen assemblage of the LGM marine pollen samples gives pollen-climate reconstructions that are statistically indistinguishable from those obtained by excluding Pinaceae from the assemblage. Including Pinaceae in marine samples also provides compatibility with terrestrial samples, particularly when calculating and plotting pollen taxa percentages. For these reasons we have included Pinaceae in the analysis of all marine pollen samples in this study, although it is important to recognize that Pinaceae in such samples can be subject to over-representation and that the results presented here from marine sites should consequently be viewed with caution. "*
* * *
Point 4 My last remark concerns the figures: It is not acceptable to propose figures as the series presented in figure 3 a,b,.. What is represented by the large white rectangle with the little cross. This need explanations??? Did crosses represent the presence, the sites? This figures are supposed to show the percentages of the taxa but there is nothing on the figures with percentages, only crosses. Please explain in the captions ; the figures must be read alone without the text.

**Response:** We are sorry, this appears to be because something went wrong when the pdf version of the word document was created

**Action:** This has been fixed in the new submission.

---

## Author Response (AR4)

Response to the Editors (O. Peyron) comments:

Many thanks to the editor for taking the time to go through so thoroughly what is now quite a long and involved paper. It is again much improved following her suggestions and comments. Almost all of the 29 changes suggested have been made to the manuscript, with only 19 and 20 being declined for the reasons outlined below.

1. Abstract, line 28 "The reconstructions are based on the modern analogue technique (MAT) with a modern calibration pollen dataset taken from the latest Eurasian Modern Pollen Database (~8000 samples)." **You don't use the standard MAT (Guiot et al papers) as the MAT is based on pollen counts; if you calculate a dissimilarity index not on raw counts but on PFTs scores as you did in Davis et al 2003, you need to clarify it. I suggest " The reconstructions are based on the modern analogue technique (MAT) adapted with PFT scores…".**

**Response:** Ok

**Action:** Text changed to "*The reconstructions are based on the modern analogue technique (MAT) adapted using PFT scores, and with a modern pollen dataset taken from the latest Eurasian Modern Pollen Database (~8000 samples).*"

2. - Abstract, lines 36-37 "Differences between our latest MAT reconstruction and those in earlier studies can be largely attributed to bias in the small modern calibration dataset previously used". **I agree but differences can also been explained by the method itself (see Brewer et al 2008 or Salonen et al 2019 for multi-method approaches). I suggest "… to bias in the small modern calibration dataset previously used and also to the method itself (Brewer et al 2008, Salonen et al 2019)".**

**Response:** Ok

**Action:** Text changed to: "*Differences between our latest MAT reconstruction and those in earlier studies can be largely attributed to bias in the small modern dataset previously used, and differences in the method itself (Brewer et al. 2008, Salonen et al. 2019).*"

3. - Introduction, line 70, **a key reference is missing, please add it. Braconnot, P., Harrison, S.P., Kageyama, M., Bartlein, P.J., Masson-Delmotte, V., Abe-Ouchi, A., Otto-Bliesner, B., and Zhao, Y.: Evaluation of climate models using palaeoclimatic data, Nat. Clim. Change, 2, 417–424, https://doi.org/10.1038/nclimate1456, 2012.**

**Response:** Ok

**Action: Braconnot et al** has been added

- Introduction, line 88 "through the use of a correction algorithm (Prentice et al. 2017)"**Here, the recent papers by Cleator et al 2020 is missing, please add it.**

**Response:** Ok

**Action:** Cleator et al 2020 has been added

- Introduction, line 91 "Pollen-climate reconstructions based on inverse modelling that account for these low CO2 effects show less cooling and drying and consequently greater agreement with climate models (Ramstein et al., 2007; Wu et al., 2007)." **Key recent papers using the INV for the LGM are missing, please add it.**
 **- Izumi and Bartlein, 2016: North American paleoclimate reconstructions for the Last Glacial Maximum using an inverse modeling through iterative forward modeling approach applied to pollen data, https://doi.org/10.1002/2016GL070152**
**- Wu et al 2019. Quantitative climatic reconstruction of the Last Glacial Maximum in China. Sci. China Earth Sci. 62, 1269–1278 (2019). https://doi.org/10.1007/s11430-018-9338-3**

**Response: Ok**

**Action:** Izumi & Bartlein 2016 & Wu et al 2019 have been added

- Introduction, line 161 "In addition, … using the Modern Analogue Technique (MAT)…" and Methods lines 277-279 "We reconstructed climate from pollen data based on a standard Modern Analogue Technique (MAT) that used PFT scores to match fossil samples with modern pollen samples (as used by Davis et al., 2003)." **same as in the abstract: you don't use the standard version of the MAT, so avoid the term standard and replace it by "a modified version of the standard MAT (Guiot et al 1989)…**

**Response:** Ok

**Action:** The text has been changed at the two places identified *"In addition, quantitative reconstructions of forest cover as well as winter, summer and annual temperatures and precipitation were undertaken using a modified version of the standard Modern Analogue Technique (MAT) (Guiot et al. 1989), " and also "We reconstructed climate from pollen data based on a modified Modern Analogue Technique (MAT) (Guiot et al. 1989).*

- Methods 2.6 Marine pollen records: **thanks so much for this new part! You provide very interesting new results which have never been discussed before. I just suggest to put it in 2.4 before the part on the vegetation cover**

**Response:** Ok

**Action:** Section 2.6 has been moved to 2.4

- Results line 576 ***Pinus* in italic**

**Response:** Ok

**Action:** *Pinus* has been italicized

- Results line 607-610 "comparisons between studies can only be made with caution because results are often heavily dependent on the nature of the modern pollen dataset used as the training set, which is not the same in all studies (Juggins, 2013)." I **agree with you; the choice of the method is also very important, so I suggest to add "and results also largely depend on the method used (Salonen et al., 2019; Brewer et al., 2008; Peyron et al., 2013)"**

**Response:** Ok

**Action:** the references and text has been added: "..*can only be made with caution because results are often heavily dependent on the nature of the modern pollen dataset used as the training set (Juggins, 2013), as well as the method used (Salonen et al. 2019, Brewer et al. 2008, Peyron et al., 2013).*"

- Discussion line 690 : **tundra not Tundra ,** and line 821 **boreal not Boreal**

**Response:** OK

**Action:** Tundra has been changed to tundra, Boreal to boreal

- Discussion line 835: (see figure 6 in Velasquez et al., 2021); **just cite the ref Velasquez et al., 2021);**

**Response:** OK

**Action:** '*see figure 6*' has been removed

- Discussion line 848: (Kageyama et al., 2021,Bartlein et al., 2011; Harrison et al., 2015; Kageyama et al., 2006). **References are missing : Braconnot et al 2007 ; Braconnot et al 2012, Cleator et al 2020**

**Response:** OK

**Action:** The references have been added

- Discussion line 855: « but instead uses a process-based vegetation model run in inverse mode." **The ref Guiot et al 2000 is missing, please add it**

**Response:** OK

**Action:** Guiot et al 2000 has been added

- Discussion line 857 : « but in inverse mode the model is reconfigured to generate climate as an output given a particular vegetation (pollen) assemblage as an input." **not true: in the inverse modelling developed by Guiot and updated by Wu et al, 2007, 2019 input data are climate data (and CO2) and output data are PFTs scores simulated by the vegetation model; these PFTs scores are compared to the pollen-inferred PFT scores following an iterative process: the climate value is selected as the most probable (when the error between the simulated and pollen-based PFTs score is the lowest). Please correct**

**Response:** OK

**Action:** The text has been changed accordingly: "..*but in inverse mode the model is reconfigured so that the input climate (and CO2) can be varied iteratively until the closest match is found between the vegetation simulated by the model (represented by PFT scores) and the fossil pollen assemblage (also represented by PFT scores).* "

- Discussion line 981 : **Lago della Costa just after in the text, please check**

**Response:** OK

**Action:** Text has been changed

- Discussion line 1011 : "A number of additional proxies have also been used to reconstruct LGM mean annual temperature**". a recent ref is missing for your comparison: Last glacial maximum cooling of 9 °C in continental Europe from a 40 kyr-long noble gas paleothermometry record Bekaert, D.V .et al 2023 Quaternary Science Reviews, 310, 108123**

**Response:** OK

**Action:** The reference has been added as well as the following text:  "*Mean annual temperatures have also been reconstructed from the Paris basin area in Eastern France by Bekaert et al. (2023) using the Noble gas proxy. The authors suggest an LGM temperature anomaly of -9.1 ± 0.9 ºC although this is actually dated to  25.6±0.5k, which is earlier than our 21k±2.0k time window that we adopt here. The sample closest to 21k is at 21.9±0.5k and suggests slightly warmer temperatures at -7.77 ºC, which compares well with our pollen reconstructions nearby at [Bergsee site #29] -8.2 ± 3.3 ºC and [La Grotte Walou site #28] -6.6 ± 3.1 ºC.*"

- Discussion line 1014 : **correct to the Vosges Mountains**

**Response:** OK

**Action:** Corrected

- Discussion line 1027 : "This compares with -7.2 C for our 63 pollen 1028 sites.": **not sure to understand which is compared**

**Response:** OK

**Action:** Text has been changed from "*This compares with -7.2 ºC for our 63 pollen sites.*" to "*These average anomalies across all sites calculated by Allen et al (2008a) compare with an average temperature anomaly of -7.2 ºC across all 63 of our pollen sites.*"

- Discussion line 1046: "Further south and west…"**A reference is missing for your comparison : Rodrigo-Gamiz et al 2022 : Padul new record, lipids biomarkers, temperature close to the current ones.**

**Response:** This reference was also suggested earlier in the review process by one of the reviewers. This was our response:  "We are reluctant to include this study by Rodrigo-Gámiz et al. 2022 because this record looks quite odd. In particular, it appears warmer than the present day for much of the glacial period and has a long-term trend very similar to pH. This is important because the brGDDT proxy has been criticised for being influenced by pH as well as temperature, although this potential bias does not appear to be mentioned in the paper. We do not think that excluding the study would make any significant difference to the conclusions of the paper."

**Action:** None

- Discussion line 1085: "Few proxies apart from pollen provide quantitative reconstructions of precipitation during the LGM." **A comparison with the paper by García-Alix et al 2021 is missing García-Alix et al 2021. Paleohydrological dynamics in the Western Mediterranean during the last glacial cycle, Global and Planetary Change, 202, 2021,103527, https://doi.org/10.1016/j.gloplacha.2021.103527.**

**Response:** This is an interesting paper, but it provides only qualitative estimates of precipitation change and is partly based on pollen data which also undermines it as an independent proxy. We deliberately didn't want to include qualitative reconstructions in the discussion (eg all the lake level studies) because it would make it too broad and the paper is long enough as it is.

**Action:** None

- Conclusion lines 1196 and 1200**: boreal not Boreal;** line 1221**: comparison not comparsons**

**Response:** OK

**Action:** Boreal has been changed to boreal. Comparsons has been changed to comparisons.

-Figure3: **in the figure correct arborel pollen**

**Response:** OK

**Action:** The figure has been corrected

-Figure 5 and 7, **the name of the sites is too small, the title of each curve also, please correct it**

**Response:** OK

**Action:** The text in the figures has been increased from 16 point to 18 point

- figure A3: **this figure`s panels are included separately with their own captions. This will be unacceptable in the final paper**

**Response:** OK

**Action:** Figures A3a-A3-f have been relabelled A3-A8.